# Near-Optimal Private and Scalable $k$-Clustering

**Vincent Cohen-Addad**[1][*]    **Alessandro Epasto**[1]    **Vahab Mirrokni**[1]
**Shyam Narayanan**[2][†]    **Peilin Zhong**[1]
[1]Google Research    [2]MIT
{cohenaddad,aepasto,mirrokni,peilinz}@google.com
shyamsn@mit.edu

## Abstract

We study the differentially private (DP) $k$-means and $k$-median clustering problems of $n$ points in $d$-dimensional Euclidean space in the massively parallel computation (MPC) model. We provide two near-optimal algorithms where the near-optimality is in three aspects: they both achieve (1). $O(1)$ parallel computation rounds, (2). near-linear in $n$ and polynomial in $k$ total computational work (i.e., near-linear running time when $n$ is a sufficient polynomial in $k$), (3). $O(1)$ relative approximation and $\text{poly}(k, d)$ additive error. Note that $\Omega(1)$ relative approximation is provably necessary even for any polynomial-time non-private algorithm, and $\Omega(k)$ additive error is a provable lower bound for any polynomial-time DP $k$-means/median algorithm. Our two algorithms provide a tradeoff between the relative approximation and the additive error: the first has $O(1)$ relative approximation and $\sim (k^{2.5} + k^{1.01}\sqrt{d})$ additive error, and the second one achieves $(1 + \gamma)$ relative approximation to the optimal non-private algorithm for an arbitrary small constant $\gamma > 0$ and with $\text{poly}(k, d)$ additive error for a larger polynomial dependence on $k$ and $d$.

To achieve our result, we develop a general framework which partitions the data and reduces the DP clustering problem for the entire dataset to the DP clustering problem for each part. To control the blow-up of the additive error introduced by each part, we develop a novel charging argument which might be of independent interest.

## 1   Introduction

Over the last decade, the leakage of private information by machine learning and data mining algorithms has had dramatic consequences, from losses of billions of dollars [60] to even costing human lives [8]. Thus, protecting data privacy has become a top priority constraint in many modern machine learning and data mining problems.

This high demand has stimulated an important research effort to design algorithmic techniques enabling privacy-preserving algorithms. In recent years, the elegant notion of *differential privacy* (DP) [34] has become the gold standard for privacy-preserving algorithms [38, 67, 33, 1]. Informally, differential privacy requires the output (distribution) of the algorithm to remain almost the same under a small adversarial perturbation of the input.

$k$-Means and $k$-median clustering are fundamental and widely-studied problems in unsupervised learning. They are used to analyze and extract information from massive datasets in machine learning and data mining tasks. In particular, given a set of $n$ points $\mathcal{X} \subseteq \mathbb{R}^d$ within a ball of radius $\Lambda$, the goal is to find a set $\mathcal{C}$ of $k$ centers such that the clustering cost $\sum_{x \in \mathcal{X}} \min_{c \in \mathcal{C}} d^p(x, c)$ is minimized,

---

[*]Authors listed in alphabetical order.

[†]Work done as a student researcher at Google.

36th Conference on Neural Information Processing Systems (NeurIPS 2022).

where the power $p = 1, 2$ stands for $k$-median and $k$-means respectively. The importance of data privacy has sparked an important research effort on designing accurate and efficient differentially private $k$-means and $k$-median algorithms [15, 62, 39, 47, 58, 73, 64, 65, 71, 42, 5, 63, 52, 70, 69, 44, 53, 19, 61, 20, 14, 24]. In the above line of work, one can distinguish two separate efforts: one effort emphasizing the approximation guarantees, namely aiming at best possible approximation guarantee (under privacy constraints) in polynomial time, and a second effort targeting practical and efficient differentially private algorithms. More concretely, none of the above works achieve optimal accuracy and efficiency at the same time, i.e., the algorithms achieving $O(1)$ relative approximation and $\Lambda^p \cdot \text{poly}(k, \log(n))$ additive error have running time at least $n^{1+\Omega(1)} \cdot d$ (see e.g., [70, 44]), and the algorithms with running time proportional to $\tilde{O}(nd)^3$ and polynomial in the number of centers, $k$, have either relative approximation $\Omega(\log n)$ or additive error $\Lambda^p \cdot n^{\Omega(1)}$ (see e.g., [24, 14]). In this work, we reconcile these two lines of work and present two algorithms that both achieve near optimal approximation guarantee and near optimal running time simutaneously. Our two algorithms provide a tradeoff between the relative approximation and the additive error. In particular, the first algorithm runs in $\tilde{O}(nd) + \text{poly}(k)$ time and outputs a solution with $O(1)$ relative approximation and roughly $(k^{2.5} + k^{1.01}\sqrt{d}) \cdot \Lambda^p$ additive error (we ignore minor dependencies on $\log n$ and the privacy parameters $\varepsilon, \delta$ in the additive error). The second algorithm runs in $\tilde{O}(ndk) + \text{poly}(k)$ time and outputs a solution with $(1 + \gamma)\rho$ relative approximation and roughly $\text{poly}(k, d) \cdot \Lambda^p$ additive error for a larger polynomial dependence on $k, d$, where $\gamma > 0$ is an arbitrarily small constant and $\rho$ is the best relative approximation of any polynomial-time non-private algorithm. Our approximation guarantee is near optimal since the $\Omega(1)$ relative approximation is necessary even for any non-private polynomial-time algorithm [28, 27, 26, 55] and $\Omega(k \cdot \Lambda^p)$ is a provable lower bound for additive error achieved by any polynomial-time DP algorithm [47]. In addition, the approximation guarantees of our algorithms match the best previous DP algorithms [70, 44] while our algorithms have faster running time. In fact, our running time is almost tight when $k = n^{o(1)}$ because $\Omega(nd)$ time is necessary to read all input points.

Importantly, our algorithms are *scalable*. In the era of massive datasets, in-memory sequential algorithms struggle to handle billions of data points. Algorithms that are suitable for large-scale distributed/parallel computational systems such as MapReduce [32] are more desired. It motivates the study of algorithms in the *massively parallel computation* (MPC) model [54, 46, 11]. In the MPC model, there are multiple machines where each has local memory that is sublinear in the data size. The computation proceeds in rounds. In each round, each machine sends/receives messages to/from other machines but the size of messages sent/received by a machine cannot exceed its local memory. As long as the local memory per machine is at least polynomial in $k$ and at least $n^\theta$ for a small constant $\theta$, our algorithms can be easily implemented in the MPC model with $O(1)$ computation rounds and total space (total memory size across all machines) linear in the input data size. Note that $\Omega(k)$ local memory is needed for all previous non-private $o(\log n)$-round MPC $O(1)$-approximate $k$-means and $k$-median clustering algorithms (see e.g., [36, 40, 4, 6, 10, 18, 41]).

## 1.1 Other Related Work

$k$-Means and $k$-median clustering have seen a large body of work over the past few decades. There have been numerous works obtaining $O(1)$-approximation algorithms for both $k$-means and $k$-median, using either local search or primal-dual techniques. The state-of-the-art approximation algorithms are a $5.912$-approximation for Euclidean $k$-means and a $2.406$-approximation for Euclidean $k$-median, due to [25]. Two other fruitful methods of improving algorithms for these problems are coresets [21, 40, 51, 31, 29], which replace the dataset of points with a smaller set, and dimensionality reduction [22, 56], which reduces the number of dimensions of the ambient Euclidean space the points reside in.

There is a line of work studying $k$-means and $k$-median problems under parallel, distributed and streaming computational models. A popular way to tackle these clustering problems at scale is via coresets. A coreset is a small weighted subset of input points such that a good clustering solution for coreset yields a good approximate clustering to the original input point set. $k$-Means and $k$-median coresets have been extensively studied in the literature (e.g, [40, 6, 10, 18, 16, 41, 17]), and most of them can be implemented in the MPC model in $O(1)$ rounds as long as the memory per machine is at

---

[3] $\tilde{O}(f(n)) := f(n) \cdot \text{poly}(\log f(n))$

least $\mathrm{poly}(k)$, i.e., is large enough to hold the entire coreset. Other approaches to solve $k$-means and $k$-median in parallel are via quad-tree [13] or locality sensitive hashing [12]. These data partitioning based approaches require less local memory size but provide worse approximations or output more than $k$ centers (and so pertain to the line of work sacrificing approximation guarantees to practicality and efficiency). Notice that none of the previous scalable algorithms are differentially private.

Our paper focuses on differentially private $k$-means and $k$-median clustering in Euclidean space, which, as described previously, has seen significant work over the past several years. The paper [70] was the first paper to provide a differentially private algorithm for $k$-means clustering with $O(1)$ multiplicative ratio, and additive error polynomial in $k, d, \log n, \varepsilon^{-1}$, and $\log \delta^{-1}$. Later, [44] improved their result by obtaining an algorithm with multiplicative ratio arbitrarily close to the best non-private $k$-means (or $k$-median) approximation, at the cost of a much larger polynomial dependence on $k$ in the additive error. The paper [24] was the first paper to study private clustering in the Massively Parallel Computing framework, obtaining a polylogarithmic multiplicative ratio for $k$-means and $k$-median with polylogarithmic rounds of communication and computation. We also remark that [63, 70, 69, 19, 20] also provided private algorithms for $k$-means clustering in the local differential privacy model, but these algorithms all have additive errors proportional to at least $\sqrt{n}$.

## 1.2 Our Results

In this paper, we provide nearly optimal algorithms for differentially private $k$-means and $k$-median, in the MPC setting with $O(1)$ total rounds of communication and computation. We gave a brief, informal description of our results previously. We now formally state the two main theorems that we will prove about private clustering in the MPC model. In our MPC setting, we assume that we have $n^{1-\theta}$ machines, each of which can store $\tilde{O}(n^{\theta})$ points in $\mathbb{R}^d$, where $\theta > 0$ can be an arbitrarily small constant. (Equivalently, each machine has $\tilde{O}(n^{\theta} \cdot d)$ space and the total amount of space is $\tilde{O}(n \cdot d)$).

**Theorem 1.1.** (Theorem E.4.) *There exists an $(\varepsilon, \delta)$-DP algorithm for $k$-means (or $k$-median) clustering with multiplicative error $O(1)$ and additive error $(k^{2.5} + k^{1.01}\sqrt{d}) \cdot \mathrm{poly}\left(\log n, \varepsilon^{-1}, \log \delta^{-1}\right)$. In addition, assuming each machine can store $\tilde{O}(n^{\theta}) \geq (k^{1.5} + d^{0.5}) \cdot \mathrm{poly}(\log n, \varepsilon^{-1}, \log \delta^{-1})$ points, the algorithm can be implemented in MPC with $O(1)$ total rounds of communication and computation, total sequential running time $\tilde{O}(nd) + \mathrm{poly}(k, \varepsilon^{-1}, \log \delta^{-1})$, and total time per machine $\tilde{O}(n^{\theta}d) + \mathrm{poly}(k, \varepsilon^{-1}, \log \delta^{-1})$.*

**Theorem 1.2.** (Theorem E.9) *Suppose that there exists a polynomial-time algorithm that can compute a $\rho$-approximation to $k$-means (resp., $k$-median). Then, for any constant $\rho' > \rho$, there exists an $(\varepsilon, \delta)$-DP algorithm for $k$-means (resp., $k$-median) with multiplicative error $\rho'$ and additive error $\mathrm{poly}\left(k, d, \log n, \varepsilon^{-1}, \log \delta^{-1}\right)$. In addition, the algorithm can be implemented in MPC with $O(1)$ total rounds of communication and computation, total sequential time $\tilde{O}(nd)$, and total time per machine $\tilde{O}(n^{\theta}d)$, assuming each machine can store $\tilde{O}(n^{\theta}) \geq \mathrm{poly}(k, d, \log n, \varepsilon^{-1}, \log \delta^{-1})$ points.*

Our two results are similar to the results of [70] and [44], respectively, except that our algorithms are implementable in MPC with $O(1)$ rounds, and our algorithm has near-linear dependence on the size of the dataset $n$ (whereas [70] and [44] have polynomial dependence on $n$). We also remark that our additive error for the first algorithm is slightly worse than that of [70] (which had $k^{1.5} + k^{1.01}d^{0.51}$), and our additive error for the second algorithm matches that of [44].

Our results improve over the best MPC algorithm for private $k$-means and $k$-median [24], as we have an $O(1)$-approximation whereas their paper had a $\mathrm{poly}\log n$-approximation. In addition, our result only needs $O(1)$ rounds of communication between machines, whereas they require $\log n$ rounds for $k$-median and $\mathrm{poly}\log n$ rounds for $k$-means.

## 1.3 Technical Overview and Roadmap

In this subsection, we assume for simplicity that we are dealing with $k$-means. The overall outline for $k$-median is quite similar.

**Some natural approaches and why they don't work.** A first natural approach is to apply a uniform sampling approach. In other words, we can consider sampling $n^{\theta}$ points at random, moving them to a single machine, and then performing a sequential private clustering algorithm on this

machine. This idea of uniform sampling for private clustering was recently used by [14]. While they proved that this method ensures differential privacy, their additive error is quite large: they prove that with $T$ samples one can obtain total additive error of roughly $n/\sqrt{T} \cdot \Lambda^2$. Even if we sampled $n/2$ points at random, this equals $\sqrt{n} \cdot \Lambda^2$, whereas we wish for a much smaller additive error of $\text{poly}(k) \cdot \Lambda^2$. One can also construct examples, even when $k = 2$, where random sampling must induce large additive error. For instance, if we co-locate $n - \sqrt{n}$ points at the origin and co-locate $\sqrt{n}$ points at a point $\Lambda$ away from the origin, the optimal cost for 2-means is 0, but a random sample of $\sqrt{n}$ points, with constant probability, consists only of points located at the origin. Therefore, any algorithm applied to the sampled points will completely fail to recognize the second cluster, and will incur an $\Omega(\sqrt{n} \cdot \Lambda^2)$ error.

Another tempting approach to obtain an efficient and scalable DP algorithm with optimal approximation guarantees is to implement the $O(1)$-approximate private algorithms of [70] or [44] in the MPC setting. Unfortunately, both of these algorithms have an intrinsic sequential behavior that makes this approach difficult. The algorithm of [70] relies on a private local search procedure of [47], which appears very difficult to implement in MPC without a number of rounds of communication of $\Omega(k)$. The algorithm of [44] relies on iteratively finding and peeling off a "privately"-dense ball of points. This method requires roughly $O(k \log n)$ iterations, and again appears very difficult to implement in $O(1)$ parallel rounds.

**Our approach:** we describe our overall procedure, and give intuition for some of the details.

---

**Overall procedure.** Our overall algorithm can be split into three main steps.

1. Reduce the dimensionality of the problem $d$ to $O(\log k)$. This will remove large dependencies on the dimension for both the runtime and additive error.

2. Compute in MPC a private and very efficient but *weak* $k$-means (or $k$-median) solution: Namely a solution with an $\omega(1)$-approximation guarantee and that may use up to $O(k \log^2 n)$ centers instead of just $k$ centers.

3. Combine the weak but efficient MPC solution with an $O(1)$-approximate private sequential algorithm to obtain an efficient $O(1)$-approximate MPC algorithm.

---

We remark that the first and third steps can be done using two different but related methods. The first method obtains an $O(1)$-approximation with additive error proportional to approximately $k^{2.5} + k^{1.01}\sqrt{d}$ (i.e., Theorem 1.1). The second method obtains a approximation factor within an arbitrary factor $\gamma$ of even the best known non-private and sequential algorithm, but with additive error proportional to approximately $k^{\tilde{O}(1/\gamma^2)}$ (i.e., Theorem 1.2). We discuss the second step in Section 3, the first method of doing the third step in Section 4, the second method of doing the third step in Section 5, and both ways of doing the first step in Section 6. We now just focus on a key idea for the third step, and will give more algorithmic description and intuition in the later sections of this paper. All formal proofs are deferred to the appendix.

**Charging Additive Error to Multiplicative Error.** As previously mentioned, a direct sampling approach fails because it incurs a too large additive error. However, we show that we can still use sampling if we sort the points appropriately and do not sample each point with the same probability. The first idea we use is inspired by Chen [21], which is to start with our weak MPC centers, and map each point $x$ to a *bucket* $(j, r)$, where $j$ represents the closest of the weak centers to $x$, and $r$ approximates the distance from $x$ to this center. Our next idea is to show that if we apply a private clustering procedure on a random sample of a fixed size in each bucket, we can charge the large additive cost obtained to a small constant-factor blowup in the multiplicative cost. The intuition for this is that if the average cost of each point is roughly $R$, then most points in our weak approximation are still roughly within $O(R)$ of the weak centers, and therefore get mapped to buckets represented by much smaller balls. So, we can then apply a private sequential algorithm on these buckets with total error depending on $R^2$ rather than on $\Lambda^2$. We will show that, by using properties of the initial weak solution, the large additive error can be replaced with a small multiplicative error, even if the sample size $T$ for each bucket is small.

To explain the above intuition further, in our analysis we apply a Chernoff bound and some properties of high-dimensional Euclidean space to show that if there are $m$ points in a bucket all within radius

$R$ of each other, we accumulate roughly $R^2 \cdot m/\sqrt{T}$ additive error, as opposed to a more naive $\Lambda^2 \cdot m/\sqrt{T}$ error (which would be obtained by [14]). We can then evenly distribute the error between each of the $m$ points, to obtain $R^2/\sqrt{T}$ error per point. We charge the error of each point to the squared distance of each point to its center in the weak approximation, which is roughly equal to $R^2$. So, when we add over all points, the error across all points is roughly $1/\sqrt{T}$ times the overall cost of the weak approximation. So, if the weak solution had cost at most $\alpha$ times the optimal $k$-means cost, if the sampling size $T$ is much bigger than $\alpha^2$ then the total error is much smaller than the optimal cost. Hence, we are able to charge the total additive error into a small multiplicative error!

## 2 Preliminaries

We present some basic definitions and setup that will be sufficient for explaining our algorithms for the main body of the paper. We defer the full set of preliminaries to Appendix A.

### 2.1 Differential Privacy

**Definition 2.1** ([34])**.** A (randomized) algorithm $\mathcal{A}$ is said to be $(\varepsilon, \delta)$-*differentially private* $((\varepsilon, \delta)$-DP for short) if for any two "adjacent" datasets $\mathcal{X}$ and $\mathcal{X}'$ and any subset $S$ of the output space of $\mathcal{A}$, we have

$$\mathbb{P}(\mathcal{A}(\mathcal{X}) \in S) \leq e^{\varepsilon} \cdot \mathbb{P}(\mathcal{A}(\mathcal{X}') \in S) + \delta.$$

In our setting, we say that two datasets $\mathcal{X}$ and $\mathcal{X}'$ are *adjacent* if we can convert $\mathcal{X}$ to $\mathcal{X}'$ either by adding, removing, or changing a single data point. We remark that in all of our algorithms, we will implicitly assume that $\varepsilon, \delta \leq \frac{1}{2}$.

A ubiquitous method we use to ensure privacy is the Laplace Mechanism. Simply, this is a method where we privatize a statistic of the data by adding noise $\mathrm{Lap}(t)$ to the statistic for some $t > 0$, where $\mathrm{Lap}(t)$ has the PDF $\frac{1}{2t} \cdot e^{-|x|/t}$. It is well-known that if $f(\mathcal{X})$ is a statistic that never changes by more than $\Delta$ between any two adjacent datasets $\mathcal{X}, \mathcal{X}'$, then $f(\mathcal{X}) + \mathrm{Lap}(\Delta/\varepsilon)$ is $(\varepsilon, 0)$-DP.

### 2.2 The Massively Parallel Computation (MPC) Model

In the MPC model, there are multiple machines where each machine has sublinear local memory. The input data points are distributed arbitrarily on the machines before computation starts. The computation proceeds in rounds. In each round, each machine reads its local data and performs computations. At the end of each round, each machine sends/receives messages to/from other machines. The total size of messages sent/received by a machine in each round does not exceed the local memory size. The output is distributed on machines at the end of the computation. The MPC algorithm with small number of rounds and small total space (total memory across all machines) is desired. Furthermore, scalability is also considered in many applications and thus we want the local memory required by the algorithm as small as possible. There are many scalable MPC algorithmic primitives. The most basic one is sorting.

**Theorem 2.2** ([46, 45])**.** *There is an MPC algorithm which sorts $N$ data items in $O(1)$ rounds using $O(N)$ total space. The local memory per machine required is at most $O(N^{\theta})$ for arbitrary small constant $\theta > 0$.*

Based on the sorting primitive above, other basic subroutines such as indexing, prefix sum and set aggregation can be easily implemented in the MPC model with same complexity as sorting. We refer readers to Appendix E of [3] for more basic MPC algorithmic primitives.

### 2.3 $k$-Means and $k$-Median Clustering

For two points $x, y \in \mathbb{R}^d$, we define $d(x, y)$ to be the Euclidean distance between $x$ and $y$. For a set $\mathcal{C} \subset \mathbb{R}^d$ of points, we define $d(x, \mathcal{C}) = d(\mathcal{C}, x)$ to be $\min_{c \in \mathcal{C}} d(x, c)$.

In both $k$-means and $k$-median clustering, we are given a dataset of points $\mathcal{X} = \{x_1, \ldots, x_n\}$ in $d$-dimensional Euclidean space $\mathbb{R}^d$. We further assume that the points in $\mathcal{X}$ are in $B(0, \Lambda)$, which is the ball of radius $\Lambda$ about the origin in $\mathbb{R}^d$. Our goal in $k$-means (resp., $k$-median) clustering is

to find a subset $\mathcal{C}$ of $k$-points that minimizes the cost of $\mathcal{X}$ with respect to $\mathcal{C}$. Specifically, for a set of centers $\mathcal{C}$, we define $\text{cost}(\mathcal{X}; \mathcal{C}) := \sum_{x \in \mathcal{X}} d(x, \mathcal{C})^p$, where $p = 2$ in the setting of $k$-means and $p = 1$ in the setting of $k$-median. Occasionally, we may assign each point $x_i \in \mathcal{X}$ a positive weight $w_i$, in which case we define $\text{cost}(\mathcal{X}; \mathcal{C}) := \sum_{x_i \in \mathcal{X}} w_i \cdot d(x_i, \mathcal{C})^p$. We also define $\text{OPT}(\mathcal{X})$ to be the minimum value of $\text{cost}(\mathcal{X}; \mathcal{C})$ for any set of $k$ points $\mathcal{C}$.

Our goal in differentially private clustering is to produce a set of $k$ centers $\mathcal{C}$ such that $\mathcal{C}$ is $(\varepsilon, \delta)$-DP with respect to $\mathcal{X}$, and such that $\text{cost}(\mathcal{X}; \mathcal{C}) \leq \alpha \cdot \text{OPT}(\mathcal{X}) + V \cdot \Lambda^p$ (where $p = 2$ for $k$-means and $p = 1$ for $k$-median). If we obtain this guarantee, we say that we have an $(\varepsilon, \delta)$-DP algorithm for $k$-means (or $k$-median) with multiplicative ratio $\alpha$ and additive error $V$. In the MPC model, our goal is to make a machine output the set of $k$ centers at the end of the algorithm.

Finally, we briefly discuss coresets. For a dataset $\mathcal{X} \subset B(0, \Lambda)$ and for some $\gamma \leq \frac{1}{2}$ and $W \geq 0$, a $(\gamma, W)$-*coreset* is a dataset $\mathcal{Y}$ that estimates $\mathcal{X}$ with respect to $k$-means (or $k$-median) clustering, i.e., $(1 - \gamma) \cdot \text{cost}(\mathcal{X}; \mathcal{C}) - W \cdot \Lambda^p \leq \text{cost}(\mathcal{Y}; \mathcal{C}) \leq (1 + \gamma) \cdot \text{cost}(\mathcal{X}; \mathcal{C}) + W \cdot \Lambda^p$ for all subsets $\mathcal{C} \subset B(0, \Lambda)$ of size at most $k$. We will also want the dataset $\mathcal{Y}$ to have much fewer distinct points than $\mathcal{X}$, though each point in $\mathcal{Y}$ may have large multiplicity. In the setting of non-private coresets, we usually have $W = 0$, in which case we write $\gamma$-coreset to mean $(\gamma, 0)$-coreset. We note the following theorem regarding MPC algorithms for non-private coresets.

**Theorem 2.3** ([17, 40, 41]). *Consider $n$ points in $\mathbb{R}^d$. There exists a non-private MPC algorithm which outputs a $k$-means/$k$-median $\gamma$-coreset with size $\text{poly}(k, d, \log n, 1/\gamma)$ in $O(1)$ rounds for $\gamma \in (0, 0.5)$. The total space needed is $O(nd) + \text{poly}(k, d, \log n, 1/\gamma)$, and the local memory per machine required is $\text{poly}(k, d, \log n, 1/\gamma)$. In addition, the total running time over all machines is at most $O(nd) + \text{poly}(k, d, \log n, 1/\gamma)$.*

## 3 A Preliminary Bicriteria Approximation

A key tool in all of our $k$-means and $k$-median algorithms is an initial, crude approximation for $k$-means (or $k$-median) clustering. Specifically, we start with a bicriteria approximation, which means that the algorithm may output a list of more than $k$ centers. Our algorithm is very efficient, only needing constant rounds of communication and computation, and essentially optimal time per machine.

**Theorem 3.1.** *There exists an $(\varepsilon, \delta)$-DP algorithm that, given a dataset $\mathcal{X} \subset B(0, \Lambda)$ of size $n$, computes a set of $O(k \log^2 n)$ points $\mathcal{F}$ that provides a $d^{O(1)}$-approximation to the best $k$-means (or $k$-median) clustering with additive error $k \cdot \text{poly}(\log n, d, \varepsilon^{-1}, \log \delta^{-1})$. In addition, the algorithm can be implemented in MPC with $O(1)$ rounds of communication and computation, and $\tilde{O}(n^\theta d)$ time per machine.*

Our techniques for this section heavily rely on a data structure called a Quadtree. This data structure involves creating a nested series of grids partitioning $\mathbb{R}^d$. The quadtree has varying levels, where the 0th level is a very coarse grid, and each subsequent level further refines the previous level with smaller grid pieces. The Quadtree can be used to embed the input points into a so-called "Hierarchically Separated Tree" (HST) metric, for which computing $k$-median cost is often much simpler than computing $k$-median in Euclidean space. Indeed, [24] uses this embedding, along with a dynamic programming approach, to provide a private approximation to $k$-median with multiplicative approximation $O(d^{3/2} \log n)$ and additive approximation $O(k) \cdot \text{poly}(\log n, d, \varepsilon^{-1})$.

Unfortunately, we cannot directly use their approach for two reasons. The first is that their dynamic programming approach requires roughly $\log n$ rounds of communication across machines for $k$-median (and $\text{poly} \log n$ rounds for $k$-means), whereas we wish for $O(1)$ rounds. The second is that their approach runs into a barrier when the memory per machine is below roughly $\sqrt{n}$, whereas we want an MPC algorithm even if each machine can only store $n^\theta$ points for an arbitrarily small $\theta > 0$.

Our method of developing a bicriteria approximation is inspired by the Quadtree and embeddings into this HST metric, but we avoid the issues above by using a greedy approach rather than a dynamic programming one. Specifically, we will consider each level of the Quadtree grid separately, and map every point $x \in \mathcal{X}$ to the grid center $x$ lies in. We then approximately choose the $O(k)$ centers that have the most points in them, using a private selection mechanism. We will have to do this for logarithmically many levels, which causes us to have a bicriteria approximation. To analyze this procedure, we avoid looking at the HST metric and instead consider the number of points $n_r$ that are

of distance $r$ away from some center for every choice of radius $r$. We then use an integration by parts-based idea to analyze the $k$-means (or $k$-median) cost based on summing a weighted combination of $n_r$ over $r$, which we use to establish our accuracy bounds. Our greedy method can be implemented in $O(1)$ rounds of communication and computation for both $k$-means and $k$-median.

The full details of the algorithm and analysis, including pseudocode, are deferred to Appendix B.

## 4 A Constant Approximation in MPC

In this section, we describe how to combine our bicriteria approximation from Section 3, along with the sequential private $O(1)$-approximate algorithm from [70], to obtain a parallel $O(1)$-approximate algorithm using only $O(1)$ rounds of communication and computation. Specifically, we prove the following.

**Theorem 4.1.** *There exists an $(\varepsilon, \delta)$-DP algorithm for $k$-means (or $k$-median) with multiplicative ratio $O(1)$ and additive error $k^{2.5} \cdot \text{poly}\left(d, \log n, \varepsilon^{-1}, \log \delta^{-1}\right)$. In addition, assuming each machine stores $\tilde{O}(n^\theta) \geq k^{1.5} \cdot \text{poly}(d, \log n, \varepsilon^{-1}, \log \delta^{-1})$ points, the algorithm can be implemented in MPC with $O(1)$ rounds of communication and computation, total sequential running time $\tilde{O}(nd) + \text{poly}(k, d, \varepsilon^{-1}, \log \delta^{-1})$, and total time per machine $\tilde{O}(n^\theta d) + \text{poly}(k, d, \varepsilon^{-1}, \log \delta^{-1})$.*

We remark that large polynomial dependencies in $d$ are not crucial, as we will use dimensionality reduction in Section 6 to set $d = O(\log k)$.

Our algorithm is inspired by the coreset framework of Chen [21], who shows how to convert a bicriteria approximation into a coreset for clustering (in the sequential, non-private setting). The rough method is to start with an approximate set of cluster centers $\mathcal{F}$ of $\beta \cdot k$ points for some $\beta > 1$, and map each point $x \in \mathcal{X}$ to its nearest neighbor in $\mathcal{F}$. Then, the points in $\mathcal{X}$ are split into "rings", based on their closest point and their approximate distance to that closest point. Finally, the points in each ring are replaced by a uniform sample of a few points in each ring, to create a small coreset.

Ordinarily, when performing DP $k$-means/$k$-median over $n$ points, we incur additive cost proportional to $\Lambda^p$, where $\Lambda$ is the diameter of the pointset. Hence, naive application of using private $k$-means/$k$-median on a random subset (or even a coreset) of $T$ points is quite bad, as we have to scale each point by a factor of $n/T$ for the weights to match, which induces a $n/T \cdot \Lambda^p$ additive error. In reality, the induced additive error is actually worse, and behaves like $n/\sqrt{T} \cdot \Lambda^p$. However, an important insight we have, as noted in Subsection 1.3, is that when using the framework of Chen [21], we can charge much of the additive cost to multiplicative cost. Namely, if we perform a private $k$-means/$k$-median algorithm on a random sample of $T$ points in each ring separately, if the ring had radius $R \ll \Lambda$, we would roughly incur additive error proportional to $R^p/\sqrt{T}$ per point, rather than $\Lambda^p/\sqrt{T}$.

We then use the properties of the bicriteria approximation to show that if we add this error across all rings, the additive error is actually only a small multiple of the optimal cost, as opposed to a much worse $n/\sqrt{T} \cdot \Lambda^p$. Recall that if $R_i$ represents the approximate distance of each point $x_i$ to its bicriteria center, then $\sum_i R_i^p$ is at most $d^{O(1)}$ times more than the optimal $k$-means cost (plus some small multiple of $\Lambda^p$), by Theorem 3.1. Hence, if $T$ is roughly $d^{O(1)}$, we can get the overall additional error induced by each $R^p/\sqrt{T}$ error per point is at most $O(1)$ times the optimal $k$-means cost. This is what allows us to get a multiplicative approximation with small additive cost. We remark that some rings may have fewer than our desired $T$ points, in which case we cannot sample $T$ random points and incur an additional small additive cost. Finally, is quite simple to sample $T$ points in each ring in parallel, and map each sample to a separate machine. We can then use a sequential private $k$-means algorithm for each sample separately, as we can store the $T$ points on a single machine.

One additional issue is that this procedure has to find $k$ points for each ring, and there will end up being roughly $O(k \cdot \text{poly} \log n)$ rings. So this means we need $O(k^2 \cdot \text{poly} \log n)$ centers, whereas we only want $k$ centers. Our way of fixing this is to convert our $k$ centers into a "semi-coreset" by mapping each point in a ring to its closest point in the corresponding private set of centers, and then adding Laplace noise to the multiplicities of the set. These semi-coresets will be private, which means we can apply a non-private MPC $k$-means algorithm on the union of all the semi-coresets to generate a final set of points, since the semi-coresets have already been privatized. Also, while they are not true coresets, we show that any $O(1)$-approximate $k$-means algorithm applied on the union of all the semi-coresets still provides an $O(1)$-approximate solution for the original dataset.

We present the algorithm pseudocode in below in Algorithm 1, and defer the full analysis to Appendix C. We remark that we replace some instances of mapping points to the nearest center with mapping them to an approximately nearest center, which is useful in speeding up runtime.

---

**Algorithm 1** A constant approximation algorithm for differentially private $k$-means (or $k$-median) in MPC.

1: **procedure** CONSTANTAPPROX($\mathcal{X}, \varepsilon, \delta$)            ▷ Will be $(O(\varepsilon), O(\delta))$-DP.
2:  Let $T = \text{poly}(d, \log n, \varepsilon^{-1}, \log \delta^{-1}) \cdot k^{1.5}$ be a sufficiently large threshold parameter.
3:  Let $\mathcal{F} = \{f_1, \ldots, f_{\beta \cdot k}\}$ be an $(\varepsilon, \delta)$-DP bicriteria approximation for $k$-clustering of $\mathcal{X}$, using Theorem 3.1, and where $\beta = O(\log^2 n)$.
4:  **for** all $x_i \in \mathcal{X}$ **do**
5:    Assign $x_i$ to bucket $(j, r)$ if $x_i$ has $f_j$ as a $O(\log n)$-approximate nearest neighbor, with $d(x_i, f_j) \approx \frac{\Lambda}{2^n} \cdot 2^r$.
6:  **for** all $j \leq \beta \cdot k, r \leq O(\log n)$ **do**
7:    Let $\mathcal{X}_{j,r}$ be set of points $x_i$ assigned to $(j, r)$.
8:    Let $\hat{N}_{j,r} := |\mathcal{X}_{j,r}| + \text{Lap}(1/\varepsilon)$.
9:    **if** $\hat{N}_{j,r} \geq 2T$ **then**
10:      Sample $T$ random points $\mathcal{Y}_{j,r}$ from $\mathcal{X}_{j,r}$.
11:      Send $\mathcal{Y}_{j,r}$ to one machine, and use [70] on each machine to find a $k$-clustering $\mathcal{G}_{j,r}$.
12:      Replace each point in $\mathcal{Y}_{j,r}$ with an $O(1)$-approximate nearest neighbor in $\mathcal{G}_{j,r}$.
13:      Add $\text{Lap}(1/\varepsilon)$ noise to the multiplicity of each point in the replaced data set, to create dataset $\hat{\mathcal{Y}}_{j,r}$.
14:    **else**
15:      Let $\hat{\mathcal{Y}}_{j,r} = \emptyset$.
16:  Let $\hat{\mathcal{Y}} = \bigcup_{j,r} \hat{\mathcal{Y}}_{j,r}$.
17:  Return a non-private MPC $k$-means clustering algorithm on $\mathcal{Y}$, using, e.g., [17].

---

## 5 An Arbitrarily Good Approximation in MPC

In this section, we present a different method of combining the bicriteria approximation from Section 3 with a sequential private $O(1)$-approximation from [70] to obtain a parallel approximate algorithm achieving arbitrarily close multiplicative ratio to the best non-private sequential algorithm. While we obtain an improved multiplicative approximation ratio, we suffer a larger additive error. Specifically, we prove the following.

**Theorem 5.1.** *Suppose that there exists a polynomial-time algorithm that can compute a $\rho$-approximation to $k$-means (resp., $k$-median). Then, for any constant $\rho' > \rho$, exists an $(\varepsilon, \delta)$-DP algorithm for $k$-means (resp., $k$-median) with multiplicative ratio $\rho'$ and additive error $\text{poly}\left(k, e^d, \log n, \varepsilon^{-1}, \log \delta^{-1}\right)$. In addition, assuming each machine can store $\tilde{O}(n^\theta) \geq \text{poly}(k, e^d, \varepsilon^{-1}, \log \delta^{-1})$ points, the algorithm is implementable in MPC with $O(1)$ total rounds of communication and computation, and $O(n^\theta k d)$ time per machine.*

We remark that while there is an exponential dependence on the dimension $d$ in both the runtime and additive error, in Section 6, we show how to replace the $e^d$ with $\text{poly}(k)$ additive error.

Our overall procedure has two steps. The first step is to convert an $O(1)$-approximate private sequential algorithm into a private sequential coreset: this is somewhat similar to Ghazi et al. [44], which provides a similar guarantee but has a runtime dependence that is polynomial in $n$ as opposed to near-linear in $n$. The second step is to convert a private sequential coreset into a private parallel coreset.

For the first step, we start with a private $O(1)$-approximate clustering, and as in Section 3, we split the dataset $\mathcal{X}$ into roughly $O(k \log n)$ rings based on each point's closest center and the distance to that center. For each ring, we use a result of Ghazi et al. [44] inspired by the theory of error-correcting codes, which forms a "cover" of each ring of size roughly exponential in the dimension, which is a method of selecting points such that every point in the ring is reasonably close to at least 1 point in the cover. We can create the cover in parallel, and then map every point to its closest point in the

cover in $e^{O(d)}$ time. We can use this cover and a Laplace mechanism to construct a private coreset of the data, though the coreset will have size exponential in the dimension.

For the second step, we apply a similar method to Section 4, again using the insight of charging additive error to multiplicative error. We again start with our weak parallel coreset from Section 3, partition the points into rings, and sample uniformly from each ring which allows each ring's sample to fit on a single machine. Now, for each sample, we apply the private coreset procedure from the first step, and together we obtain a large private coreset. We finally apply a non-private MPC clustering algorithm on the already privatized coreset.

The full details of the algorithm and analysis, including pseudocode, are deferred to Appendix D.

## 6 Dimensionality Reduction

Our algorithm for dimensionality reduction relies on [56], which shows that a random projection $\Pi$ down to $O(\log k)$ dimensions can approximate every $k$-clustering of $n$ points simultaneously. While this may seemingly imply that we can automatically assume $d = O(\log k)$, this is not so obvious, because once we have found a set of centers $\mathcal{C}'$ in $\mathbb{R}^{O(\log k)}$, it is unclear how to pull this back up to a set of centers in $\mathbb{R}^d$.

Ghazi et al. [44] use dimensionality reduction to improve their exponential dependence on $d$ to a polynomial dependence on $k$ (as $e^{O(\log k)} = \text{poly}(k)$). They start by constructing a random projection $\Pi$ down to $O(\log k)$ dimensions and compute a private set $\mathcal{C}' = \{c'_1, \ldots, c'_k\}$ in the reduced dimension. They then map each point $x_i$ to a cluster $\mathcal{X}_j$ based on which point $c'_j \in \mathcal{C}'$ is closest to $\Pi x_i$. Finally, to obtain a set of centers in high dimensions, they compute a private average (for $k$-means) or geometric median (for $k$-median) of the points in each cluster $\mathcal{X}_j$.

The main issue in applying this method is that we wish for a very fast scalable algorithm that works in MPC, and it is unknown how to compute a private geometric median that is sufficiently scalable, i.e., runs in $O(1)$ MPC rounds and $\tilde{O}(nd)$ total work. Our first observation in obtaining an efficient algorithm is that, rather than computing a geometric median of the points, a coordinate-wise median serves as a reasonable proxy for the geometric median. Specifically, it has the important property of being within distance $O(r)$ of the geometric median of $\tilde{X}_j$ if at least 2/3 of the points in $\mathcal{X}_j$ are within $r$ of the geometric median [59]. In addition, the coordinate-wise median can be computed on a small sample without significant error, and can be computed privately without much error either. We can therefore use an approximate coordinate-wise median for each cluster $\mathcal{X}_j$ to obtain an $O(1)$-approximate private and scalable clustering with minimal dependence on the dimension.

If we want arbitrarily good approximation, such as in Section 5, the coordinate-wise median is unfortunately not strong enough. To fix this, our rough intuition is to compute a private geometric median of a small uniform subsample of points, but only among those points close to the private coordinate-wise median. This idea of uniform subsampling to speed up runtime has been used in several previous algorithms for $k$-means, such as [23, 14, 30, 37]. As we perform the geometric median on a small sample, a polynomial-time algorithm is sufficient. In addition, by restricting ourselves to points close to the coordinate-wise median, we are able to replace additive error proportional to $\Lambda$ with additive error proportional to some smaller radius $R$, which we can charge to multiplicative error in a similar manner as in Sections 4 and 5. We remark that this algorithm may take $\tilde{O}(ndk)$ total time instead of $\tilde{O}(nd)$ time, but we fix this by performing another uniform subsampling trick at the beginning of the algorithm, which improves the runtime at the cost of a slight blowup in the additive error.

Overall, we obtain our main theorems of this paper, Theorems 1.1 and 1.2, which correspond to dimension reduction applied to Theorems 4.1 and 5.1, respectively.

We provide pseudocode for Theorem 1.1 below, in Algorithm 2. The full algorithm and analysis for both theorems, as well as pseudocode for Theorem 1.2, are deferred to Appendix E.

**Algorithm 2** A constant approximation algorithm for differentially private $k$-means (or $k$-median) in MPC, with improved dependence on $d$.

---

1: **procedure** CONSTANTAPPROXHIGHDIM$(\mathcal{X}, \varepsilon, \delta)$         $\triangleright$ Will be $(O(\varepsilon), O(\delta))$-DP.
2:   Let $T = O(\log(nd) \cdot \varepsilon^{-1}\sqrt{d \log \delta^{-1}})$ be a threshold parameter.
3:   Let $d' = O(\log k)$, and pick a random projection $\Pi \in \mathbb{R}^{d' \times d}$.
4:   Use Algorithm 1 to find a $k$-means clustering $\mathcal{C}' = \{c_1', \ldots, c_k'\}$ of $\Pi\mathcal{X} = \{\Pi x_1, \ldots, \Pi x_n\}$.
5:   Let $S \subset [n]$ be a random sample where each $i \in S$ with probability $k^{-0.01}$.
6:   Map each point $\Pi x_i : i \in S$ to a 10-approximate nearest neighbor in $\mathcal{C}'$.
7:   **for** $j = 1$ to $k$ **do**
8:     Let $\mathcal{X}_j$ be the set of points $x_i$ such that $\Pi x_i$ is mapped to $c_j'$.
9:     Let $\hat{N}_j = |\mathcal{X}_j| + \text{Lap}(1/\varepsilon)$.
10:     **if** $\hat{N}_j \geq 2T$ **then**
11:       Sample $T$ points in $\mathcal{X}_j$, and let $c_j$ be a private coordinate-wise median of the sampled points, where we use $(\varepsilon/(2\sqrt{d \log \delta^{-1}}), 0)$-DP in each direction.
12:     **else**
13:       Let $c_j$ be the origin.
14:   Return $\mathcal{C} = \{c_1, \ldots, c_k\}$

---

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
