# A Preliminaries

In this section, we restate some of the preliminaries from Section 2, and also include some new important definitions and preliminary results that are of use. First, we recall that $\Lambda > 0$ is some fixed real number and $B(0, \Lambda)$ is the ball of radius $\Lambda$ around the origin, which we assume all points in our dataset $\mathcal{X}$ will be part of. We will use $p$ to represent the exponent $p = 1$ for $k$-median and $p = 2$ for $k$-means.

We use the phrase *with overwhelming probability* to denote probability at least $1 - 1/n^C$, where $C$ can be an arbitrarily large constant.

## A.1 Differential Privacy

Recall the definition of differential privacy from Section 2.

**Definition A.1** ([34]). A (randomized) algorithm $\mathcal{A}$ is said to be $(\varepsilon, \delta)$-*differentially private* ($(\varepsilon, \delta)$-DP for short) if for any two "adjacent" datasets $\mathcal{X}$ and $\mathcal{X}'$ and any subset $S$ of the output space of $\mathcal{A}$, we have
$$\mathbb{P}(\mathcal{A}(\mathcal{X}) \in S) \leq e^\varepsilon \cdot \mathbb{P}(\mathcal{A}(\mathcal{X}') \in S) + \delta.$$
When $\delta > 0$, this is often referred to as *approximate* differential privacy, as opposed to *pure* differential privacy when $\delta = 0$.

In our setting, we say that two datasets $\mathcal{X}$ and $\mathcal{X}'$ are *adjacent* if we can convert $\mathcal{X}$ to $\mathcal{X}'$ either by adding, removing, or changing a single data point. We remark that in all of our algorithms, we will implicitly assume that $\varepsilon, \delta \leq \frac{1}{2}$.

In Section 2, we described a simplified *Laplace mechanism* for approximating functions $f : \mathcal{X} \to \mathbb{R}$. We now describe a more generalized Laplace mechanism for approximating a function $f : \mathcal{X} \to \mathbb{R}^m$, where $m \geq 1$. The Laplace mechanism works by replacing the $i$th coordinate $f(\mathcal{X})_i$ with $f(\mathcal{X})_i + \mathrm{Lap}(T)$ for some choice of $T > 0$, where $\mathrm{Lap}(T)$ is the Laplace distribution with PDF $\frac{1}{2T} \cdot e^{-|x|/T}$ at $x$. Importantly, the choices of $\mathrm{Lap}(T)$ for each dimension $i \in [m]$ is chosen independently. It is well-known (see, for instance, [35] or [72]) that if a function $f$ has *sensitivity* $\Delta$, meaning that $\|f(\mathcal{X}) - f(\mathcal{X}')\|_1 \leq \Delta$ for all adjacent $\mathcal{X}, \mathcal{X}'$, then the Laplace Mechanism with parameter $T$ is $(\Delta/T, 0)$-DP. We remark that we often will just use the simpler Laplace mechanism where $m = 1$. In this case, note that the sensitivity just means an upper bound on $|f(\mathcal{X}) - f(\mathcal{X}')|$ for adjacent $\mathcal{X}, \mathcal{X}'$.

Similar to the Laplace Mechanism, we can also implement the *Truncated Laplace mechanism*[43]: we will only use it for $m = 1$, i.e., for approximating functions $f : \mathcal{X} \to \mathbb{R}$. If $f$ has sensitivity $\Delta$, the Truncated Laplace Mechanism outputs $f(\mathcal{X}) + \mathrm{TLap}(\Delta, \varepsilon, \delta)$, where $\mathrm{TLap}(\Delta, \varepsilon, \delta)$ is the distribution with PDF proportional to $e^{-|x| \cdot \varepsilon / \Delta}$ on the region $[-A, A]$, where $A = \frac{\Delta}{\varepsilon} \cdot \log\left(1 + \frac{e^\varepsilon - 1}{2\delta}\right)$. Assuming $\varepsilon, \delta \leq \frac{1}{2}$, is known that if $f$ has sensitivity $\Delta$, then this mechanism is $(\varepsilon, \delta)$-DP, and is always accurate up to error $\frac{\Delta}{\varepsilon} \cdot \log \frac{1}{\delta}$.

Next, we note two classic theorems regarding the privacy of composing private mechanisms (see, for instance, [35] or [72]).

**Theorem A.2** (Basic Adaptive Composition). *Let $\mathcal{A}_1, \ldots, \mathcal{A}_k$ be adaptive mechanisms on a dataset $\mathcal{X}$ such that each $\mathcal{A}_i$ is $(\varepsilon_i, \delta_i)$-differentially private as a function of $\mathcal{X}$, assuming that the previous outputs $\mathcal{A}_1, \ldots, \mathcal{A}_{i-1}$ are fixed. Then, the mechanism $\mathcal{A}$ which concatenates the outputs of $\mathcal{A}_1, \ldots, \mathcal{A}_k$ is $(\sum \varepsilon_i, \sum \delta_i)$-differentially private.*

**Theorem A.3** (Strong Adaptive Composition). *Let $\mathcal{A}_1, \ldots, \mathcal{A}_k$ be adaptive mechanisms on a dataset $\mathcal{X}$ such that each $\mathcal{A}_i$ is $(\varepsilon, \delta)$-differentially private as a function of $\mathcal{X}$, assuming that the previous outputs $\mathcal{A}_1, \ldots, \mathcal{A}_{i-1}$ are fixed. Then, for any $\delta' > 0$, the mechanism $\mathcal{A}$ which concatenates the outputs of $\mathcal{A}_1, \ldots, \mathcal{A}_k$ is $(\sqrt{2k \log \delta^{-1}} \cdot \varepsilon + k\varepsilon(e^\varepsilon - 1), k\delta + \delta')$-differentially private.*

## A.2 Clustering

First, we recall the basic definitions relating to distance and clustering cost.

**Definition A.4.** For two points $x, y \in \mathbb{R}^d$, we define $d(x, y) := \|x - y\|_2$ to be the Euclidean distance between $x$ and $y$. In addition, for a set $\mathcal{C} \subset \mathbb{R}^d$ of points and a point $x \in \mathbb{R}^d$, we define $d(x, \mathcal{C}) = d(\mathcal{C}, x)$ to be $\min_{c \in \mathcal{C}} d(x, c)$.

**Definition A.5.** For a set $\mathcal{X} \subset \mathbb{R}^d$ and a set of points $\mathcal{C} \subset \mathbb{R}^d$ of size $k$, we define the $k$-means cost $\mathrm{cost}(\mathcal{X}; \mathcal{C}) := \sum_{x \in \mathcal{X}} d(x, \mathcal{C})^2$. Likewise, we define the $k$-median cost $\mathrm{cost}(\mathcal{X}; \mathcal{C}) := \sum_{x \in \mathcal{X}} d(x, \mathcal{C})$. Occasionally, we may assign each point $x_i \in \mathcal{X}$ a positive weight $w_i$, in which case we define $\mathrm{cost}(\mathcal{X}; \mathcal{C}) := \sum_{x_i \in \mathcal{X}} w_i \cdot d(x_i, \mathcal{C})^p$ ($p = 1$ for $k$-median and $p = 2$ for $k$-means). If the context is clear, we will not specify whether we are talking about $k$-means or $k$-median cost.

In addition, we define $\mathrm{OPT}(\mathcal{X}) = \min_{\mathcal{C}: |\mathcal{C}| = k} \mathrm{cost}(\mathcal{X}; \mathcal{C})$ to be the minimum $k$-means (or $k$-median) cost.

In $k$-means or $k$-median clustering, given a dataset $\mathcal{X} \subset B(0, \Lambda)$ of size $n$, our goal is to efficiently find a set of points $\mathcal{C}$ such that $\mathrm{cost}(\mathcal{X}; \mathcal{C})$ is a good approximation to $\mathrm{OPT}(\mathcal{X})$. In general, we wish for purely multiplicative approximations, but due to the nature of private $k$-means (and $k$-median), we will additionally have a small additive approximation that is proportional to $\Lambda^p$. We now define approximate $k$-means/$k$-median solutions.

**Definition A.6.** For any $\alpha \geq 1$, a set $\mathcal{C}$ of size $k$ is said to be an $\alpha$-approximate for $\mathcal{X}$ *with additive error $V$* if $\mathrm{cost}(\mathcal{X}; \mathcal{C}) \leq \alpha \cdot \mathrm{OPT}(\mathcal{X}) + V \cdot \Lambda^p$.

We also define bicriteria solutions for $k$-means and $k$-median: here, we are allowed to use a larger dataset $\mathcal{C}$ that may have more than $k$ points, but still compare to the optimal $k$-clustering.

**Definition A.7.** For any $\alpha, \beta \geq 1$, a set $\mathcal{C}$ is said to be an $(\alpha, \beta)$-bicriteria approximate solution for $\mathcal{X}$ with additive error $V$ if $|\mathcal{C}| \leq \beta \cdot k$ and $\mathrm{cost}(\mathcal{X}; \mathcal{C}) \leq \alpha \cdot \mathrm{OPT}(\mathcal{X}) + V \cdot \Lambda^p$.

Finally, we also note an important result on dimensionality reduction for clustering, due to [56].

**Theorem A.8.** *Let $\mathcal{X} = \{x_1, \ldots, x_n\}$ be a set of points in $\mathbb{R}^d$, and fix some integer $k \leq n$ and a real number $\gamma \leq \frac{1}{2}$. Let $d' = C\gamma^{-2} \log(k/\gamma)$ for a sufficiently large constant $C$. Then, for $\Pi \in \mathbb{R}^{d' \times d}$ chosen where each entry of $\Pi$ is i.i.d. $\frac{1}{\sqrt{d'}} \cdot \mathcal{N}(0, 1)$ (we will call this matrix $\Pi$ a "random projection" down to $d'$ dimensions), the optimal $k$-means costs $\mathrm{OPT}(\mathcal{X})$ and $\mathrm{OPT}(\Pi\mathcal{X})$ are equal up to a $1 \pm \gamma$ multiplicative cost. Likewise, the optimal $k$-median costs $\mathrm{OPT}(\mathcal{X})$ and $\mathrm{OPT}(\Pi\mathcal{X})$ are also equal up to a $1 \pm \gamma$ multiplicative cost.*

*In addition, the following stronger claim holds simultaneously for every partition of $\mathcal{X}$ into $\mathcal{X}_1, \ldots, \mathcal{X}_k$. We have that the minimum $k$-means (resp., $k$-median) cost where each $\mathcal{X}_i$ is a single cluster, and the minimum $k$-means (resp., $k$-median) cost where each $\Pi\mathcal{X}_i$ is a single cluster, are equal up to a $1 \pm \gamma$ multiplicative factor.*

We note that the minimum $k$-means (resp., $k$-median) cost when the clusters $\mathcal{X}_1, \ldots, \mathcal{X}_k$ are fixed is determined by setting each center $c_i$ to be the mean (resp., geometric median) of the points in $\mathcal{X}_i$.

## A.3 Massively Parallel Computation (MPC)

We recall the description of the MPC model from Subsection 2.2. In our setting of clustering, we will assume that there are $O(n^{1-\theta})$ machines, each of which can store $\tilde{O}(n^\theta)$ points of dimension $d$. So, the total memory is $\tilde{O}(nd)$.

We remark that we will frequently use the phrase *near-linear time*: this will represent an MPC algorithm that uses at most $\tilde{O}(nd)$ total time across all machines, and at most $\tilde{O}(n^\theta d)$ time in any individual machine.

We recall the primitive of sorting, and copy Theorem 2.2 here.

**Theorem A.9.** *(Sorting) [46, 45] There is an MPC algorithm which sorts $N$ data items in $O(1)$ rounds using $O(N)$ total space. The local memory per machine required is at most $O(N^\theta)$ for an arbitrary small constant $\theta > 0$.*

We next will need the following primitive that we call *aggregation*, which follows from the primitive "Sizes of sets" in [3].

**Lemma A.10.** *(Aggregation) Suppose we have $N$ items, each of which has a label $u$ in a universe $\mathcal{U}$ (which may be larger than $N$). There is an MPC algorithm which stores all tuples $(u, n_u)$, where $u \in U$ has at least one item labeled with $u$ and $n_u$ counts the number of such items, using $O(1)$ rounds and $O(N)$ total space. The local memory per machine required is at most $O(N^\theta)$ for an arbitrary small constant $\theta > 0$.*

In addition, we will need the following primitive that we call *sampling*, which follows from the primitive "Indexing elements in sets" in [3].

**Lemma A.11.** *(Sampling) Suppose we have $N$ items which are partitioned into $k$ sets $S_1, \ldots, S_k$. Each item comes with an index in $[k]$ representing its set. Then, for any positive integer $T$, we can simultaneously choose a random sample of size $T$ from each $S_i$ (and if $|S_i| \leq T$, we simply choose all elements of $S_i$). This can be done in MPC with $O(1)$ rounds, $\tilde{O}(N)$ total space, and local memory at most $O(N^\theta)$ for an arbitrary small constant $\theta > 0$.*

*Proof.* For each element $x$, we assign it a random key $\ell$ between 1 and $N^{O(1)}$. With high probability there is no collision of keys. For each element $x$, say it is assigned the pair $(j, \ell)$, where $x \in S_j$ and $\ell$ is the key. We sort the elements based on $j$ and tiebreaking with $\ell$, using Theorem A.9.

Now, the "indexing elements in sets" procedure of [3] allows us to determine the relative position of each $x$ in its set $S_j$, for all $j \in [k]$ simultaneously. In other words, it can determine how many elements $x'$ also in $S_j$ have the same or lower key than $x$, since we have sorted all points in $S_j$ in order of key. Thus, if we store all elements $x$ such that this value is at most $T$, we are exactly storing the elements with the $T$ lowest keys among elements in $S_j$, simultaneously for all $j \in [k]$. Since the keys were random, this is exactly the sampling procedure we want. □

Finally, we require an algorithm for solving $k$-means or $k$-median in MPC, which follows from the MPC coreset algorithm of Theorem 2.3.

**Theorem A.12.** *(MPC $k$-means/$k$-median) [17, 40, 41] Consider $n$ points in $\mathbb{R}^d$. Suppose there exists a polynomial-time (sequential, non-private) $\rho$-approximation to $k$-means (resp., $k$-median). Then, there exists a non-private MPC algorithm which outputs a $\rho(1 + \gamma)$ $k$-means (resp.,$k$-median) solution, for any arbitrarily small constant $\gamma > 0$, in $O(1)$ rounds. The total space needed is $O(nd) + \text{poly}(k, d, \log n, 1/\gamma)$, and the local memory per machine required is $\text{poly}(k, d, \log n, 1/\gamma)$. In addition, the total running time over all machines is at most $O(nd) + \text{poly}(k, d, \log n, 1/\gamma)$.*

## A.4 Approximate Near Neighbors and Randomly Shifted Grids

In our algorithms, we will also make use of Approximate Nearest Neighbor (ANN) data structures, as well as the Quadtree data structure which is composed of randomly shifted grids.

**Definition A.13.** *[48] The $K$-approximate nearest neighbor ($K$-ANN) problem with failure probability $\tau$ involves constructing a data structure over a set of points $\mathcal{C}$ in $\mathbb{R}^d$ supporting the following query: given any fixed query point $q \in \mathbb{R}^d$, return a point $c \in \mathcal{C}$ such that with probability at least $1 - \tau$, $d(q, c) \leq K \cdot \min_{c' \in \mathcal{C}} d(q, c')$.*

This has been a very well-studied problem, with the following algorithm as the best-known ANN data structure in high-dimensional Euclidean space.

**Theorem A.14.** *[2, 48] Given a dataset $\mathcal{C}$ of size $m \leq n$, there exists a data structure that can solve $K$-ANN with failure probability $\tau = \frac{1}{\text{poly}(n)}$, with total space $O\left((dm + m^{1+1/K^2 + o(1)}) \cdot (\log n)\right)$ and query time $O\left(d \cdot m^{1/K^2 + o(1)} \cdot (\log n)\right)$. In addition, if $K = O(\log n)$, we can improve the total space to $O(dm \cdot \text{poly} \log n)$ and the query time to $O(d \cdot \text{poly} \log n)$.*

Next, we describe the Quadtree data structure. This data structure has also been useful for approximate nearest neighbor algorithms, though we will analyze this data structure more directly.

**Definition A.15.** *A randomly shifted Quadtree* is constructed as follows. We start with a top level of some size $\Lambda$ and let level 0 be a single grid cell, which is the $d$-dimensional hypercube $[-\Lambda, \Lambda]^d$. Next, we choose a uniformly random point $\nu = (\nu_1, \ldots, \nu_d) \in [-\Lambda, \Lambda]^d$, which will represent our shift vector. Now, for each level $\ell \geq 1$, we partition the region $[-\Lambda, \Lambda]^d$ into grid cells of size $\Lambda/2^\ell$, shifted by $\nu$. In other words, each cell is the form $[\nu_1 + a_1 \cdot \Lambda/2^\ell, \nu_1 + (a_1 + 1) \cdot \Lambda/2^\ell] \times \cdots \times [\nu_d + a_d \cdot \Lambda/2^\ell, \nu_d + (a_d + 1) \cdot \Lambda/2^\ell]$, where $a_1, \ldots, a_d \in \mathbb{Z}$. We say that $\Lambda/2^\ell$ is the *grid size* at level $\ell$. (We remark that we may truncate some grid cells so that they do not escape $[-\Lambda, \Lambda]^d$.) We continue this for some number of levels, until we reach some bottom level.

We will only need the following straightforward fact about Quadtrees.

**Proposition A.16.** *Let $B$ be an $\ell_\infty$ ball of radius $r$ contained in $[-\Lambda, \Lambda]^r$ (so it forms a $d$-dimensional cube with each side length $2r$). Then, for a randomly shifted quadtree and any level $\ell$ with grid size at least $r' \geq 2r$, $B$ is split by the grid in each dimension $j \in [d]$ independently with probability $1 - \frac{2r}{r'}$.*

## A.5 List-Decodable Covers

In this subsection, we note an important result on list-decodable covers, which we will use in a similar manner as [44] to obtain a differentially private coreset of the data for either $k$-means or $k$-median. First, we need the following definition, which is restated from [44].

**Definition A.17.** *Given a ball $B$ centered around some point $\mu \in \mathbb{R}^d$ and radius $R$, we say that a set of points $\mathcal{H}$ is a $\gamma$-cover of $B$ if for every point $x \in B$, there exists $h \in \mathcal{X}$ such that $d(x, h) \leq \gamma \cdot R$.*

In addition, we say that a $\gamma$-cover of $B$ is *list-decodable* with list size $\ell$ at distance $\gamma' \geq \gamma$ if, for any $x \in B$, the number of points $h \in \mathcal{H}$ with $d(x, h) \leq \gamma' \cdot R$ is at most $\ell$. Furthermore, we say that the $\gamma$-cover is *efficiently list-decodable* at distance $\gamma'$ if there exists an efficient algorithm that, for any $x \in B$, recovers all such points $h \in \mathcal{H}$ with $d(x, h) \leq \gamma \cdot R$ in time $\text{poly}(\ell, d, \log \gamma^{-1})$.

We will need the following lemma from [44], which is based on previous work of [66, 57].

**Lemma A.18.** *For every $0 < \gamma < 1$, and any ball $B \subset \mathbb{R}^d$ there exists a $\gamma$-cover $\mathcal{H}$ that is efficiently list-decodable at distance $\gamma'$ with list size $\ell = (1 + \frac{\gamma'}{\gamma})^{O(d)}$. In addition, the cover is formed by a lattice, for which a basis can be constructed in time $e^{O(d)}$.*

As a corollary, we have the following, by setting $\gamma' = 1$ and applying the efficient list-decodable property with radius $\gamma'$ around the center $\mu$ of $B$.

**Corollary A.19.** *For any ball $B \subset \mathbb{R}^d$ with center $\mu$ and radius $R$, and for any $\gamma \leq 1$, there exists a $\gamma$-cover $\mathcal{H}$ of $B$ of size at most $O(1/\gamma)^{O(d)}$. In addition, all the points of the cover can be found in time $O(1/\gamma)^{O(d)}$.*

## A.6 Sampling Lemmas

In this subsection, we note some useful lemmas relating to subsampling a dataset. First, we prove a sampling lemma that shows that via uniform sampling, one can approximate the $k$-means cost, as well as the $k$-median cost, of a set of points, up to some combined additive and multiplicative error. This lemma is essentially proven in [21, 49], but due to some minor changes in the statement we want, we prove it for completeness.

**Lemma A.20.** *Let $0 < \kappa, \gamma < 1$, and let $n \geq N \geq T$ be fixed integers such that $T \geq \Theta\left(\frac{k \cdot d \cdot \log((\kappa\gamma)^{-1}) + \log n}{\kappa^2 \cdot \gamma^2}\right)$. Let $\mathcal{X}$ be a dataset of size $N$ entirely contained in a unit ball of radius $1$. Consider selecting a random set $\mathcal{Y}$ of size $T$ in $\mathcal{X}$, selected without replacement. Then, with probability at least $1 - 1/\text{poly}(n)$ over $\mathcal{Y}$, for any set $\mathcal{C} = \{c_1, \ldots, c_{k'}\} \in \mathbb{R}^d$ of size at most $k$, and for either $p = 1$ or $p = 2$,*

$$\sum_{y \in \mathcal{Y}} d(\mathcal{C}, y)^p = (1 \pm \kappa) \cdot \frac{T}{N} \cdot \sum_{x \in \mathcal{X}} d(\mathcal{C}, x)^p \pm \gamma \cdot T. \tag{1}$$

*Proof.* First, we may assume WLOG that the center of the ball is the origin, by shifting. In addition, we may assume that all of the distances from $c_i$ to the origin are within $2$ of each other, i.e., $\max_i \|c_i\|_2 - \min_i \|c_i\|_2 \leq 2$. Otherwise, $d(\mathcal{C}, y)$ is the same as $d(\mathcal{C} \backslash \{\arg\max_{c_i \in \mathcal{C}} \|c_i\|_2\}, y)$ for any point $y$ in the unit ball. Next, we may assume that $\min_i \|c_i\|_2 \leq O(1/\kappa)$ (which means $\max_i \|c_i\|_2 \leq O(1/\kappa)$). This is because otherwise, $d(\mathcal{C}, y)^2 = (1 \pm \kappa) \cdot d(\mathcal{C}, x)^2$ (and similarly $d(\mathcal{C}, y) = (1 \pm \kappa) \cdot d(\mathcal{C}, x)$) for any points $x, y$ in the unit ball. So, we just have to focus on $\mathcal{C}$ only containing points within $O(1/\kappa)$ of the origin.

Now, we fix a subset $\mathcal{C}$ of size at most $k$ beforehand, and enumerate the points $\mathcal{X} = \{x_1, \ldots, x_N\}$. For each $i \leq N$, define $a_i = d(\mathcal{C}, x_i) \leq O(1/\kappa)$. Let $S \subset [N]$ be a random subset of $T$ elements chosen without replaccement. By using a version of Hoeffding's inequality without replacement [50], we have that $\mathbb{P}\left(\left|\sum_{i \in S} a_i - \frac{T}{N} \cdot \sum_{i \in [N]} a_i\right| \leq t\right) \leq 2 \cdot e^{-\Omega(t^2)/T}$, because the $a_i$'s are all contained in an interval of radius $4$. This is because, as in the previous paragraph, we noted that all of the values $\|c_i\|$

are within 2 of each other, and all points $x_i$ are in a ball of radius 1, so $d(c_i, x) = \|c_i\|_2 \pm 1$ for any $x$ in the unit ball. Likewise, $\mathbb{P}\left(\left|\sum_{i \in S} a_i^2 - \frac{T}{N} \cdot \sum_{i \in [N]} a_i^2\right| \le t\right) \le 2 \cdot e^{-\Omega(\kappa^2 \cdot t^2)/T}$. This is true since the $a_i$'s are in an interval of radius 4 and are all at most $O(1/\kappa)$, which means the $a_i^2$'s are concentrated in an interval of radius $O(1/\kappa)$. So, if we set $t = \frac{\gamma \cdot T}{2}$, then we have that Equations (1) holds with failure probability at most $2e^{-\Omega(\kappa^2 \cdot \gamma^2 \cdot T)} \le 2e^{-\Omega(kd \log((\kappa\gamma)^{-1}) + \log n)} \le \frac{1}{\text{poly}(n)} \cdot (\kappa \cdot \gamma)^{\Omega(kd)}$, for either $p = 1$ or $p = 2$.

Now, we must perform a union bound over all subsets $\mathcal{C}$. Although there are infinitely many of them, we may assume WLOG that the subset $\mathcal{C}$ is contained in some $(\frac{\gamma \cdot \kappa}{100})^{O(1)}$-cover of the $O(1/\kappa)$-sized ball, which has size $(1/(\kappa \cdot \gamma))^{O(d)} = (\kappa \cdot \gamma)^{-O(d)}$. Otherwise, we may move each point in $\mathcal{C}$ to the closest point in the net, which does not affect each $a_i = d(x_i, \mathcal{C})$ or $a_i^2$ by more than $\gamma/10$ additively. So, the number of possible subsets $\mathcal{C}$ of size at most $k$ is at most $(\kappa \cdot \gamma)^{-O(kd)}$, which means the overall failure probability is $\frac{1}{\text{poly}(n)}$. $\qquad\square$

Finally, we note that any private algorithm applied on a sampled dataset is still private. Specifically, we have the following result.

**Lemma A.21.** *Let $\varepsilon, \delta \le \frac{1}{2}$, and let $\mathcal{A}$ be an $(\varepsilon, \delta)$-DP algorithm on a dataset $\mathcal{Y}$. Now, let $\mathcal{B}$ be an algorithm on a dataset $\mathcal{X}$, where we first either sample every point in $\mathcal{X}$ independently with some fixed probability or sample $T$ random points in $\mathcal{X}$ without replacement for some fixed $T$, and then apply $\mathcal{A}$ to the sampled data points. Then, $\mathcal{B}$ is also $(\varepsilon, \delta)$-DP.*

In fact, one can actually get slightly stronger bounds on the privacy of $\mathcal{B}$ if the sampling probability of each element is small (see, e.g., [7, 14]), but this will not end up being particularly useful in improving our theoretical guarantees significantly.

# B  A Private MPC Bicriteria Approximation

In this section, we create a bicriteria approximation algorithm for $k$-means and $k$-median clustering, which runs in near-linear time and only uses $O(1)$ rounds of communication. In other words, this algorithm will have a multiplicative approximation but will also use more than $k$ centers. We will use this procedure as a starting point which will eventually give us an $O(1)$-approximation algorithm using only $k$ centers. Our technique for this section is somewhat inspired by [24], except we use a greedy approach rather than their dynamic programming approach to speed up the runtime and reduce the number of rounds of communication.

**Theorem B.1.** *There exists an $(\varepsilon, \delta)$-DP algorithm that, given a dataset $\mathcal{X} \subset B(0, \Lambda)$ of size $n$, computes an $(O(d^3), \log^2 n)$-bicriteria approximation to the optimal $k$-means (or $k$-median) cost, with additive error $k \cdot \text{poly}(\log n, d, \varepsilon^{-1}, \log \delta^{-1})$. In addition, the algorithm only requires $O(1)$ rounds of communication in MPC in near-linear time.*

*Remark* B.2. We are not particular about polynomial dependencies on $d$, as we will later show how to reduce $d$ to roughly $O(\log k)$.

**Algorithm:**   We create a randomly shifted quadtree with bottom level having grid size $\Theta(\Lambda/n^2)$ and the top (0th) level being $[-\Lambda, \Lambda]^d$. Note that the quadtree has $\Theta(\log n)$ levels. Now, at each level $\ell$ of the quadtree, we count the number of points in each cell $\mathbf{c}$ and add Truncated Laplace noise $\text{TLap}(1, \varepsilon/\log^2 n, \delta)$ noise. In other words, for every $\ell \le O(\log n)$ and every cell $\mathbf{c}$ in level $\ell$, we compute a quantity $\tilde{N}_{\ell, \mathbf{c}}$ which equals $|\mathcal{X} \cap \mathbf{c}| + \text{TLap}(1, \varepsilon/\log^2 n, \delta/\log^2 n)$. We then, for each level $\ell$, pick the (up to) $4k$ cells with largest $\tilde{N}_{\ell, \mathbf{c}}$, assuming the counts are at least $2\varepsilon^{-1} \log^2 n \cdot \log \frac{\log^2 n}{\delta}$.

We will run the above algorithm $O(\log n)$ times in parallel (with independent randomness in each quadtree) and let $\mathcal{F}$ be the union of the centers of all cells picked, across all parallel iterations and all levels. So, $\mathcal{F}$ will have $O(k \log^2 n)$ cells in total.

**Privacy:**   We claim that a single parallel iteration of the algorithm is $(O(\varepsilon/\log^2 n), O(\delta/\log^2 n))$-DP at each level. Indeed, if a single point in $\mathcal{X}$ is added, deleted, or moved, at most 2 cells change in size, each by at most 1. So, outputting the counts $\tilde{N}_{\ell, \mathbf{c}}$ is at most $(2\varepsilon/\log^2 n, 2\delta/\log^2 n)$-DP because

of our Truncated Laplace noise. In addition, choosing the (up to $4k$) largest cells only depends on the noisy counts $\tilde{N}_{\ell,\mathbf{c}}$ so it does not affect privacy. So, doing this over all levels and applying basic composition tells us this algorithm is $(O(\varepsilon/\log n), O(\delta/\log n))$-DP. Because we run $O(\log n)$ parallel copies of this procedure, so we get $(O(\varepsilon), O(\delta))$-DP.

**Runtime:** We focus on a single iteration and single level of the algorithm. First, we assign each point $x$ a pointer to its grid cell at that level. Then, using $O(1)$ rounds, we can *sort* the points based on the location of the grid cell, using Theorem A.9. Now, in $O(1)$ rounds we can *aggregate* (without privacy) to count the total number of points in each (nonempty) grid cell, using Lemma A.10. We then add Truncated Laplace noise to each grid cell, and then sort again to find the heaviest grid cells after imposing privacy constraints. This can all be done using a nearly linear amount of time per machine. In addition, there is no need to add noise to any empty cell, because the error induced by the Truncated Laplace noise is at most $\varepsilon^{-1}\log^2 n \cdot \log\frac{\log^2 n}{\delta}$, which is below the threshold of $2\varepsilon^{-1}\log^2 n \cdot \log\frac{\log^2 n}{\delta}$ for outputting the count. There are at most $O(\log n)$ levels per quadtree and at most $O(\log n)$ quadtrees, so we can create $O(\log^2 n)$ copies of the dataset $\mathcal{X}$ and generate the centers for each level-iteration pair. Overall, we will have a total of $O(k\log^2 n)$ points, which can all be sent to a single machine for storage to generate our set $\mathcal{F}$.

In total, the algorithm runs in near-linear time, with $O(1)$ rounds of communication.

**Accuracy:** Let $\mathcal{X} = \{x_1, \ldots, x_n\}$ be our original set of points, and let $\mathcal{C} = \{c_1, \ldots, c_k\}$ be the optimal set of $k$ centers. For any radius $r$, let $n_r$ be the number of points $x \in \mathcal{X}$ such that $d(x, \mathcal{C}) \geq r$. Then, note that the $k$-means cost and $k$-medians cost, up to an $O(1)$-approximation, equal

$$\sum_{t\in\mathbb{Z}} 2^{2t} \cdot n_{2^t} \qquad \text{and} \qquad \sum_{t\in\mathbb{Z}} 2^t \cdot n_{2^t},$$

respectively.

Now, let's fix some radius $r = 2^t$, and consider a randomly shifted grid of size $20r \cdot d$. Note that for any point $c \in \mathcal{C}$, the Euclidean ball $B(c, r)$ of radius $r$ around $c$ is contained in the $\ell_\infty$ ball $B_\infty(c, r)$. Thus, by Proposition A.16, the probability this ball is split in any fixed dimension by the grid of size $20r \cdot d$ is at least $1 - 1/(10d)$. So, the probability that it is split in a total of $q$ dimensions is at most $1/(10d)^q \cdot \binom{d}{q} \leq 10^{-q}$. So, each ball $B(c_i, r)$ in expectation is split into at most $\sum_{q\geq 0} 2^q \cdot 10^{-q} \leq 2$ total cells by the corresponding level of the quadtree. Hence, $\bigcup_{i=1}^k B(c_i, r)$ is contained in at most $4k$ cells of grid size $20rd$ with at least $50\%$ probability, by a Markov bound.

Now, suppose that all points in $\bigcup_{i=1}^k B(c_i, r)$ are contained in at most $4k$ total cells at this level. Since each cell has side length $20r \cdot d$, it has $\ell_2$-radius $10r \cdot d^{3/2}$. Therefore, with at least $50\%$ probability, the number of points that are not within $10r \cdot d^{3/2}$ of the $4k$ heaviest cells' centers at this level is at most $n_r$. However, the noisy counts affect which cells the algorithm outputs to be heaviest. But each count is accurate up to error $3\varepsilon^{-1}\log^2 n \cdot \log\frac{\log^2 n}{\delta}$ even if we replace any $\tilde{N}_{\ell,\mathbf{c}}$ below $2\varepsilon^{-1}\log^2 n \cdot \log\frac{\log^2 n}{\delta}$ with 0. So, overall, with at least $50\%$ probability, we have found at most $4k$ grid cell centers such that the number of points not within $10d^{3/2} \cdot r$ of any of these points is at most $n_r + O\left(k \cdot \varepsilon^{-1} \cdot \log^2 n \cdot \log\frac{\log^2 n}{\delta}\right)$.

By repeating this across $\log n$ parallel copies, we can boost this failure probability sufficiently high. We then repeat this across all levels, to get that for all choices of $r$ between $\Lambda/n^2$ and $\Lambda$, the number of points of distance at least $O(d^{3/2} \cdot r)$ from some point in our final set $\mathcal{F}$ is at most $n_r + O\left(k \cdot \varepsilon^{-1} \cdot \log^2 n \cdot \log\frac{\log^2 n}{\delta}\right)$. We also do not need to check $r$ beyond $\Lambda$, as the center of the

top level of the quadtree contains all points. So, the $k$-means cost is at most

$$\sum_{t \in \mathbb{Z}: r = 2^t \leq \Lambda} O(d^{3/2} \cdot r)^2 \cdot \left( n_r + O\left( k \cdot \varepsilon^{-1} \cdot \log^2 n \cdot \log \frac{\log^2 n}{\delta} \right) \right)$$

$$\leq O(d^3) \cdot \left[ \sum_{t \in \mathbb{Z}} 2^{2t} \cdot n_{2^t} + O\left( k \cdot \varepsilon^{-1} \cdot \log^2 n \cdot \log \frac{\log^2 n}{\delta} \right) \cdot \sum_{t \in \mathbb{Z}: r = 2^t \leq \Lambda} 2^{2t} \right]$$

$$\leq O(d^3) \cdot \mathrm{OPT} + O\left( k d^3 \log^3 n \cdot \varepsilon^{-1} \log \delta^{-1} \right) \cdot \Lambda^2.$$

Hence, for $k$-means we obtain a $\mathrm{poly}(d)$-multiplicative and a $O(k \cdot \mathrm{poly}(d, \log n, \varepsilon^{-1}, \log \delta^{-1})) \cdot \Lambda^2$-additive error, using $O(k \log^2 n)$ total centers. A very similar calculation gives us the same result (with only a $O(d^{3/2})$-multiplicative factor) for $k$-median.

To finish, we provide algorithm pseudocode, as Algorithm 3.

---

**Algorithm 3** A bicriteria approximation algorithm for differentially private $k$-means (or $k$-median).

1: **procedure** BICRITERIA($\mathcal{X}, \varepsilon, \delta$)                      ▷ Will be $(O(\varepsilon), O(\delta))$-DP.
2:     $\mathcal{F} \leftarrow \emptyset$                      ▷ Initialize set of centers to $\emptyset$, we will add centers to it.
3:     **for** rep $= 1$ to $O(\log n)$ **do**
4:         Create a randomly shifted quadtree with top level $[-\Lambda, \Lambda]^d$ and bottom level with grid size $\Lambda/n^2$.
5:         **for** each $0 \leq \ell \leq \log_2(n^2)$ **do**
6:             **for** each cell **c** in level $\ell$ **do**
7:                 Let $N_{\ell,\mathbf{c}}$ be the number of points in $\mathcal{X} \cap \mathbf{c}$.
8:                 Let $\tilde{N}_{\ell,\mathbf{c}} = N_{\ell,\mathbf{c}} + \mathrm{TLap}(1, \varepsilon/\log^2 n, \delta/\log^2 n)$ (only compute for $N_{\ell,\mathbf{c}} > 0$).
9:             Add the centers of the $4k$ cells **c** with largest $\tilde{N}_{\ell,\mathbf{c}}$ to $\mathcal{F}$.
10:     Return $\mathcal{F}$.

---

## C   Obtaining a Constant Approximation in Near-Linear Time

In this section, we show a black-box method for creating an efficient parallel algorithm with $O(1)$-approximation factor, given a polynomial-time sequential algorithm with $O(1)$-approximation factor (for which we will use [70]) and an efficient parallel bicriteria algorithm with weaker approximation guarantees, as in Section B.

We begin by showing how one can obtain a "semi-coreset" given an $O(1)$-approximate algorithm for $k$-means and $k$-median. Namely, our semi-coreset will also have an additive error proportional to the optimal cost, which prevents it from being a standard coreset. However, this will end up being sufficient for our purposes. The following lemma is inspired by a result proven in Kaplan and Stemmer [70], but differs in that we extend their procedure to work for approximate nearest neighbor algorithms. While the proof of the following lemma is written for $k$-means, a nearly identical proof holds for $k$-median.

**Lemma C.1.** *Let $\mathcal{A}$ be an $(\varepsilon, \delta)$-DP (sequential) algorithm that takes as input a dataset $\mathcal{Y}$ of size at least $\Omega(k \cdot \varepsilon^{-1} \log n)$ contained in a ball $B$ of radius $R$ (with known center) and outputs outputs a set of $k$ centers $\mathcal{G}$ contained in $B$ such that $\mathrm{cost}(\mathcal{X}; \mathcal{G}) \leq O(1) \cdot \mathrm{OPT}(\mathcal{X}) + V(n, d, k, \varepsilon, \delta) \cdot R^p$.*

*Then, for any fixed $\eta > 0$, there exists a $(3\varepsilon, \delta)$-DP (sequential) algorithm $\mathcal{B}$ on the dataset $\mathcal{Y}$ that outputs a set $\hat{\mathcal{Y}} \subset B$ such that for any set $\mathcal{C}$ of size at most $k$.*

$$\mathrm{cost}(\hat{\mathcal{Y}}; \mathcal{C}) \leq O(1) \cdot \left[ \mathrm{OPT}(\mathcal{Y}) + \mathrm{cost}(\mathcal{Y}; \mathcal{C}) + \left( V(n, d, k, \varepsilon, \delta) + k \cdot \varepsilon^{-1} \log n \right) \cdot R^p \right] \quad (2)$$

$$\mathrm{cost}(\mathcal{Y}; \mathcal{C}) \leq O(1) \cdot \left[ \mathrm{OPT}(\mathcal{Y}) + \mathrm{cost}(\hat{\mathcal{Y}}; \mathcal{C}) + \left( V(n, d, k, \varepsilon, \delta) + k \cdot \varepsilon^{-1} \log n \right) \cdot R^p \right]. \quad (3)$$

*In addition, constructing the set requires only $O(k^{1+\eta} \cdot d \cdot \log n)$ space and at most $O(k^\eta \cdot d \cdot \log n)$ time per point in $\mathcal{Y}$, for some arbitrarily chosen constant $\eta > 0$.*

*Proof.* For simplicity of the presentation, we present the proof for $k$-means. The proof for $k$-median is almost identical. We begin by constructing a $K$-approximate nearest neighbor (ANN) data structure

for the dataset $\mathcal{G}$, which uses $O(k^{1+\eta} \cdot d \log n)$ space and $O(k^\eta \cdot d \log n)$ query time per point. By Theorem A.14, we can set $K = \Theta(\eta^{-1/2})$, which for $\eta > 0$ fixed, is a constant.

Let the dataset $\tilde{\mathcal{Y}}$ be generated by replacing point in $\mathcal{Y}$ with a $K$-approximate nearest neighbor in $\mathcal{G}$ (counting multiplicity). For each point $y \in \mathcal{Y}$, let $\tilde{y}$ represent its corresponding point in $\tilde{\mathcal{Y}}$. Then, for any set of at most $k$ centers $\mathcal{C}$ (even if $\mathcal{C}$ is not contained in $B$),

$$
\begin{aligned}
\mathrm{cost}(\tilde{\mathcal{Y}}; \mathcal{C}) &= \sum_{\tilde{y} \in \tilde{\mathcal{Y}}} d(\tilde{y}, \mathcal{C})^2 \\
&\leq \sum_{y \in \mathcal{Y}} (d(\tilde{y}, y) + d(y, \mathcal{C}))^2 \tag{4} \\
&\leq \sum_{y \in \mathcal{Y}} 2 \left[ (d(\tilde{y}, y))^2 + d(y, \mathcal{C})^2 \right] \\
&\leq 2 \left[ \sum_{y \in \mathcal{Y}} K^2 \cdot d(y, \mathcal{G})^2 + \sum_{\tilde{y} \in \tilde{\mathcal{Y}}} d(y, \mathcal{C})^2 \right] \\
&\leq O(K^2) \cdot \mathrm{OPT}(\mathcal{Y}) + 2K^2 V(n, d, k, \varepsilon, \delta) \cdot R^2 + 2 \cdot \mathrm{cost}(\mathcal{Y}; \mathcal{C}). \tag{5}
\end{aligned}
$$

Conversely, we have that

$$
\begin{aligned}
\mathrm{cost}(\mathcal{Y}; \mathcal{C}) &= \sum_{y \in \mathcal{Y}} d(y, \mathcal{C})^2 \\
&\leq \sum_{y \in \mathcal{Y}} (d(y, \tilde{y}) + d(\tilde{y}, \mathcal{C}))^2 \\
&\leq 2 \left[ \sum_{y \in \mathcal{Y}} K^2 \cdot d(y, \mathcal{G})^2 + \sum_{\tilde{y} \in \tilde{\mathcal{Y}}} d(\tilde{y}, \mathcal{C})^2 \right] \\
&\leq O(K^2) \cdot \mathrm{OPT}(\mathcal{Y}) + 2K^2 V(n, d, k, \varepsilon, \delta) \cdot R^2 + 2 \cdot \mathrm{cost}(\tilde{\mathcal{Y}}; \mathcal{C}). \tag{6}
\end{aligned}
$$

Note that $\tilde{\mathcal{Y}}$ is contained in $\mathcal{G}$, but each point in $\tilde{\mathcal{Y}}$ may have large multiplicity. Let $\hat{\mathcal{Y}}$ be the set where we add $\mathrm{Lap}(1/\varepsilon)$ noise to the multiplicity of each point in $\mathcal{G}$. In other words, if $g \in \mathcal{G}$ has multiplicity $m_g$ in $\tilde{\mathcal{Y}}$, it will have multiplicity $\max(0, m_g + \mathrm{Lap}(1/\varepsilon))$ in $\hat{\mathcal{Y}}$.

First, note that since $\mathcal{G}$ is $(\varepsilon, \delta)$-DP with respect to $\mathcal{Y}$, the ANN data structure also satisfies $(\varepsilon, \delta)$-DP with respect to $\mathcal{Y}$, as it only depends on $\mathcal{G}$. This implies that $\hat{\mathcal{Y}}$ is $(3\varepsilon, \delta)$-DP. To see why, if we fix the ANN data structure, adding, removing, or changing a single point from $\mathcal{Y}$ changes the multiplicity $m_g$ of at most two points in $\mathcal{G}$ by at most 1, which makes the sensitivity of the multiplicities at most 2. So, the standard Laplace mechanism to generate the multiplicities for $\hat{\mathcal{Y}}$ is $(2\varepsilon, 0)$-DP. Hence, the adaptive composition of the ANN data structure with the construction of $\hat{\mathcal{Y}}$ is $(3\varepsilon, \delta)$-DP.

Next, note that with overwhelming probability, every $\mathrm{Lap}(1/\varepsilon)$ noise added is at most $O(\varepsilon^{-1} \log n)$. If $\mathcal{C}$ has nonempty intersection with the ball $B_2$ that is concentric with $B$ with radius $2R$, we have that $d(\tilde{y}, \mathcal{C}) \leq O(R)$ as $\tilde{y} \in \mathcal{G} \subset B$. So, the maximum additional error we gain is at most $O(k \cdot \varepsilon^{-1} \cdot \log n \cdot R^2)$. Alternatively, if all points in $\mathcal{C}$ are not in $B_2$, then $d(y, \mathcal{C})$ for every point $y \in B$ is equivalent up to a factor of 3. In this case, as we assume the number of points in $\tilde{\mathcal{Y}}$ (counting multiplicity) is at least $\Theta(k \cdot \varepsilon^{-1} \cdot \log n) = \Theta(|\mathcal{G}| \cdot \varepsilon^{-1} \cdot \log n)$, we have that the number of points in $\tilde{\mathcal{Y}}$ and the number of points in $\hat{\mathcal{Y}}$ (counting multiplicity) are equal up to a constant factor, as the multiplicity of each point in $\mathcal{G}$ does not change by more than $O(\varepsilon^{-1} \cdot \log n)$. Thus, in this case, $\mathrm{cost}(\tilde{\mathcal{Y}}; \mathcal{C})$ and $\mathrm{cost}(\hat{\mathcal{Y}}; \mathcal{C})$ are equal up to a constant factor.

By combining this observation with Equations (5) and (6), if we treat $K$ as a constant, we have that for any set $\mathcal{C}$ of size at most $k$,

$$
\begin{aligned}
\mathrm{cost}(\hat{\mathcal{Y}}; \mathcal{C}) &\leq O(1) \cdot \left[ \mathrm{OPT}(\mathcal{Y}) + \mathrm{cost}(\mathcal{Y}; \mathcal{C}) + \left( V(n, d, k, \varepsilon, \delta) + k \cdot \varepsilon^{-1} \log n \right) \cdot R^2 \right] \\
\mathrm{cost}(\mathcal{Y}; \mathcal{C}) &\leq O(1) \cdot \left[ \mathrm{OPT}(\mathcal{Y}) + \mathrm{cost}(\hat{\mathcal{Y}}; \mathcal{C}) + \left( V(n, d, k, \varepsilon, \delta) + k \cdot \varepsilon^{-1} \log n \right) \cdot R^2 \right]. \quad \square
\end{aligned}
$$

Now, consider some set $\mathcal{X}_0 \subset B(\mu, R)$ of size $N_0$, for some center $\mu$ and some radius $R$. Let $\mathcal{Y}_0$ be a random subset of $T$ points in $\mathcal{X}_0$ selected without replacement, where $\gamma \leq \frac{1}{2}$ will be chosen later, and $N_0 \geq T \geq \Omega\left(\frac{k \cdot d \cdot \log \gamma^{-1}}{\gamma^2} + \frac{V(n,d,k,\varepsilon,\delta) + k\varepsilon^{-1} \log n}{\gamma}\right)$. Then, by Lemma A.20 with $\kappa$ set to $\frac{1}{2}$, we have that $\mathrm{cost}(\mathcal{Y}_0; \mathcal{C}) \in (1 \pm 0.5) \cdot \frac{T}{N_0} \cdot \mathrm{cost}(\mathcal{X}_0; \mathcal{C}) \pm \gamma \cdot T \cdot R^p$ for any set $\mathcal{C}$ of size at most $k$. Now, consider applying $\mathcal{B}$ (the algorithm created by Lemma C.1) to the dataset $\mathcal{Y}_0$, restricted to points in the ball $B(\mu, R)$. This would obtain a set $\hat{\mathcal{Y}}_0$, such that for any set $\mathcal{C}$ of size $k$,

$$
\begin{aligned}
\frac{N_0}{T} \cdot \mathrm{cost}(\hat{\mathcal{Y}}_0; \mathcal{C}) &\leq O\left(\frac{N_0}{T}\right) \cdot \left[\mathrm{OPT}(\mathcal{Y}_0) + \mathrm{cost}(\mathcal{Y}_0; \mathcal{C}) + \left(V(n,d,k,\varepsilon,\delta) + k\varepsilon^{-1} \log n\right) \cdot R^p\right] \\
&\leq O(1) \cdot \left[\mathrm{OPT}(\mathcal{X}_0) + \mathrm{cost}(\mathcal{X}_0; \mathcal{C})\right] + O\left(\gamma \cdot N_0 \cdot R^p\right) \\
&= O(1) \cdot \mathrm{cost}(\mathcal{X}_0; \mathcal{C}) + O\left(\gamma \cdot N_0 \cdot R^p\right),
\end{aligned} \tag{7}
$$

and

$$
\begin{aligned}
\mathrm{cost}(\mathcal{X}_0; \mathcal{C}) &\leq O\left(\frac{N_0}{T}\right) \cdot \left[\mathrm{cost}(\mathcal{Y}_0; \mathcal{C}) + \gamma \cdot T \cdot R^p\right] \\
&\leq O\left(\frac{N_0}{T}\right) \cdot \left[\mathrm{OPT}(\mathcal{Y}_0) + \mathrm{cost}(\hat{\mathcal{Y}}_0; \mathcal{C}) + \gamma \cdot T \cdot R^p\right] \\
&\leq O\left(\frac{N_0}{T}\right) \cdot \mathrm{cost}(\hat{\mathcal{Y}}_0; \mathcal{C}) + O(1) \cdot \mathrm{OPT}(\mathcal{X}_0) + O(\gamma \cdot N_0 \cdot R^p).
\end{aligned} \tag{8}
$$

In addition, by Lemma A.21, we have that $\hat{\mathcal{Y}}_0$ is $(O(\varepsilon), O(\delta))$-DP with respect to $\mathcal{X}_0$, since $\mathcal{Y}_0$ is a randomly sampled subset of $\mathcal{X}_0$ and $\hat{\mathcal{Y}}_0$ is $(O(\varepsilon), O(\delta))$-DP with respect to $\mathcal{Y}_0$.

We now state the main result of this section, which provides a parallel and differentially private algorithm for $k$-means (or $k$-median) with an $O(1)$-multiplicative approximation. We will focus on $k$-means in the proof, but an identical argument also holds for $k$-median.

**Theorem C.2.** *Suppose we have a sequential $(\varepsilon, \delta)$-DP algorithm $\mathcal{A}$ that takes as input a dataset $\mathcal{X}$ contained in $B(0, \Lambda)$, and outputs a set of $k$ centers $\mathcal{G}$ such that $\mathrm{cost}(\mathcal{X}; \mathcal{G}) \leq O(1) \cdot \mathrm{OPT}(\mathcal{X}) + V(n,d,k,\varepsilon,\delta) \cdot \Lambda^p$. In addition, suppose we have a near-linear time $(\varepsilon, \delta)$-private MPC algorithm that can generate an $(\alpha, \beta)$-bicriteria approximate solution to $k$-means (resp., $k$-median) clustering with additive error $W = W(n,d,k,\varepsilon,\delta)$, where $\alpha = d^{O(1)}$ and $\beta = (\log n)^{O(1)}$. Finally, assume that each machine can store at least $k^{1+\eta}(\log n)^{O(1)} + T$ points, where $T = (d + \log n)^{O(1)} \cdot \Theta(V(n,d,k,\varepsilon,\delta) + k/\varepsilon)$ is some threshold and $\eta > 0$ is any small constant.*

*Then, there exists an $(O(\varepsilon), O(\delta))$-DP algorithm for $k$-means (or $k$-median) clustering with multiplicative error $O(1)$ and additive error*

$$
\mathrm{poly}(d, \log n) \cdot \left(W(n,d,k,\varepsilon,\delta) + k^2\varepsilon^{-1} + k \cdot V(n,d,k,\varepsilon,\delta)\right).
$$

*In addition, the algorithm has total sequential time $\tilde{O}(nd) + T^{O(1)}$ and parallel time $\tilde{O}(n^\theta d) + T^{O(1)}$, i.e., the runtime is near-linear assuming $n \geq T^C$ for some constant $C$.*

**Algorithm:** Suppose we are given a set $\mathcal{X}$ of points across many machines. We start by running the $(\varepsilon, \delta)$-DP MPC algorithm, which outputs a set $\mathcal{F} = \{f_1, \dots, f_{\beta \cdot k}\}$ of size at most $\beta \cdot k$, such that $\mathrm{cost}(\mathcal{X}; \mathcal{F}) \leq \alpha \cdot \mathrm{OPT}(\mathcal{X}) + W(n,d,k,\varepsilon,\delta) \cdot \Lambda^p$.

Now, for each $1 \leq j \leq \beta \cdot k$ and each $1 \leq r \leq \log_2(n^2)$, let $B_{j,r}$ represent the ball of radius $2^r \cdot \frac{\Lambda}{n^2}$ around the point $f_j \in \mathcal{F}$. Now, for each point $x \in \mathcal{X}$, we assign it to some $(j, r)$ where $j \leq \beta \cdot k, r \leq \log_2(n^2)$, using a $L = O(\log n)$-approximate nearest neighbor data structure on the dataset $\mathcal{F}$. More precisely, for each $x \in \mathcal{X}$, we find an $L$-approximate nearest neighbor $f_j$, and then find the smallest $r$ such that $x \in B_{j,r}$. For $L = O(\log n)$, we can store the data structure with failure probability $\tau = \frac{1}{\mathrm{poly}(n)}$ using space $O(\beta \cdot k \cdot (\log n)^{O(1)})$, and compute the $L$-approximate nearest neighbor in $\mathcal{F}$ per point $x \in \mathcal{X}$ in time $(\log n)^{O(1)}$.

Now, for each $j \leq \beta \cdot k$ and $r \leq \log_2(n^2)$, we define $\mathcal{X}_{j,r}$ as the set of points assigned to $B_{j,r}$. Via MPC aggregation (Lemma A.10), we compute $N_{j,r} = |\mathcal{X}_{j,r}|$ and $\hat{N}_{j,r} = N_{j,r} + \mathrm{Lap}(1/\varepsilon)$ for each

$j, r$. We also define some threshold parameter $T = \Theta\left(\frac{k \cdot d \cdot \log \gamma^{-1}}{\gamma^2} + \frac{V(n,d,k,\varepsilon,\delta) + k\varepsilon^{-1}\log n}{\gamma}\right)$ (where $\gamma$ will be chosen later) - we will see that $T$ matches the threshold in the theorem statement. Now, for each $j, r$, if $\hat{N}_{j,r} \geq 2T$, let $\mathcal{Y}_{j,r}$ be a random sample of $T$ points from $\mathcal{X}_{j,r}$, which we obtain using Lemma A.11. Next, we use Lemma C.1 to compute a private approximation $\hat{\mathcal{Y}}_{j,r}$ of $\mathcal{Y}_{j,r}$ based on our sequential private algorithm $\mathcal{A}$, where each point in $\hat{\mathcal{Y}}_{j,r}$ is scaled by a factor of $\frac{\hat{N}_{j,r}}{T}$. Otherwise, if $\hat{N}_{j,r} < 2T$, we let $\hat{\mathcal{Y}}_{j,r} = \emptyset$.

Finally, we compute a non-private, scalable MPC $k$-means (or $k$-median) algorithm on $\bigcup_{j,r} \hat{\mathcal{Y}}_{j,r}$ with an $O(1)$ approximation factor, such as using Theorem A.12.

**Runtime:** We start by using the bicriteria of Section B, which takes near-linear time and $O(1)$ rounds. Next, the assignment of each point $x \in \mathcal{X}$ to $(j, r)$ can be done in near-linear time as well, as we can compute the ANN data structure for $\mathcal{F}$ on a single machine using $O(\beta \cdot k \cdot (\log n)^{O(1)})$ space, and then broadcast the data structure to all machines in $O(1)$ rounds. The broadcasting only needs $O(1)$ rounds since the space on each machine is at least $k^{1+\eta}(\log n)^{O(1)}$, which means that $n^{\theta(1-\eta)} \geq \tilde{O}(\beta \cdot k)$. Then, each point can be sent to an $O(\log n)$-approximate nearest neighbor in poly $\log n$ time per point.

Next, computing the values $N_{j,r}$ and $\hat{N}_{j,r}$ are simple via MPC aggregation (Lemma A.10), and we can sample $T$ points from each $(j, r)$ with $\hat{N}_{j,r} \geq 2T$ using MPC sampling (Lemma A.11). Then, in linear time and 1 round of communication, we can assign each $(j, r)$ to some machine, and then send the sampled points $\mathcal{Y}_{j,r}$ to the corresponding machine. Note that $\mathcal{Y}_{j,r}$ has size at most $T$, so it fits on a machine, and the number of machines needed is $O(\log n \cdot \beta \cdot k) = k \cdot (\log n)^{O(1)}$. Then, we must compute a private approximation $\hat{\mathcal{Y}}_{j,r}$ for each $\mathcal{Y}_{j,r}$ using Lemma C.1, which can be done in time poly$(T)$ in each machine. This is because $\mathcal{Y}_{j,r}$ already has at most $T$ distinct elements, so applying $\mathcal{A}$ takes poly$(T)$ time [70], and by Lemma C.1, we only need an additional $O(T \cdot k^\eta \cdot d)$ time to compute $\hat{\mathcal{Y}}_{j,r}$. In addition, there are only $O(k \cdot \text{poly} \log n)$ machines in total for this to be done, so the total sequential and total parallel runtimes are both at most poly$(T, d, \log n) = T^{O(1)}$. Finally, we apply Theorem A.12 with $\gamma = 0.5$ on $\hat{\mathcal{Y}}$. Because each $\hat{\mathcal{Y}}_{j,r}$ has at most $k$ distinct points, $\hat{\mathcal{Y}}$ has at most poly$(k, \log n)$ distinct points. Therefore, the total time for the final step of applying Theorem A.12 is at most poly$(k, d, \log n)$. We will later show that our threshold parameter $T$ exactly matches the choice of $T$ in the theorem statement, which will complete the proof of the runtime argument.

**Privacy:** First, note that the initial set $\mathcal{F}$ is $(\varepsilon, \delta)$-DP, and the ANN data structure constructed from $\mathcal{F}$ only depends on $\mathcal{F}$. So, fixing this data structure, adding, removing, or changing a single point in $\mathcal{X}$ can only change at most two of the $\mathcal{X}_{j,r}$'s, each by at most 1 point. So, the set of values $\hat{N}_{j,r}$ are also $(2\varepsilon, 0)$-DP. If we fix $\mathcal{F}$ and $\hat{N}_{j,r}$ for all $j, r$, then for each $\mathcal{X}_{j,r}$ that changes between two adjacent datasets, we have that $\hat{\mathcal{Y}}_{j,r}$ is $(O(\varepsilon), O(\delta))$-DP for the same reason that $\hat{\mathcal{Y}}_0$ was $(O(\varepsilon), O(\delta))$-DP with respect to $\mathcal{X}_0$. Therefore, the set of $\hat{\mathcal{Y}}_{j,r}$ is $(O(\varepsilon), O(\delta))$-DP given $\mathcal{F}$ and $\hat{N}_{j,r}$ since at most 2 of the $\mathcal{X}_{j,r}$'s change. So, by adaptive composition, the final construction of $\hat{\mathcal{Y}}$ is $(O(\varepsilon), O(\delta))$-DP.

**Accuracy:** We focus on the $k$-means setting for simplicity, but the proof is almost identical for the $k$-median setting. Because $x$ being assigned to $(j, r)$ means that $x \in B_{j,r}$ but $x \notin B_{j,r-1}$, this means that $\frac{\Lambda}{n^2} \cdot 2^r \leq 2 \cdot d(x, f_j) + \frac{\Lambda}{n^2} \leq 2L \cdot d(x, \mathcal{F}) + \frac{\Lambda}{n^2}$. If we define $r(x) = \frac{\Lambda}{n^2} \cdot 2^r$ if $x$ is assigned to some $(j, r)$, then we have $r(x) \leq 2L \cdot d(x, \mathcal{F}) + \frac{\Lambda}{n^2}$, which means

$$\sum_{x \in \mathcal{X}} r(x)^2 \leq \sum_{x \in \mathcal{X}} 4L^2 \cdot \left(d(x, \mathcal{F}) + \frac{\Lambda}{n^2}\right)^2$$
$$\leq 8L^2 \cdot \sum_{x \in \mathcal{X}} \left[d(x, \mathcal{F})^2 + \frac{\Lambda^2}{n^4}\right]$$
$$\leq O(\alpha \cdot \log^2 n) \cdot \text{OPT}(\mathcal{X}) + O(\log^2 n) \cdot W(n, d, k, \varepsilon, \delta) \cdot \Lambda^2.$$

The final inequality holds since $\text{cost}(\mathcal{X}; \mathcal{F}) \leq \alpha \cdot \text{OPT}(\mathcal{X}) + W(n, d, k, \varepsilon, \delta) \cdot \Lambda^2$ and $L = O(\log n)$.

Recall that we have computed some $\hat{\mathcal{Y}}_{j,r}$ for each $j \leq \beta \cdot k$ and $r \leq \log_2(n^2)$, which is contained in the ball $B$ around $f_j$ of radius $\frac{\Lambda}{n^2} \cdot 2^r$. If $\hat{N}_{j,r} \geq 2T$, then for any set $\mathcal{C}$ of $k$ points in $B(0, \Lambda)$, we have that by Equations (7) and (8),

$$\text{cost}(\hat{\mathcal{Y}}_{j,r}; \mathcal{C}) \leq O(1) \cdot \text{cost}(\mathcal{X}_{j,r}; \mathcal{C}) + O(\gamma \cdot N_{j,r}) \cdot \left( \frac{\Lambda}{n^2} \cdot 2^r \right)^2,$$

and

$$\text{cost}(\mathcal{X}_{j,r}; \mathcal{C}) \leq O(1) \cdot \text{cost}(\hat{\mathcal{Y}}_{j,r}; \mathcal{C}) + O(1) \cdot \text{OPT}(\mathcal{X}_{j,r}) + O(\gamma \cdot N_{j,r}) \cdot \left( \frac{\Lambda}{n^2} \cdot 2^r \right)^2.$$

We remark that above, we scaled each point in $\hat{\mathcal{Y}}_{j,r}$ by $\hat{N}_{j,r}/T$, which is $\Theta(N_{j,r}/T)$ with overwhelming probability.

Let $\hat{\mathcal{Y}}$ be the union over all of the $\hat{\mathcal{Y}}_{j,r}$'s (with their corresponding weights). Adding over all machines, we have that

$$\text{cost}(\hat{\mathcal{Y}}; \mathcal{C}) \leq O(1) \cdot \text{cost}(\mathcal{X}; \mathcal{C}) + O(\gamma) \cdot \sum_{j,r} N_{j,r} \cdot \left( \frac{\Lambda}{2^n} \cdot 2^r \right)^2$$

$$= O(1) \cdot \text{cost}(\mathcal{X}; \mathcal{C}) + O(\gamma) \cdot \sum_{x \in \mathcal{X}} (r(x))^2$$

$$\leq O(1) \cdot \text{cost}(\mathcal{X}; \mathcal{C}) + O(\gamma \cdot \alpha \cdot \log^2 n) \cdot \text{OPT}(\mathcal{X}) + O(\gamma \cdot \log^2 n) \cdot W(n, d, k, \varepsilon, \delta) \cdot \Lambda^2,$$

and that

$$\text{cost}(\mathcal{X}; \mathcal{C}) = \sum_{j,r: \hat{N}_{j,r} \geq 2T} \text{cost}(\mathcal{X}_{j,r}; \mathcal{C}) + \sum_{j,r: \hat{N}_{j,r} < 2T} \text{cost}(\mathcal{X}_{j,r}; \mathcal{C})$$

$$\leq O(1) \cdot \text{cost}(\hat{\mathcal{Y}}; \mathcal{C}) + O(1) \cdot \text{OPT}(\mathcal{X}) + O(\gamma) \cdot \sum_{j,r} N_{j,r} \cdot \left( \frac{\Lambda}{2^n} \cdot 2^j \right)^2 + \sum_{j,r: \hat{N}_{j,r} < 2T} \text{cost}(\mathcal{X}_{j,r}; \mathcal{C})$$

$$\leq O(1) \cdot \text{cost}(\hat{\mathcal{Y}}; \mathcal{C}) + O(1 + \gamma \cdot \alpha \cdot \log^2 n) \cdot \text{OPT}(\mathcal{X}) + O(\gamma \cdot \log^2 n) \cdot W(n, d, k, \varepsilon, \delta) \cdot \Lambda^2$$
$$+ O(T \cdot \beta k \cdot \log n) \cdot \Lambda^2.$$

Hence, if we are able to obtain an $O(1)$-approximate clustering $\mathcal{C}$ for $\hat{\mathcal{Y}}$ in the MPC setting, we will have that $\text{cost}(\hat{\mathcal{Y}}; \mathcal{C}) \leq O(1) \cdot \text{OPT}(\hat{\mathcal{Y}}) \leq O(1 + \gamma \cdot \alpha \cdot \log^2 n) \cdot \text{OPT}(\mathcal{X}) + O(\gamma \cdot \log^2 n) \cdot W(n, d, k, \varepsilon, \delta)$. Therefore, we have that

$$\text{cost}(\mathcal{X}; \mathcal{C}) \leq O(1 + \gamma \cdot \alpha \cdot \log^2 n) \cdot \text{OPT}(\mathcal{X}) + O\left( \gamma \cdot \log^2 n \cdot W(n, d, k, \varepsilon, \delta) + T \cdot \beta \cdot \log n \cdot k \right) \cdot \Lambda^2.$$

Hence, we can set $\gamma = \frac{1}{\alpha \log^2 n}$, so if we treat $\alpha = \text{poly}(d)$ and $\beta = \log^2 n$, then

$$T = \Theta\left( \frac{k \cdot d \cdot \log \gamma^{-1}}{\gamma^2} + \frac{V(n, d, k, \varepsilon, \delta) + k\varepsilon^{-1} \log n}{\gamma} \right) = \text{poly}(d, \log n) \cdot \Theta\left( V(n, d, k, \varepsilon, \delta) + k \cdot \varepsilon^{-1} \right).$$

Hence, we obtain a multiplicative cost of $O(1)$ and an additive cost of

$$\text{poly}(d, \log n) \cdot \left( W(n, d, k, \varepsilon, \delta) + k \cdot V(n, d, k, \varepsilon, \delta) + k^2 \varepsilon^{-1} \right).$$

This concludes the proof of accuracy.

To finish, we can set the functions $V(n, d, k, \varepsilon, \delta) = \text{poly}(\log n, \log d, \varepsilon^{-1}, \log \delta^{-1}) \cdot (k^{1.01} d^{0.51} + k^{1.5})$ based on [70][4] and $W(n, d, k, \varepsilon, \delta) = k \cdot \text{poly}(\log n, d, \varepsilon^{-1})$ based on Section B to obtain the following.

**Theorem C.3.** *There exists an $(\varepsilon, \delta)$-DP algorithm for $k$-means (or $k$-median) clustering with multiplicative error $O(1)$ and additive error $k^{2.5} \cdot \text{poly}\left( d, \log n, \varepsilon^{-1}, \log \delta^{-1} \right)$. In addition, the algorithm can be implemented in MPC, assuming each machine can store $\tilde{O}(n^\theta) \geq k^{1.5} \cdot \text{poly}(d, \log n, \varepsilon^{-1}, \log \delta^{-1})$ points, with $O(1)$ total rounds of communication, total sequential running time $\tilde{O}(nd) + \text{poly}(k, d, \varepsilon^{-1}, \log \delta^{-1})$, and total time per machine $\tilde{O}(n^\theta d) + \text{poly}(k, d, \varepsilon^{-1}, \log \delta^{-1})$.*

---

[4]While [70] only writes their proof for the $k$-means case, their argument also goes through for $k$-median.

# D   Obtaining an arbitrarily good approximation (for low dimensions)

## D.1   Converting $O(1)$-approximation to sequential coreset

In this subsection, we only deal with sequential algorithms (so we ignore the MPC model), and show how to convert a differentially private $O(1)$-approximate $k$-means (or $k$-median) algorithm into a coreset containing roughly $\operatorname{poly}(k, \log n, e^d)$ distinct points, in roughly $n \cdot \operatorname{poly}(k, e^d)$ time, where $d$ is the dimension. This will allow us to reconstruct many of the results of Ghazi et al. [44] from an $O(1)$-approximation such as by Kaplan and Stemmer [70], in a more efficient manner as our runtime has only linear dependence on $n$ as opposed to polynomial dependence. While the exponential dependence on $d$ will be quite large, we will show later that we can reduce $d$ to $O(\log k)$, which gives us a $\tilde{O}(n) \cdot \operatorname{poly}(k)$-time algorithm for generating a coreset. We will focus on the $k$-means problem, but the same (in fact, even simpler) analysis works for $k$-median also.

**Theorem D.1.** *For any fixed constant $0 < \gamma < 1$, there exists an $(\varepsilon, \delta)$-DP sequential algorithm that operates on a dataset $\mathcal{X} \subset B(0, \Lambda)$ of size $n$, with the following properties. Then, in running time $\tilde{O}(n) \cdot \operatorname{poly}(k, O(1/\gamma)^d, \varepsilon^{-1}, \log \delta^{-1})$, the algorithm computes a weighted coreset $\hat{\mathcal{Y}}$ with at most $O(k \log n) \cdot O(1/\gamma)^{O(d)}$ distinct points, such that for any set $\mathcal{C} \subset \mathbb{R}^d$ of size at most $k$ that has nonzero intersection with the slightly larger ball $B(0, \gamma^{-1} \cdot \Lambda)$, we have*

$$\operatorname{cost}(\hat{\mathcal{Y}}; \mathcal{C}) \in (1 \pm O(\gamma)) \cdot \operatorname{cost}(\mathcal{X}; \mathcal{C}) \pm \operatorname{poly}\left(k, (1/\gamma)^{O(d)}, \log n, \varepsilon^{-1}, \log \delta^{-1}\right) \cdot \Lambda^2.$$

**Algorithm:**   Given a dataset $\mathcal{X} = \{x_1, \ldots, x_n\} \in B(0, \Lambda)$, we start by applying a black-box $(\varepsilon, \delta)$-DP algorithm that outputs $\mathcal{G} = \{g_1, \ldots, g_k\}$ of centers, such that $\operatorname{cost}(\mathcal{X}; \mathcal{G}) \le O(1) \cdot \operatorname{OPT}(\mathcal{X}) + U(n, d, k, \varepsilon, \delta) \cdot \Lambda^2$. (For $k$-median, the $\Lambda^2$ would be $\Lambda$.) Now, for each $1 \le j \le k$ and each $1 \le r \le \log_2(n^2)$, we define $\mathcal{T}_{j,r}$ to be the subset of $\mathbb{R}^d$ that is closest to cluster center $g_j \in \mathcal{G}$ and has distance from $g_j$ in the range $[\frac{\Lambda}{n^2} \cdot 2^{r-1}, \frac{\Lambda}{n^2} \cdot 2^r)$. We will not explicitly compute $\mathcal{T}_{j,r}$, but for a given point $x \in \mathcal{X}$, one can easily determine which region it belongs to in $O(kd)$ time.

Given this, we will show how to construct a private coreset of the data. For each $j \in [k]$ and $r \le \log_2(n^2)$, define $B_{j,r}$ to be the ball of radius $\frac{\Lambda}{n^2} \cdot 2^r$ around $g_j$. We use Lemma A.18 to create an efficiently list-decodable $\gamma$-cover $\mathcal{H}_{j,r}$ of $B_{j,r}$ at distance 1. Importantly, the size of the cover is at most $(1/\gamma)^{O(d)}$, and this covers $\mathcal{T}_{j,r}$ which is a subset of the ball. In addition, we can compute all the points in the cover in time $(1/\gamma)^{O(d)}$. So, for each $t \in [(1/\gamma)^{O(d)}]$, we can let the point $y_{j,r,t}$ be the $t^{\text{th}}$ point in the cover. Now, for each $(j, r)$, we map each point $x_i \in \mathcal{X} \cap \mathcal{T}_{j,r}$ to its closest point $y_{j,r,t} \in \mathcal{H}_{j,r}$, and aggregate to compute an overall vector $v = \{v_{j,r,t}\}$ which counts the number of points in $\mathcal{X}$ mapped to $y_{j,r,t}$. Next, we let $\tilde{v}$ be the vector where we replace each $v_{j,r,t}$ with $\tilde{v}_{j,r,t} = \max(0, v_{j,r,t} + \operatorname{Lap}(1/\varepsilon))$. Our final coreset $\hat{\mathcal{Y}}$ will be the set of points $y_{j,r,t}$ each with multiplicity $\tilde{v}_{j,r,t}$.

**Runtime and size:**   We note that applying the black-box algorithm of either Kaplan and Stemmer [70] or Ghazi et al. [44] to obtain $\mathcal{G}$ may require $\operatorname{poly}(n, d)$ time. However, we can get $\tilde{O}\left(nd + \operatorname{poly}(k, d, \varepsilon^{-1}, \log \delta^{-1})\right)$ runtime by using the algorithm we devised in Section C. As the algorithm only needs to work in the sequential setting, we remark that many aspects of this algorithm can be simplified, while still obtaining the same accuracy, privacy, and runtime guarantees.

Next, in $O(knd)$ time, we can map each point $x \in \mathcal{X}$ to its region $\mathcal{T}_{j,r}$. Then, for each $(j, r)$, we compute a $\gamma$-cover $\mathcal{H}_{j,r}$, which takes time at most $O(k \cdot \log n) \cdot O(1/\gamma)^{O(d)}$. Finally, mapping each point $x \in \mathcal{X} \cap \mathcal{T}_{j,r}$ to its closest center in $\mathcal{H}_{j,r}$ for all $j, r$ takes total time at most $n \cdot O(1/\gamma)^{O(d)}$, and aggregating the sets and adding Laplace noise to create $\hat{\mathcal{Y}}$ takes time at most $O(k \cdot \log n) \cdot O(1/\gamma)^{O(d)}$. So, the overall runtime is $\tilde{O}(n) \cdot \operatorname{poly}\left(k, \varepsilon^{-1}, \log \delta^{-1}, O(1/\gamma)^{O(d)}\right)$.

Finally, we remark that the number of distinct points in the cover $\mathcal{H}_{j,r}$ is at most $O(1/\gamma)^{O(d)}$, so the total number of distinct points in the coreset is at most $O(k \cdot \log n) \cdot O(1/\gamma)^{O(d)}$.

**Privacy:**   We assume the original construction of $\mathcal{G}$ is $(\varepsilon, \delta)$-private. Then, the vector $\tilde{v}$ will also be $(3\varepsilon, \delta)$-DP. To see why, if we treat $\mathcal{G}$ as fixed, changing a single point in $\mathcal{X}$ changes at most two values of $v_{j,r,t}$, so the $\ell_1$-sensitivity of $v$ is at most 2. Since we add $\operatorname{Lap}(1/\varepsilon)$ error to each coordinate, we

incur at most an additional $(2\varepsilon, 0)$-privacy loss. So, by adaptive composition, $\tilde{v}$ is $(3\varepsilon, \delta)$-DP. In addition, the covers $\mathcal{H}_{j,r}$ only depend on $\mathcal{G}$, so the overall algorithm is also $(3\varepsilon, \delta)$-DP.

**Accuracy:**   Consider any set of $k$ centers $\mathcal{C} = \{c_1, \ldots, c_k\}$, where at least one point $c_i$ is in $B(0, \gamma^{-1} \cdot \Lambda)$. In addition, suppose that we replaced $\hat{\mathcal{Y}}$ with the set $\mathcal{Y}$ where we used multiplicities based on the vector $v$ instead of $\tilde{v}$. Suppose a point $x_i$ has distance $\tilde{d}_i$ from its closest center $g_j$ in the solution $\mathcal{G}$ and distance $d_i$ from its closest center in the solution $\mathcal{C}$. Then, $x_i \in \mathcal{T}_{j,r}$ for some $r$, meaning that $\tilde{d}_i \leq \frac{\Lambda}{n^2} \cdot 2^r \leq 2\tilde{d}_i$. Then, $x_i$ is moved to a point of distance at most $2\gamma \cdot \tilde{d}_i$ away, by the property of the cover $\mathcal{H}_{j,r}$. So, the induced error per point is at most

$$(d_i + 2\gamma \cdot \tilde{d}_i)^2 - d_i^2 = 4\gamma d_i \tilde{d}_i + 4\gamma^2 \tilde{d}_i^2 \leq 2\gamma(d_i^2 + \tilde{d}_i^2) + 4\gamma^2 \tilde{d}_i^2 \leq 2\gamma d_i^2 + 6\gamma \tilde{d}_i^2.$$

Finally, noting that $d_i$ must be at most $2\gamma^{-1} \cdot \Lambda$ because every point $x \in B(0, \Lambda)$ and $\mathcal{C}$ has nonzero intersection with $B(0, \gamma^{-1} \cdot \Lambda)$, the overall error is at most

$$2\gamma \cdot \sum_{i=1}^{n} d_i^2 + 6\gamma \cdot \sum_{i=1}^{n} \tilde{d}_i^2 = 2\gamma \cdot \text{cost}(\mathcal{X}; \mathcal{C}) + 6\gamma \cdot \text{cost}(\mathcal{X}; \mathcal{G})$$

$$\leq 2\gamma \cdot \text{cost}(\mathcal{X}; \mathcal{C}) + 6\gamma \cdot [O(1) \cdot \text{OPT}(\mathcal{X}) + U(n, d, k, \varepsilon, \delta) \cdot (2\gamma^{-1} \cdot \Lambda)^2]$$

$$\leq O(\gamma) \cdot \text{cost}(\mathcal{X}; \mathcal{C}) + O(\gamma^{-1} \cdot U(n, d, k, \varepsilon, \delta) \cdot \Lambda^2).$$

Hence, this means that for any set of $k$ centers $\mathcal{C}$, the cost of the original dataset $\mathcal{X} = \{x_1, \ldots, x_n\}$ and the modified weighted set created by the vector $v$ have multiplicative cost ratios $1 \pm O(\gamma)$ and additive error $U(n, d, k, \varepsilon, \delta) = \text{poly}(k, d, \log n, \varepsilon^{-1}, \log \delta^{-1}) \cdot \Lambda^2$, using Theorem C.3.

Finally, we ask what happens when we replace $v$ with $\tilde{v}$? In expectation, each $v_{j,r,t}$ changes by $O(\varepsilon^{-1})$, which changes the overall cost with respect to $\mathcal{C}$ by at most $O(\varepsilon^{-1} \cdot k \cdot \log n \cdot (1/\gamma)^{O(d)}) \cdot O(\gamma^{-1}\Lambda)^2$ with high probability. Hence, we obtain an $(O(\gamma), U)$-coreset, where

$$U = O\left(k \cdot \varepsilon^{-1} \cdot \log n \cdot (1/\gamma)^d + \gamma^{-1} \cdot U(n, d, k, \varepsilon, \delta)\right) = \text{poly}\left(k, \log n, \varepsilon^{-1}, \log \delta^{-1}, (1/\gamma)^{O(d)}\right).$$

To finish, we provide algorithm pseudocode, as Algorithm 4.

---

**Algorithm 4** A sequential coreset algorithm for differentially private $k$-means (or $k$-median).

---

1: **procedure** SEQUENTIALCORESET($\mathcal{X}, \varepsilon, \delta, \gamma$)                     ▷ Will be $(O(\varepsilon), O(\delta))$-DP.
2:     Use Algorithm 1 (or related procedure) to find private $O(1)$-approximate centers $\mathcal{G} = \{g_1, \ldots, g_k\}$.
3:     **for** each $j \leq k$, $r \leq \log_2(n^2)$ **do**
4:         Let $B_{j,r}$ be the ball of radius $\frac{\Lambda}{n^2} \cdot 2^r$ around $g_j$.
5:         Construct an efficiently list-decodable $\gamma$-cover $\mathcal{H}_{j,r}$ of $B_{j,r}$.
6:         **for** each $t \leq (1/\gamma)^{O(d)}$ **do**
7:             Let $y_{j,r,t}$ be the $t$th point in $\mathcal{H}_{j,r}$.
8:     **for** all $x_i \in \mathcal{X}$ **do**
9:         Assign $x_i$ to $(j, r, t)$ if $g_j$ is $x_i$'s closest center in $\mathcal{G}$, $d(x_i, g_j) \approx \frac{\Lambda}{n^2} \cdot 2^r$, and $y_{j,r,t}$ is the closest point to $x_i$ in $\mathcal{H}_{j,r}$.
10:     **for** each $j \leq k$, $r \leq \log_2(n^2)$, $t \leq (1/\gamma)^{O(d)}$ **do**
11:         Let $v_{j,r,t}$ be the number of points $x_i$ assigned to $y_{j,r,t}$
12:         Let $\tilde{v}_{j,r,t} = \max(0, v_{j,r,t} + \text{Lap}(1/\varepsilon))$
13:     Return $\hat{\mathcal{Y}}$ : set of points $y_{j,r,t}$ with multiplicity $\tilde{v}_{j,r,t}$.

---

### D.2   Converting a sequential coreset to a parallel coreset

In this subsection, we combine the sequential coreset we generated in Subsection D.1 with the parallel bicriteria approximation we generated in Section B to generate a differentially private parallel coreset. This will allow us to obtain an approximation factor that is arbitrarily close to the best-known approximation factor of 5.912 for $k$-means and 2.406 for $k$-median [25]. While both the error and runtime will have exponential dependence on the dimension $d$, we will later show how to reduce $d$ to roughly $\log k$, which will imply only polynomial dependence on $k$.

**Theorem D.2.** *For any fixed $0 < \gamma < 1$, there exists an $(\varepsilon, \delta)$-DP algorithm that operates on a dataset $\mathcal{X} \subset B(0, \Lambda)$ of size $n$ and outputs $\hat{\mathcal{Y}}$ with at most $\mathrm{poly}(k, O(1/\gamma)^{O(d)}, \log n)$ distinct points, such that for any set $\mathcal{C} \subset B(0, \Lambda)$ of size $k$,*

$$\mathrm{cost}(\hat{\mathcal{Y}}; \mathcal{C}) \in (1 \pm O(\gamma)) \cdot \mathrm{cost}(\mathcal{X}; \mathcal{C}) \pm \mathrm{poly}\left(k, (1/\gamma)^{O(d)}, \log n, \varepsilon^{-1}, \log \delta^{-1}\right) \cdot \Lambda^2.$$

*($\Lambda^2$ is replaced with $\Lambda$ in the $k$-median case.) In addition, if each machine can store $\tilde{O}(n^\theta) \geq \mathrm{poly}\left(k, \log n, \varepsilon^{-1}, \log \delta^{-1}, (1/\gamma)^{O(d)}\right)$ points, then the algorithm can be implemented in MPC with $O(1)$ rounds of communication, total sequential time $\tilde{O}(nkd) + \mathrm{poly}(k, O(1/\gamma)^d, \varepsilon^{-1}, \log \delta^{-1})$, and parallel time $\tilde{O}(n^\theta kd) + \mathrm{poly}(k, O(1/\gamma)^d, \varepsilon^{-1}, \log \delta^{-1})$ per machine.*

**Algorithm:** We start with a dataset $\mathcal{X}$ of $n$ points in $B(0, \Lambda)$, spread out across many machines. We start by applying the parallel and $(\varepsilon, 0)$-DP $(O(d^3), \log^2 n)$-bicriteria of Section B with additive error $W(n, k, \varepsilon, \delta) \cdot \Lambda^2$ (where $W(n, d, k, \varepsilon, \delta) = k \cdot \mathrm{poly}(\log n, d, \varepsilon^{-1})$) to generate a set of points $\mathcal{F} = \{f_1, \ldots, f_K\}$ where $K = k \cdot \beta$. For each $1 \leq j \leq K$ and each $1 \leq r \leq R := \log_2(n^2)$, we define $\mathcal{T}_{j,r}$ to be the subset of $\mathbb{R}^d$ that is closest to cluster center $s_j \in S$ and has distance from $s_j$ in the range $[\frac{\Lambda}{n^2} \cdot 2^{r-1}, \frac{\Lambda}{n^2} \cdot 2^r)$. In addition, define $\mathcal{X}_{j,r}$ to be the set of points $\mathcal{X} \cap \mathcal{T}_{j,r}$. Finally, define $B_{j,r} \supset \mathcal{T}_{j,r}$ to be the ball of radius $\frac{\Lambda}{n^2} \cdot 2^r$ around $s_j$.

Now, for each pair $(j, r)$, we compute $N_{j,r} := |\mathcal{X}_{j,r}|$ and $\hat{N}_{j,r} := N_{j,r} + \mathrm{Lap}(1/\varepsilon)$. Now, we let $T = \mathrm{poly}\left(k, (1/\gamma)^{O(d)}, \log n, \varepsilon^{-1}, \log \delta^{-1}\right)$ be some sufficiently large threshold parameter. Next, if $\hat{N}_{j,r} \geq 2T$, we sample $T$ points uniformly at random from $\mathcal{X}_{j,r}$ (call this set $\mathcal{Z}_{j,r}$) and send it to a specific machine. Note that $\mathcal{Z}_{j,r} \subset B_{j,r}$. Finally, we apply Theorem D.1 to the sample $\mathcal{Z}_{j,r}$ to generate a private coreset $\hat{\mathcal{Y}}_{j,r}$, except that we replace $B(0, \Lambda)$ with $B_{j,r}$.

Finally, we scale each $\hat{\mathcal{Y}}_{j,r}$ by $\frac{\hat{N}_{j,r}}{T}$, and let $\hat{\mathcal{Y}} = \bigcup_{j,r} \hat{\mathcal{Y}}_{j,r}$ be our final coreset.

**Runtime:** First, note that constructing $\mathcal{F}$ takes near-linear time with $O(1)$ rounds of communication, and we can broadcast $\mathcal{F}$ to all machines in $O(1)$ rounds. We do not need to explicitly compute $\mathcal{T}_{j,r}$, but for each point $x \in \mathcal{X}$, we can determine which $(j, r)$ $x$ is assigned to in time $O(kd)$ per point, so the total time per machine is $O(n^\theta \cdot k \cdot d)$. Next, we can use MPC aggregation (Lemma A.10) to compute $N_{j,r}$ and $\hat{N}_{j,r}$ for all $(j, r)$, and then use MPC sampling (Lemma A.11) to send $\mathcal{Z}_{j,r}$ in near-linear time and $O(1)$ rounds. Note that there are $K \cdot \log_2(n^2) = O(k \cdot \log^3 n)$ choices of $(j, r)$, so each $\mathcal{Z}_{j,r}$ can be sent to a distinct machine. In addition, $\mathcal{Z}_{j,r}$ has size at most $T$, so it can be stored in a single machine. Finally, since $\mathcal{Z}_{j,r}$ has size at most $T$, the total time of computing each $\hat{\mathcal{Y}}_{j,r}$ is at most $\tilde{O}(T) \cdot \mathrm{poly}(k, O(1/\gamma)^d, \varepsilon^{-1}, \log \delta^{-1}) = \mathrm{poly}(k, O(1/\gamma)^d, \varepsilon^{-1}, \log \delta^{-1})$, which can be done individually on each machine without any communication.

We only need $O(k \cdot \log^3 n)$ machines to compute the $\hat{\mathcal{Y}}_{j,r}$'s, so the total sequential running time is $\tilde{O}(nkd) + \mathrm{poly}(k, O(1/\gamma)^d, \varepsilon^{-1}, \log \delta^{-1})$ and the total parallel running time is $\tilde{O}(n^\theta kd) + \mathrm{poly}(k, O(1/\gamma)^d, \varepsilon^{-1}, \log \delta^{-1})$. Finally, we only used $O(1)$ total rounds of communication.

**Privacy:** First, the initial construction of $\mathcal{F}$ and the rings is $(\varepsilon, \delta)$-DP with respect to $\mathcal{X}$, by Theorem B.1. Next, note that the construction of $\hat{\mathcal{Y}}_{j,r}$ is $(O(\varepsilon), O(\delta))$-DP with respect to $\mathcal{X}_{j,r}$. This is because computing $\hat{N}_{j,r}$ is $(\varepsilon, 0)$-DP, and conditioned on $\hat{N}_{j,r} \geq 2T$, the applying an $(O(\varepsilon), O(\delta))$-DP algorithm on a sample $T$ points of $\mathcal{X}_{j,r}$ is also $(O(\varepsilon), O(\delta))$-DP, by Lemma A.21. To finish, we note that if we change $\mathcal{X}$ by a single point, assuming $\mathcal{F}$ is fixed, at most two groups $\mathcal{X}_{j,r}$ change (each by at most 1 point). Finally, our construction of $\mathcal{Y}$ only depends on the $\mathcal{Y}_{j,r}$ and $\hat{N}_{j,r}$'s, which were already included in our calculation of privacy. So by adaptive composition, the overall algorithm is $(O(\varepsilon), O(\delta))$-DP.

**Accuracy:** We start by comparing $\mathcal{X}_{j,r}$ to $\mathcal{Z}_{j,r}$, assuming $\hat{N}_{j,r} \geq 2T$. Note that $T \geq \Theta\left(\frac{k \cdot d \cdot \log((\kappa'\gamma')^{-1}) + \log n}{(\kappa')^2 \cdot (\gamma')^2}\right)$, where $\kappa' = \gamma' = c\gamma/d^3$ for some small constant $c$. Therefore, by Lemma A.20 with parameters $\gamma'$ and $\kappa' = \gamma'$, for any set $\mathcal{C} = \{c_1, \ldots, c_k\}$ of $k$ points and for all

$j, r$,

$$\text{cost}(\mathcal{Z}_{j,r}; \mathcal{C}) = (1 \pm \gamma') \cdot \frac{T}{N_{j,r}} \cdot \text{cost}(\mathcal{X}_{j,r}; \mathcal{C}) \pm \gamma' \cdot T \cdot \left( \frac{\Lambda}{n^2} \cdot 2^r \right)^2.$$

(We remark that Lemma A.20 works for any $\mathcal{C}$, even if not contained in $B_{j,r}$ or even in $B(0, \Lambda)$. Now, define $B'_{j,r}$ to be the concentric ball around $B_{j,r}$ of radius $\gamma^{-1} \cdot \frac{\Lambda}{n^2} \cdot 2^r$. By applying Theorem D.1 but replacing $B(0, \Lambda)$ with $B_{j,r}$, we obtain that if $\mathcal{C}$ has nonempty intersection with $B'_{j,r}$, then

$$\text{cost}(\hat{\mathcal{Y}}_{j,r}; \mathcal{C}) = (1 \pm O(\gamma)) \cdot \frac{T}{N_{j,r}} \cdot \text{cost}(\mathcal{X}_{j,r}, \mathcal{C}) \pm O(\gamma') \cdot T \cdot \left( \frac{\Lambda}{n^2} \cdot 2^r \right)^2 \pm V(n, d, k, \gamma, \varepsilon, \delta) \cdot \left( \gamma^{-1} \cdot \frac{\Lambda}{n^2} \cdot 2^r \right)^2.$$

Therefore, assuming $T \geq (\gamma')^{-3} \cdot V(n, d, k, \gamma, \varepsilon, \delta)$, we have that

$$\text{cost}(\hat{\mathcal{Y}}_{j,r}; \mathcal{C}) = (1 \pm O(\gamma)) \cdot \frac{T}{N_{j,r}} \cdot \text{cost}(\mathcal{X}_{j,r}, \mathcal{C}) \pm O(\gamma') \cdot T \cdot \left( \frac{\Lambda}{n^2} \cdot 2^r \right)^2. \qquad (9)$$

Otherwise, we have that every point in $\mathcal{Z}_{j,r}$ and every point in $\hat{\mathcal{Y}}_{j,r}$ have the same distance to $\mathcal{C}$ up to a $1 \pm O(\gamma)$ factor. In addition, the total weight of points in $\mathcal{Z}_{j,r}$ is $T$, but when creating $\hat{\mathcal{Y}}_{j,r}$ using Theorem D.1, we update the vector $v = \{v_{j,r,t}\}$ by adding $\text{Lap}(1/\varepsilon)$ noise to each component. Since there are a total of $O(k \cdot \log n \cdot (1/\gamma)^{O(d)})$ choices for the triple $(j, r, t)$, assuming that $T$ is at least $\frac{\log n}{\varepsilon} \cdot \frac{1}{\gamma}$ times the number of choices, we have that the total weight of $\hat{\mathcal{Y}}_{j,r}$ and $\mathcal{Z}_{j,r}$ are the same up to a $1 \pm O(\gamma)$ factor with overwhelming probability. Therefore, in this case, we still have that (9) holds.

Overall, assuming that $\hat{N}_{j,r} \geq 2T$, we have that after scaling $\hat{\mathcal{Y}}_{j,r}$ by a factor of $\frac{\hat{N}_{j,r}}{T}$, that

$$\frac{\hat{N}_{j,r}}{T} \cdot \text{cost}(\hat{\mathcal{Y}}_{j,r}, \mathcal{C}) = (1 \pm O(\gamma)) \cdot \frac{\hat{N}_{j,r}}{N_{j,r}} \cdot \text{cost}(\mathcal{X}_{j,r}; \mathcal{C}) \pm O(\gamma') \cdot \hat{N}_{j,r} \cdot \left( \frac{\Lambda}{n^2} \cdot 2^r \right)^2$$

$$= (1 \pm O(\gamma)) \cdot \text{cost}(\mathcal{X}_{j,r}; \mathcal{C}) \pm O(\gamma') \cdot N_{j,r} \cdot \left( \frac{\Lambda}{n^2} \cdot 2^r \right)^2,$$

since $\hat{N}_{j,r} = N_{j,r} + \text{Lap}(1/\varepsilon)$ and $N_{j,r}$ are equal up to a $1 \pm \gamma$ factor if $\hat{N}_{j,r} \geq 2T$. Finally, if $\hat{N}_{j,r} \leq 2T$, then we let $\hat{\mathcal{Y}}_{j,r}$ be empty, so it has no cost, but $\mathcal{X}_{j,r}$ has cost at most $O(T \cdot \Lambda^2)$. In addition, there are at most $O(K \cdot R) = O(k \cdot \log^3 n)$ such choices of $(j, r)$.

Adding these over all machines, we have that

$$\text{cost}(\hat{\mathcal{Y}}; \mathcal{C}) = \sum_{(j,r): \hat{N}_{j,r} \geq 2T} \frac{\hat{N}_{j,r}}{T} \cdot \text{cost}(\hat{\mathcal{Y}}_{j,r}; \mathcal{C})$$

$$= (1 \pm O(\gamma)) \cdot \sum_{(j,r): \hat{N}_{j,r} \geq 2T} \text{cost}(\mathcal{X}_{j,r}; \mathcal{C}) \pm O(\gamma') \cdot \sum_{j,r} N_{j,r} \cdot \left( \frac{\Lambda}{n^2} \cdot 2^r \right)^2$$

$$= (1 \pm O(\gamma)) \cdot \text{cost}(\mathcal{X}; \mathcal{C}) \pm O(k \log^3 n \cdot T \cdot \Lambda^2) \pm O(\gamma') \cdot \sum_{x \in \mathcal{X}} d(x, \mathcal{F})^2$$

$$= (1 \pm O(\gamma)) \cdot \text{cost}(\mathcal{X}; \mathcal{C}) \pm O(k \log^3 n \cdot T \cdot \Lambda^2) \pm O(\gamma') \cdot \left[ O(d^3) \cdot \text{OPT}(\mathcal{X}) + k \cdot \text{poly}(\log n, d, \varepsilon^{-1}) \cdot \Lambda^2 \right].$$

Finally, using our bounds for $T$, we have that

$$\text{cost}(\hat{\mathcal{Y}}; \mathcal{C}) = (1 \pm O(\gamma)) \cdot \text{cost}(\mathcal{X}; \mathcal{C}) \pm \text{poly}\left( k, \log n, \varepsilon^{-1}, \log \delta^{-1}, (1/\gamma)^{O(d)} \right) \cdot \Lambda^2.$$

We can apply Theorem D.2 to obtain the following theorem.

**Theorem D.3.** *Suppose that there exists a polynomial-time algorithm that can compute a $\rho$-approximation to $k$-means (resp., $k$-median). Then, for any constant $\rho' > \rho$, exists an $(\varepsilon, \delta)$-DP algorithm for $k$-means (resp., $k$-median) with multiplicative ratio $\rho'$ and additive error $\text{poly}\left( k, e^d, \log n, \varepsilon^{-1}, \log \delta^{-1} \right)$. In addition, assuming each machine can store some $\tilde{O}(n^\theta) \geq \text{poly}(k, e^d, \varepsilon^{-1}, \log \delta^{-1})$ points, the algorithm is implementable in MPC with $O(1)$ total rounds of communication and $O(n^\theta k d)$ time per machine.*

*Proof.* We set $\gamma$ so that $\rho'/\rho = 1 + O(\gamma)$. Given Theorem D.2, the algorithm is quite simple. First, we apply Theorem D.2 to find a weighted coreset $\hat{\mathcal{Y}}$ with at most $\text{poly}(k, (1/\gamma)^{O(d)}, \log n)$ distinct points. Then, move $\hat{\mathcal{Y}}$ to a single machine and then apply a non-private algorithm which can be implemented in $\text{poly}(|\hat{\mathcal{Y}}|)$ time.

The runtime and privacy are straightfoward to check, where the additional $\text{poly}(k, (1/\gamma)^d, \varepsilon^{-1}, \log \delta^{-1})$ time needed is at most $\tilde{O}(n^\theta d)$ by our assumption, and since $\hat{\mathcal{Y}}$ is already private so any output that only depends on $\hat{\mathcal{Y}}$ must also be private. Finally, accuracy is simple to verify, since any $\rho$-approximate $k$-means (or $k$-median) clustering centers for $\hat{\mathcal{Y}}$ must be a $\rho(1 + O(\gamma)) = \rho'$-approximate clustering with additive error $\text{poly}(k, (1/\gamma)^d, \log n, \varepsilon^{-1}, \log \delta^{-1})$. Since $\rho'$ is a fixed constant, this means $\gamma > 0$ is a fixed constant, which completes the proof. $\quad\square$

To finish, we provide algorithm pseudocode for Theorem D.3, as Algorithm 5.

---

**Algorithm 5** A parallel approximation algorithm for differentially private $k$-means (or $k$-median) with arbitrarily good approximation ratio.

---

1: **procedure** ARBITRARILYGOODAPPROX($\mathcal{X}, \varepsilon, \delta, \gamma$) $\qquad\qquad\qquad$ $\triangleright$ Will be $(O(\varepsilon), O(\delta))$-DP.
2: $\qquad$ Let $T = \text{poly}(k, (1/\gamma)^{O(d)}, \log n, \varepsilon^{-1}, \log \delta^{-1})$ be sufficiently large.
3: $\qquad$ Use Algorithm 3 to find private bicriteria approximation $\mathcal{F} = \{f_1, \ldots, f_K\}$.
4: $\qquad$ **for** all $x_i \in \mathcal{X}$ **do**
5: $\qquad\qquad$ Assign $x_i$ to $(j, r)$ if $f_j$ is $x_i$'s closest center in $\mathcal{F}$, $d(x_i, f_j) \approx \frac{\Lambda}{n^2} \cdot 2^r$.
6: $\qquad$ **for** each $j \le K, r \le \log_2(n^2)$ **do**
7: $\qquad\qquad$ Let $B_{j,r}$ be the ball of radius $\frac{\Lambda}{n^2} \cdot 2^r$ around $g_j$.
8: $\qquad\qquad$ Let $\mathcal{X}_{j,r}$ be the set of points $x_i$ assigned to $(j, r)$, and $N_{j,r}$ be the number of points $x_i$ assigned to $(j, r)$.
9: $\qquad\qquad$ Let $\hat{N}_{j,r} = N_{j,r} + \text{Lap}(1/\varepsilon)$.
10: $\qquad\qquad$ Let $\mathcal{Z}_{j,r}$ be a uniform sample $T$ points from $\mathcal{X}_{j,r}$.
11: $\qquad\qquad$ Send each $\mathcal{Z}_{j,r}$ to a specific machine.
12: $\qquad\qquad$ Use Algorithm 4 on $\mathcal{Z}_{j,r}$, which returns $\hat{\mathcal{Y}}_{j,r}$.
13: $\qquad$ Let $\hat{\mathcal{Y}} = \bigcup_{j,r} \hat{\mathcal{Y}}_{j,r}$, where each $\hat{\mathcal{Y}}_{j,r}$ is weighted by $\frac{\hat{N}_{j,r}}{T}$. Move $\hat{\mathcal{Y}}$ to a single machine.
14: $\qquad$ Apply the best $k$-means/$k$-median algorithm of [25], and return the final set of $k$ centers.

---

# E   Dimensionality Reduction

In this section, we show how to reduce the dependencies on the dimension $d$ by showing a reduction to $d$ being roughly $\log k$.

## E.1   Coordinate-Wise Median

A major piece of our dimensionality reduction procedure is to show that one can compute a private coordinate-wise median, and that this coordinate-wise median serves as a good proxy for both the mean and geometric median. First, we note the standard guarantees of private median (in the sequential setting for one dimension). The following result immediately follows from applying the PrivateQuantile algorithm of [68], after rounding each point to the nearest multiple of $\Lambda/n^2$.

**Lemma E.1.** *Given $T$ numbers $z_1, \ldots, z_T \in [-\Lambda, \Lambda] \subset \mathbb{R}$, some $\tau < \frac{1}{T}$, and some choice of quantile $\alpha \in [0.1, 0.9]$, there exists an $(\varepsilon, 0)$-DP algorithm on $z_1, \ldots, z_T$ that can successfully output a point $\tilde{z}$ that is an "approximate" $\alpha$-quantile of $z_1, \ldots, z_T$, with probability at least $1 - \tau$. By this, we mean that the number of these points that are below $z^* - \tau \cdot \Lambda$ is at most $\alpha \cdot T + C\varepsilon^{-1} \log \tau^{-1}$ for some sufficiently large constant $C$, and likewise the number of these points that are above $z^* + \tau \cdot \Lambda$ is at most $(1 - \alpha) \cdot T + C\varepsilon^{-1} \log \tau^{-1}$. In addition, the algorithm can be implemented in $O(T) \cdot \text{poly}(\log \tau^{-1})$ time and space.*

We will also need the following lemma, showing that an approximate coordinate-wise median is a good estimate for a dataset of points.

**Lemma E.2.** *[59] Let $\mathcal{Z}$ be a set of points in $\mathbb{R}^d$, and let $z$ be any point in $\mathcal{Z}$ such that for all $d$ coordinate directions $j \in [d]$, the $j$th coordinate $z_j$ of $z$ is between the $35\%$ and $65\%$ percentile of the $j$th coordinate of the points in $\mathcal{Z}$. Then, for any ball $B$ of any radius $R > 0$, if $B$ contains at least $9/10$ of the points in $\mathcal{Z}$, then the distance between $z$ and the center of $B$ is at most $O(R)$.*

We remark that the original theorem statement in [59] assumed $z$ was exactly the coordinate-wise median, but an identical analysis also implies this stronger version above.

We now show how to compute a private coordinate-wise median in the MPC setting, and establish an important property of this point which will later be crucial in showing it is a good proxy for both the mean and geometric median.

**Lemma E.3.** *Let $n \geq 1$, and let $\mathcal{Z}$ be some set of $T \geq 20C \log(nd) \cdot \varepsilon^{-1} \sqrt{d \log \delta^{-1}}$ points in $B(0, \Lambda) \subset \mathbb{R}^d$, where $C$ is the same constant as in Lemma E.1. Assume we can fit $T$ points into a machine. Then, there exists an $(\varepsilon, \delta)$-DP algorithm that returns a point $\tilde{z}$ with the following property. For any positive $R$ and any unknown ball of radius $R$ around some point $y \in B(0, \Lambda)$ that contains at least $9/10$ of the points in $\mathcal{Z}$, the distance between $\tilde{z}$ and $y$ is at most $CR + \Lambda/n^2$.*

*In addition, the computation can be done in near-linear time with $O(1)$ rounds in MPC.*

*Proof.* Our algorithm is as follows. Define $\varepsilon' = \varepsilon/(2\sqrt{d \log \delta^{-1}})$. Now, we sample $T = 20C \log(nd) \cdot \cdot \varepsilon^{-1} \sqrt{d \log \delta^{-1}} = 10C \log(nd)/\varepsilon'$ points at random from $z_1, \ldots, z_T$, which can be sent to a single machine in $O(1)$ rounds. Next, among these selected points, we compute the private median in each coordinate with failure probability $\tau = \frac{1}{\text{poly}(n,d)}$, to output a point $\tilde{z}$. It is clear that the overall algorithm takes near-linear time, as we can fit $T$ points on a machine.

Among the randomly sampled points, the algorithm is a composition of $d$ $\varepsilon'$-DP algorithms (one in each direction), so the overall algorithm is $(\varepsilon, \delta)$-DP.

To verify accuracy, first note that for any fixed direction, assuming $C$ is sufficiently large, with probability at least $1 - \frac{1}{\text{poly}(n,d)}$, the $40\%$ and $60\%$ percentiles of the sampled points are at least the $35\%$ and at most the $65\%$ percentiles of the true points in that direction. Therefore, we can apply Lemma E.1 with $\varepsilon$ replaced with $\varepsilon'$. With probability at least $1 - \tau = 1 - \frac{1}{\text{poly}(n,d)}$, the private median we find in the fixed direction is between the $40\%$ and $60\%$ percentiles of the sampled points up to error $\Lambda/(n^2 d)$, which is between the $35\%$ and at most the $65\%$ percentiles of the true points up to error $\Lambda/(n^2 d)$. We can take a union bound over each coordinate to say this happens for all coordinates with probability at least $1 - \frac{1}{\text{poly}(n)}$.

For now, let us ignore the error of $\Lambda/(n^2 d)$ per coordinate. Then, we can use Lemma E.2, which implies that if at least $9/10$ of the points are in a ball of radius $R$, then $\tilde{z}$ is in a ball of radius at most $O(R)$ from the center. Thus, by adding back this error per coordinate, the final point $\tilde{z}$ we select is still within $O(R) + \Lambda/n^2$ of the center of any such ball of radius $R$ containing at least $9/10$ of the points in $\mathcal{X}$. In addition, our algorithm did not require $R$, so this holds for all $R$ simultaneously. $\square$

### E.2 Obtaining a Constant Approximation

Here, we show how to convert an approximation algorithm in $d' = O(\log k)$ to an approximation algorithm in $d$ dimensions without blowing up the approximation factor significantly.

**Theorem E.4.** *There exists an $(\varepsilon, \delta)$-DP algorithm for $k$-means (or $k$-median) clustering with multiplicative error $O(1)$ and additive error $(k^{2.5} + k^{1.01}\sqrt{d}) \cdot \text{poly}\left(\log n, \varepsilon^{-1}, \log \delta^{-1}\right)$. In addition, assuming $n^\theta \geq (k^{1.5} + d^{0.5}) \cdot \text{poly}(\log n, \varepsilon^{-1}, \log \delta^{-1})$, the algorithm can be implemented in MPC with $O(1)$ total rounds of communication, total sequential running time $\tilde{O}(nd) + \text{poly}(k, \varepsilon^{-1}, \log \delta^{-1})$, and total time per machine $\tilde{O}(n^\theta d) + \text{poly}(k, \varepsilon^{-1}, \log \delta^{-1})$.*

**Algorithm:** First, by using a random projection $\Pi \in \mathbb{R}^{d' \times d}$, map each point $x_i \in \mathcal{X}$ to $\Pi x_i \in \mathbb{R}^{d'}$. Next, we privately solve $k$-means (or $k$-median) in the lower-dimensional space, using Theorem C.3, to generate a set $\mathcal{C}' = \{c_1', \ldots, c_k'\} \in \mathbb{R}^{d'}$.

Let $S \subset [n]$ be a random sample of points where each point in $[n]$ is in $S$ independently with some probability $p = k^{-\eta}$, for some small constant $\eta > 0$. Next, we construct a $K = \sqrt{1/\eta}$-approximate

nearest neighbor (ANN) data structure for $\mathcal{C}' \in \mathbb{R}^{d'}$. Define $\mathcal{X}_j \subset \mathcal{X}$ as the set of points $x_i$ such that $\Pi x_i$ is mapped to $c'_j \in \mathcal{C}$, and $S_j \subset S$ to be the set of corresponding indices.

Then, for all $1 \le j \le k$, we compute $\hat{N}_j := |S_j| + \text{Lap}(1/\varepsilon)$. If $\hat{N}_j \ge 2T$, where $T := 20C \log(nd) \cdot \varepsilon^{-1} \sqrt{d \log \delta^{-1}}$, we sample $T$ points in each such $\mathcal{X}_j$, and apply Lemma E.3 to find a point $c_j \in \mathbb{R}^d$, which is the point $\tilde{z}$ created from Lemma E.3. Otherwise, we just let $c_j$ be the origin. Our final set is $\mathcal{C} = \{c_1, \ldots, c_k\}$.

**Runtime:** First, we note that $\Pi$ can be generated in a single machine and broadcast to all machines, so all points $\Pi x_i$ can be computed in near-linear time and $O(1)$ rounds. We can then use the runtime guarantees of Theorem C.3 with $d$ replaced by $d' = O(\log k)$.

Next, sampling $S$ is easy (as we just sample each $x_i$ independently). Since $k^{1+\eta} \cdot d$ space (equivalent to $k^{1+\eta}$ points) fits in a machine, we can send $\mathcal{C}'$ to a single machine and use the $K = (1/\sqrt{\eta})$-approximate nearest neighbor data structure from Theorem A.14. This ANN structure uses $O(k^{1+\eta} \cdot d \cdot \log n)$ space and can be broadcast to all machines in $O(1)$ rounds, and then it takes $O(k^\eta \cdot d \cdot \log n)$ time per machine. As we sampled each point to be in $S$ with probability $k^{-\eta}$, with very high probability we do not use more than $\tilde{O}(n^\theta \cdot d)$ time per machine, or $\tilde{O}(n \cdot d)$ time total.

Finally, we can use MPC aggregation (Lemma A.10) to compute $\hat{N}_j$, and use MPC sampling (Lemma A.11) to sample $T$ points from each $\mathcal{X}_j$, in near-linear time and $O(1)$ rounds. Finally, we can store each of the (up to) $k$ samples of $T$ points on a separate machine, and compute each $c_j$ in $\tilde{O}(T \cdot d) \le \tilde{O}(n^\theta \cdot d)$ time. So, the total sequential running time is $\tilde{O}(nd) + \text{poly}(k, \varepsilon^{-1}, \log \delta^{-1})$, and the total parallel running time is $\tilde{O}(n^\theta d) + \text{poly}(k, \varepsilon^{-1}, \log \delta^{-1})$.

**Privacy:** First, the construction of $\mathcal{C}'$ is $(O(\varepsilon), O(\delta))$-DP, since $\Pi$ is oblivious to the dataset $\mathcal{X}$. Next, the sensitivity of the vector $(|S_1|, \ldots, |S_k|)$ if a single point changes is at most 2, so determining $\hat{N}_j = |S_j| + \text{Lap}(1/\varepsilon)$ for all $j$ is $(2\varepsilon, 0)$-DP. Finally, assuming that $\mathcal{C}'$ and the $\hat{N}_j$'s are fixed, constructing $c_j$ for all sets is $(2\varepsilon, 2\delta)$-DP, since changing a single point in $\mathcal{X}$ can change at most two sets $\mathcal{X}_j$ by at most 1 point each. So, by adaptive composition, the overall procedure is $(O(\varepsilon), O(\delta))$-DP.

**Accuracy:** Define $\text{OPT}(\mathcal{X})$ as the optimum cost of $\mathcal{X}$ and $\text{OPT}(\Pi\mathcal{X})$ as the optimum cost of $\Pi\mathcal{X}$ (either for $k$-means or $k$-median). By Theorem A.8, we have that if $d' \ge O(\log k)$, $0.5 \cdot \text{OPT}(\mathcal{X}) \le \text{OPT}(\Pi\mathcal{X}) \le 2\text{OPT}(\mathcal{X})$. In addition, with overwhelming probability, no point in $\Pi\mathcal{X}$ has norm greater than $O(\sqrt{\log n}) \cdot \Lambda$: this would be true even for a random projection down to a single direction. Hence, $\text{cost}(\Pi\mathcal{X}, \mathcal{C}') \le O(\text{OPT}(\mathcal{X})) + O(\log n) \cdot U(n, d', k, \varepsilon, \delta) \cdot \Lambda^2$, where $U(n, d', k, \varepsilon, \delta)$ represents the additive error from applying Theorem C.3 in $d' = O(\log k)$ dimensions.

For each $j \in [k]$, let $\mathcal{X}'_j \subset \mathcal{X}$ be the full set of points $x_i$ such that $\Pi x_i$ would have been mapped to $c'_j$ if we did not sample $S$ from $[n]$, and define $S'_j \subset [n]$ to be the corresponding set of indices. So, $S'_j$ partitions $[n]$ instead of $S$. Hence, if we applied the full ANN data structure to every point in $\Pi\mathcal{X}$, the cost obtained, treating $K = \sqrt{1/\eta}$ as a constant, is still at most $O(\text{OPT}(\mathcal{X})) + O(\log n) \cdot U(n, d', k, \varepsilon, \delta) \cdot \Lambda^2$. In other words, $\sum_{j \le k} \sum_{i \in S'_j} d(\Pi x_i, c'_j)^2 \le O(\text{OPT}(\mathcal{X})) + O(\log n) \cdot U(n, d', k, \varepsilon, \delta) \cdot \Lambda^2$. (The same applies for the $k$-median case, replacing $d(\Pi x_i, c'_j)^2$ with $d(\Pi x_i, c'_j)$ and $\Lambda^2$ with $\Lambda$.) By applying Theorem A.8 again, if we pick the true mean (or geometric median) of each cluster $\mathcal{X}'_j$ as $\tilde{c}_j \in \mathbb{R}^d$, then $\sum_{j \le k} \sum_{i \in S'_j} d(x_i, \tilde{c}_j) \le O(\text{OPT}(\mathcal{X})) + O(\log n) \cdot U(n, d', k, \varepsilon, \delta) \cdot \Lambda^2$.

If $|S'_j| \le \Theta(T \cdot k^\eta)$, we may run into trouble with $|S_j| + \text{Lap}(1/\varepsilon) \le 2T$, in which case the additive cost of the points in $S'_j$ may be very large. There could be up to $\Theta(k)$ such bad choices of $j$, which gives us an additive cost of up to $\Theta(T \cdot k^{1+\eta}) \cdot \Lambda^2$. Otherwise, we will have that $|S_j| \ge 3T$, which means $|S_j| + \text{Lap}(1/\varepsilon) \ge 2T$, so we can sample $T$ points and apply Lemma E.3 to find some $c_j$. Let $R_j$ be the 90% percentile of distances between $\tilde{c}_j$ and the points $x_i \in \mathcal{X}_j$, which by a Chernoff bound is at most the 95% percentile of distances between $\tilde{c}_j$ and the points $x_i \in \mathcal{X}'_j$. By Lemma E.3, $d(\tilde{c}_j, c_j) \le O(R_j + \Lambda/n^2)$. However, we know that the average distance (or squared distance) between $\tilde{c}_j$ and the points $x_i \in \mathcal{X}_j$ is at least $\Omega(R_j)$ (or $\Omega(R_j^2)$), because at least 5% of points in

$\mathcal{X}'_j$ are distance at least $R_j$ from $\tilde{c}_j$. Thus, choosing $c_j$ instead of $\tilde{c}_j$ cannot blow up the cost of its respective cluster by more than an $O(1)$ multiplicative factor and more than an $O(\Lambda)$ additive factor.

If we had used the centers $\{\tilde{c}_j\}$, we would have obtained a cost of $O(\text{OPT}(\mathcal{X})) + O(\log n) \cdot U(n, d', k, \varepsilon, \delta) \cdot \Lambda^2$. So, overall, since we use the centers $\{c_j\}$ instead, we obtain a cost of $O(\text{OPT}(\mathcal{X})) + O(\log n) \cdot U(n, d', k, \varepsilon, \delta) \cdot \Lambda^2 + O(T \cdot k^{1+\eta}) \cdot \Lambda^2$. As $T = O(\log(nd) \cdot \varepsilon^{-1}\sqrt{d \log \delta^{-1}})$, by setting $\eta = 0.01$ and applying the bound for $U$ from Theorem C.3, we obtain an additive error of

$$(k^{2.5} + k^{1.01}\sqrt{d}) \cdot \text{poly}(\log n, \log d, \varepsilon^{-1}, \log \delta^{-1}).$$

### E.3  Obtaining nearly optimal approximation factor

Here, we show how to convert a $\rho$-approximation algorithm in $d' = O(\gamma^{-2}\log(k/\gamma))$ dimensions to a $\rho \cdot (1 + \gamma)$-approximation algorithm in $d$ dimensions for an arbitrarily small constant $\gamma$.

First, we need the following result about private empirical risk minimization, due to Bassily et al. [9] (and slightly restated).

**Lemma E.5.** *[9] Let $f(\theta, x)$ be a convex function in $\theta$ that is $L$-Lipschitz for some $L$. Suppose we are attempting to minimize $\ell(\theta) := \sum_{i=1}^{N} f(\theta; x_i)$ for a dataset $x_1, \ldots, x_N$, over $\theta$ in a ball $B$ of radius $\Lambda$. Then, there exists an $(\varepsilon, \delta)$-DP algorithm that runs in polynomial time that outputs some $\theta'$ such that $\ell(\theta') - \min_{\theta \in B} \ell(\theta) \leq O_L\left(\frac{\sqrt{d}}{\varepsilon} \cdot \text{poly}\log(\frac{n}{\delta})\right) \cdot \Lambda$ with overwhelming probability.*

Note that the function $f(\theta, x) = \|\theta - x\|_2$ is 1-Lipschitz and convex. This observation will be crucial in applying the above lemma.

**Theorem E.6.** *Suppose that there exists a polynomial-time algorithm that can compute a $\rho$-approximation to $k$-means (resp., $k$-median). Then, for any constant $\rho' > \rho$, exists an $(\varepsilon, \delta)$-DP algorithm for $k$-means (resp., $k$-median) with multiplicative error $\rho'$ and additive error $\text{poly}\left(k, d, \log n, \varepsilon^{-1}, \log \delta^{-1}\right)$. In addition, the algorithm can be implemented in MPC with $O(1)$ total rounds of communication, total sequential time $\tilde{O}(nkd)$, and total time per machine $\tilde{O}(n^\theta kd)$, assuming each machine can store $\tilde{O}(n^\theta) \geq \text{poly}(k, d, \log n, \varepsilon^{-1}, \log \delta^{-1})$ points.*

**Algorithm:**  We will set $\gamma$ such that $\rho' = (1 + O(\gamma)) \cdot \rho$, and $d' = O(\gamma^{-2}\log(k/\gamma))$. Similar to in Theorem E.4, we start with a random projection $\Pi \in \mathbb{R}^{d' \times d}$, map each point $x_i \in X$ to $\Pi x_i \in \mathbb{R}^{d'}$. (Note that $d'$ is slightly larger here than in Subsection E.2). Now, we can privately solve $k$-means (or $k$-median) in $\mathbb{R}^{d'}$, using Theorem D.2, to generate a set $\mathcal{C}' = \{c'_1, \ldots, c'_k\} \in \mathbb{R}^{d'}$.

Now, rather than sampling $S$ and using approximate nearest neighbor, we simply map each point $\Pi x_i$ to its closest point $c'_j \in \mathcal{C}'$, which partitions the dataset $\mathcal{X}$ into $\mathcal{X}_1, \ldots, \mathcal{X}_k$ and the set of indices $[n]$ into $S_1, \ldots, S_k$.

In the case of $k$-means, we will simply compute an $(\varepsilon, \delta)$-differentially private mean for each cluster of points $x_i \in \mathcal{X}_j$ to obtain some $c_j$. More precisely, we define $\hat{N}_j = |S_j| + \text{Lap}(1/\varepsilon)$, and define $c_j = \left(O(\Lambda \cdot \varepsilon^{-1}\log \delta^{-1}) \cdot \mathcal{N}(0, I) + \sum_{x_i \in \mathcal{X}_j} x_i\right)/\hat{N}_j$. If for some reason $\|c_j\|_2 \geq 2\Lambda$, we replace $c_j$ with the origin. Our final output will be $\mathcal{C} = \{c_1, \ldots, c_k\}$.

In the $k$-median case, as in Theorem E.4, we compute some approximate coordinate-wise median per cluster. Specifically, for all $1 \leq j \leq k$, we let $N_j = |S_j|$ and $\hat{N}_j := |S_j| + \text{Lap}(1/\varepsilon)$. If $\hat{N}_j \geq 2T$, where $T := 20C\gamma^{-4}\log \gamma^{-1} \cdot \text{poly}\log(\frac{nd}{\delta}) \cdot \varepsilon^{-1} \cdot d \geq 20C\log(nd) \cdot \varepsilon^{-1} \cdot \sqrt{d \log \delta^{-1}}$, we sample $T$ points $\hat{\mathcal{X}}_j$ in each such $\mathcal{X}_j$ (let $\hat{S}_j$ be the corresponding indices), and apply Lemma E.3 to find a point $\hat{c}_j \in \mathbb{R}^d$, which is the point $\tilde{z}$ created from Lemma E.3. For each such $j$ for which $\hat{N}_j \geq 2T$, we again use Lemma E.1 to compute an $(\varepsilon, 0)$-DP estimate of the 70% percentile distance from $\hat{c}_j$ to the $T$ sampled points $x_i \in \hat{\mathcal{X}}_j$. Let this distance be $\hat{R}_j$. To finish, we set a threshold $\tilde{R}_j := C(\hat{R}_j + \Lambda/n^2)$ for a sufficiently large constant $C$, and compute $c_j$ to be a private minimizer of the loss function $\ell(c) := \sum_{x \in \hat{\mathcal{X}}_j, \|x - \hat{c}_j\|_2 \leq \gamma^{-1} \cdot \tilde{R}_j} d(x, \hat{c}_j)$, which we compute using Lemma E.5.

**Runtime:** As in Theorem E.4, we can compute all points $\Pi x_i$ in near-linear time and $O(1)$ rounds. We then use the runtime guarantees of Theorem D.2 with $d$ replaced by $d' = O(\gamma^{-2}\log(k/\gamma))$, which means $(1/\gamma)^{d'} = k^{\tilde{O}(\gamma^{-2})}$. Next, we can broadcast $\mathcal{C}'$ to all machines, and for each $x \in \mathcal{X}$ compute its nearest neighbor in $\mathcal{C}'$ in time $\tilde{O}(n^\theta \cdot k \cdot d)$ time per machine or $\tilde{O}(n \cdot k \cdot d)$ time total.

In the $k$-means case, we can use MPC Aggregation (Lemma A.10) to compute $\sum_{x_i \in \mathcal{X}_j} x_i$ and $\hat{N}_j$ for all $j \in [k]$, using near-linear time and $O(1)$ rounds. Then, we can compute each $c_j$ in $O(d)$ time. So, after computing $\mathcal{C}'$ and the nearest neighbor of each $x_i$, the rest takes near-linear time and $O(1)$ rounds.

In the $k$-median case, as in Theorem E.4, we can compute $\hat{N}_j$ and sample $T$ points from each $S_j$, in near-linear time and $O(1)$ rounds. Finally, we can store each of the (up to) $k$ groups of $T$ sampled points on a separate machine, and compute each $\hat{c}_j$ in $\tilde{O}(T \cdot d)$ time. Then, we use the private median algorithm to compute $\hat{R}_j$ (and consequently, $\tilde{R}_j$) for each $j$, which also takes $\tilde{O}(T \cdot d)$ time. Finally, we privately minimize the empirical loss of points within $\tilde{R}_j/\gamma$ of $\hat{c}_j$, which takes $\text{poly}(T, d)$ time.

Since $T$ points fit in a single machine, we therefore have the total sequential running time is $\tilde{O}(nkd) + \text{poly}(k^{\tilde{O}(\gamma^{-2})}, d, \varepsilon^{-1}, \log \delta^{-1})$, and the total parallel running time per machine is $\tilde{O}(n^\theta kd) + \text{poly}(k^{\tilde{O}(\gamma^{-2})}, d, \varepsilon^{-1}, \log \delta^{-1})$. Finally, we only use $O(1)$ total rounds of communication.

**Privacy:** First, note that $\mathcal{C}'$ is $(\varepsilon, \delta)$-DP, as in Theorem E.4. Next, note that $\hat{N}_j$ is $(\varepsilon, 0)$-DP with respect to the points in $S_j$.

We now consider privacy for the rest of the algorithm with respect to an individual $\mathcal{X}_j$. In the $k$-means case, if $\mathcal{X}_j$ changes by inserting, removing, or changing one point, this affects $\sum_{x_i \in \mathcal{X}_j} x_i$ by at most $2\Lambda$ in $\ell_2$ distance. So, the Gaussian Mechanism tells us that $c_j$ is $(\varepsilon, \delta)$-DP assuming that $\hat{N}_j$ is fixed. So, by adaptive composition, the algorithm is $(O(\varepsilon), O(\delta))$-DP, since changing $\mathcal{X}$ affects at most 2 of the $\mathcal{X}_j$'s.

In the $k$-median case, we first consider privacy with respect to the sampled dataset $\hat{\mathcal{X}}_j$. The creation of $\hat{c}_j$ is clearly $(O(\varepsilon), O(\delta))$-DP. Also, assuming that $\tilde{c}_j$ is fixed, the value $\hat{R}_j$ (and thus $\tilde{R}_j$) is $(\varepsilon, \delta)$-DP. Finally, assuming $\hat{c}_j, \tilde{R}_j$ is fixed we are using an $(\varepsilon, \delta)$-DP empirical risk minimization algorithm on points within $\gamma^{-1}\tilde{R}_j$ of $\hat{c}_j$: if $\hat{\mathcal{X}}_j$ changes by 1 point then this set of points we perform the private empirical risk minimization algorithm (Lemma E.5) on changes by at most 1 point. Therefore, the remainder of the algorithm for computing $c_j$ is $(O(\varepsilon), O(\delta))$-DP with respect to $\hat{\mathcal{X}}_j$, which means by Lemma A.21, it is also $(O(\varepsilon), O(\delta))$-DP with respect to the points in $\mathcal{X}_j$.

Finally, we have that for the full dataset $\mathcal{X}$, assuming $\mathcal{C}'$ is fixed, changing a single point in $\mathcal{X}$ affects at most two of the datasets $\mathcal{X}_j$, each by at most 1 point. So, by adaptive composition, the overall algorithm is $(O(\varepsilon), O(\delta))$-DP.

**Accuracy:** Let $\text{OPT}(\mathcal{X})$ be the optimum cost of $\mathcal{X}$ and $\text{OPT}(\Pi\mathcal{X})$ be the optimum cost of $\Pi\mathcal{X}$ (either for $k$-means or $k$-median). By Theorem A.8, we have that if $d' \geq O(\gamma^{-2}\log(k/\gamma))$, $(1-\gamma) \cdot \text{OPT}(\mathcal{X}) \leq \text{OPT}(\Pi\mathcal{X}) \leq (1+\gamma) \cdot \text{OPT}(\mathcal{X})$. As in Theorem E.4, no point in $\Pi\mathcal{X}$ has norm greater than $O(\sqrt{\log n}) \cdot \Lambda$, so $\text{cost}(\Pi\mathcal{X}, \mathcal{C}') \leq (1+\gamma) \cdot (\text{OPT}(\Pi\mathcal{X})) + O(\log n) \cdot U(n, d', k, \varepsilon, \delta) \cdot \Lambda^2$, where $U(n, d', k, \varepsilon, \delta)$ represents the additive error from applying Theorem D.3 in $d' = O(\gamma^{-2}\log(k/\gamma))$ dimensions.

Now, since we used the entire dataset (instead of sampling $S$), we have that

$$\sum_{j \leq k} \sum_{i \in S_j} d(\Pi x_i, c_j')^2 \leq \rho \cdot \text{OPT}(\Pi\mathcal{X}) + O(\log n) \cdot U(n, d', k, \varepsilon, \delta) \cdot \Lambda^2$$

$$\leq \rho \cdot (1+\gamma) \cdot \text{OPT}(\mathcal{X}) + O(\log n) \cdot U(n, d', k, \varepsilon, \delta) \cdot \Lambda^2,$$

where the last inequality follows by Theorem A.8. (The same applies for the $k$-median case, replacing $d(\Pi x_i, c_j')^2$ with $d(\Pi x_i, c_j')$ and $\Lambda^2$ with $\Lambda$.) We can then apply Theorem A.8 again to obtain that if

we pick the true mean (or geometric median) of each cluster $\mathcal{X}_j$ as $\tilde{c}_j \in \mathbb{R}^d$, then

$$\sum_{j \leq k} \sum_{i \in S_j} d(x_i, \tilde{c}_j) \leq \rho \cdot (1 + O(\gamma)) \cdot \text{OPT}(\mathcal{X}) + O(\log n) \cdot U(n, d', k, \varepsilon, \delta) \cdot \Lambda^2 \qquad (10)$$

We now focus on the $k$-**means** case and consider how far $c_j$ deviates from $\tilde{c}_j$. Note that $c_j = \frac{1}{\hat{N}_j} \cdot \left( O(\Lambda \cdot \varepsilon^{-1} \log \delta^{-1}) \cdot \mathcal{N}(0, I) + \sum_{x_i \in \mathcal{X}_j} x_i \right)$, whereas $\tilde{c}_j = \frac{1}{N_j} \cdot \sum_{x_i \in \mathcal{X}_j} x_i$. If $N_j \geq \Omega(\varepsilon^{-1} \log n)$, then with overwhelming probability, $|\hat{N}_j - N_j| \leq O(\varepsilon^{-1} \log n) \leq \frac{1}{2} N_j$. Therefore, we have

$$\|c_j - \tilde{c}_j\|_2 \leq \frac{1}{\hat{N}_j} \cdot O(\Lambda \cdot \varepsilon^{-1} \log \delta^{-1}) \cdot O(\sqrt{d \log n}) + \left( \frac{1}{\hat{N}_j} - \frac{1}{N_j} \right) \cdot \left\| \sum_{x_i \in \mathcal{X}_j} x_i \right\|_2$$

$$\leq \Lambda \cdot \frac{O(\varepsilon^{-1} \log \delta^{-1} \cdot \sqrt{d \log n})}{N_j} + \frac{O(\varepsilon^{-1} \log n)}{N_j^2} \cdot \sum_{x_i \in \mathcal{X}_j} \|x_i\|_2$$

$$\leq \Lambda \cdot \frac{O(\varepsilon^{-1} \log \delta^{-1} \cdot \sqrt{d \log n})}{N_j}.$$

Since $\tilde{c}_j$ is the true center of the cluster $\mathcal{X}_j$, this means that $\sum_{i \in S_j} d(c_j, x_i)^2 = N_j \cdot d(\tilde{c}_j, c_j)^2 + \sum_{i \in S_j} d(\tilde{c}_j, x_i)^2 = O(\Lambda^2) \cdot O(\varepsilon^{-2} \log^2 \delta^{-1} \cdot d \cdot \log^2 n) + \sum_{i \in S_j} d(\tilde{c}_j, x_i)^2$. This is all assuming $N_j \geq \Omega(\varepsilon^{-1} \log n)$, but otherwise, we have that since $N_j \leq O(\varepsilon^{-1} \log n)$, the maximum error we can have for $c_j$ (since we have ensured $\|c_j\|_2 \leq O(\Lambda)$) is at most $O(\Lambda^2) \cdot \varepsilon^{-1} \log n$.

Overall, the final cost is

$$\text{cost}(\mathcal{X}; \mathcal{C}) \leq \rho \cdot (1 + O(\gamma)) \cdot \text{OPT}(\mathcal{X}) + \left[ O(\log n) \cdot U(n, d', k, \varepsilon, \delta) + O(k \cdot \varepsilon^{-2} \log^2 \delta^{-1} \cdot d \cdot \log^2 n) \right] \cdot \Lambda^2.$$

Applying $d' = O(\gamma^{-2} \log(k/\gamma))$, we get the desired accuracy guarantees.

Next, we focus on the $k$-**median** case, and consider some cluster $j$ such that $\hat{N}_j \geq 3T$. Define $R_j$ to be the smallest real number such that at least 95% of the points in $\mathcal{X}_j$ are within $R_j$ of the true geometric median $\tilde{c}_j$ of $\mathcal{X}_j$. We claim the following.

**Proposition E.7.** *With overwhelming probability, $\hat{R}_j \leq O(R_j + \Lambda/n^2)$.*

*Proof.* By a Chernoff bound, at least 90% of the $T$ sampled points are within $R_j$ of $\tilde{c}_j$ with overwhelming probability. Therefore, $d(\tilde{c}_j, \hat{c}_j) \leq O(R_j + \Lambda/n^2)$ by Lemma E.3, which we may apply because $T \geq 20C \log(nd) \cdot \varepsilon^{-1} \cdot \sqrt{d \log \delta^{-1}}$. This also implies that at least 90% of the sampled points are within $O(R_j + \Lambda/n^2)$ of $\hat{c}_j$, so with overwhelming probability, since $\hat{R}_j$ is a private estimator of the 70% percentile distance from $\hat{c}_j$ to the sampled points, we have that $\hat{R}_j \leq O(R_j + \Lambda/n^2)$. $\quad\square$

**Proposition E.8.** *With overwhelming probability, $d(\hat{c}_j, \tilde{c}_j) \leq O(\hat{R}_j + \Lambda/n^2)$.*

*Proof.* Note that with overwhelming probability, $\hat{R}_j$ is at least the 65% percentile of distances between $\hat{c}_j$ and the $T$ sampled points, up to error $O(\Lambda/n^2)$, which means by a Chernoff bound, it is at least the 60% percentile of distances between $\hat{c}_j$ and points in $\mathcal{X}_j$, up to error $O(\Lambda/n^2)$.

Let $Q_j$ be the true 60% percentile of distances between $\hat{c}_j$ and points in $\mathcal{X}_j$. It suffices to show that $\hat{Q}_j := d(\hat{c}_j, \tilde{c}_j) \leq O(Q_j)$. To see why, any point $x$ within $Q_j$ of $\hat{c}_j$, we have that $d(x, \tilde{c}_j) \geq d(\tilde{c}_j, \hat{c}_j) - d(x, \hat{c}_j) \geq \hat{Q}_j - Q_j$. So, $d(x, \tilde{c}_j) - d(x, \hat{c}_j) \geq \hat{Q}_j - 2Q_j$. So, this holds for at least 60% of points. However, for the remaining 40% of points, we still have that $d(x, \tilde{c}_j) - d(x, \hat{c}_j) \geq -d(\tilde{c}_j, \hat{c}_j) \geq -\hat{Q}_j$. Now, adding $d(x, \tilde{c}_j) - d(x, \hat{c}_j)$ across all $x \in \mathcal{X}_j$ should be negative, because $\tilde{c}_j$ is the true geometric median of $\mathcal{X}_j$. So, $0.6 \cdot (\hat{Q}_j - 2Q_j) + 0.4 \cdot (-\hat{Q}_j) \leq 0$, which means that $\hat{Q}_j \leq 6Q_j$ by rearranging. $\quad\square$

Proposition E.8 implies that $d(\hat{c}_j, \tilde{c}_j) \leq \tilde{R}_j$, based on how we chose $\tilde{R}_j$.

Now, consider any point $c$ in the ball $B(\hat{c}_j, \tilde{R}_j)$ of radius $\tilde{R}_j$ around $\hat{c}_j$, and define $\ell(c) := \sum_{x \in \mathcal{X}_j \cap B(\hat{c}_j, \gamma^{-1}\tilde{R}_j)} d(c, x)$, or equivalently, $\ell(c) = \sum_{x \in \mathcal{X}_j} d(c, x) \cdot \mathbb{I}(d(x, \hat{c}_j) \leq \gamma^{-1} \cdot \tilde{R}_j)$. Define $\ell'(c) := \sum_{x \in \hat{\mathcal{X}}_j} d(c, x) \cdot \mathbb{I}(d(x, \hat{c}_j) \leq \tilde{R}_j)$. Note that our private empirical risk minimization algorithm attempts to minimize $\ell'(c)$, since we are restricting ourselves to loss from the sampled points in $B(\hat{c}_j, \gamma^{-1} \cdot \tilde{R}_j)$. Note that for $c \in B(\hat{c}_j, \tilde{R}_j)$, each summand $d(c, x) \cdot \mathbb{I}(d(x, \hat{c}_j) \leq \gamma^{-1} \cdot \tilde{R}_j)$ in $\ell(c)$ is bounded by $O(\gamma^{-1} \cdot \tilde{R}_j)$. Therefore, we can apply Hoeffding's inequality to say for any fixed $c$, with probability at least $2 \cdot \exp\left(-\gamma^4 \cdot T\right)$, $\ell'(c) = \frac{T}{N_j} \cdot \ell(c) \pm \gamma^2 \cdot T \cdot \tilde{R}_j$. In addition, note that there exists a $\gamma^2$-cover of $B(\hat{c}_j, \tilde{R}_j)$ of size $(1/\gamma)^{O(d)}$, which means that by our choice of $T$, we have that with overwhelming probability, $\ell'(c) = \frac{T}{N_j} \cdot \ell(c) \pm \gamma^2 \cdot T \cdot \tilde{R}_j$ holds for all $c$ in this cover. Now, for all $c \in B(\hat{c}_j, \tilde{R}_j)$, we can round $c$ to a point in the cover of distance at most $\gamma^2 \cdot \tilde{R}_j$ away, affects $\ell(c)$ by at most $N_j \cdot \tilde{R}_j \cdot \gamma^2$ and $\ell'(c)$ by at most $T \cdot \tilde{R}_j \cdot \gamma^2$. So, we have that with overwhelming probability, for all $c \in B(\hat{c}_j, \tilde{R}_j)$, that $\ell'(c) = \frac{T}{N_j} \cdot \ell(c) \pm O(\gamma^2 \cdot T) \cdot \tilde{R}_j$.

Now, by applying Lemma E.5, we return a point $c_j \in B(\hat{c}_j, \tilde{R}_j)$ such that $\ell'(c_j) \leq \min_{c \in B(\hat{c}_j, \gamma^{-1}\tilde{R}_j)} \ell'(c) + O\left(\frac{\sqrt{d}}{\varepsilon} \cdot \operatorname{poly}\log(\frac{n}{\delta})\right) \cdot \gamma^{-1}\tilde{R}_j \leq \ell'(\tilde{c}_j) + O\left(\frac{\sqrt{d}}{\varepsilon} \cdot \operatorname{poly}\log(\frac{n}{\delta})\right) \cdot \tilde{R}_j$, since we think of $\gamma$ as a fixed constant. Now, using our bounds comparing $\ell(c)$ with $\ell'(c)$, we have that $\ell(c_j) \leq \ell(\tilde{c}_j) + O(\gamma^2 \cdot N_j) \cdot \tilde{R}_j + \frac{N_j}{T} \cdot O\left(\frac{\sqrt{d}}{\varepsilon} \cdot \operatorname{poly}\log(\frac{n}{\delta})\right) \cdot \tilde{R}_j$. Now, based on our definition of $T$ and Proposition E.7, we can further simplify this as $\ell(c_j) \leq \ell(\tilde{c}_j) + O(\gamma^2 \cdot N_j) \cdot \tilde{R}_j \leq \ell(\tilde{c}_j) + O(\gamma^2) \cdot (R_j + \Lambda/n^2) \cdot N_j$. Finally, we note that $\ell(c)$ does not deal with points outside the ball $B(\hat{c}_j, \gamma^{-1} \cdot \tilde{R}_j)$. However, for any such point, since $d(c_j, \tilde{c}_j) \leq d(c_j, \hat{c}_j) + d(\hat{c}_j, \tilde{c}_j) \leq O(\tilde{R}_j)$, we have that $d(c_j, x) = (1 \pm O(\gamma)) \cdot d(\tilde{c}_j, x)$. Therefore, with overwhelming probability, we have that

$$\sum_{x \in \mathcal{X}_j} d(c_j, x) \leq (1 + O(\gamma)) \cdot \sum_{x \in \mathcal{X}_j} d(\tilde{c}_j, x) + O(\gamma^2) \cdot \left(R_j + \frac{\Lambda}{n^2}\right) \cdot N_j.$$

By Markov's inequality, $R_j$ is at most $O(1)$ times the average distance between $\tilde{c}_j$ and the points in $\mathcal{X}_j$, which implies that

$$\sum_{x \in \mathcal{X}_j} d(c_j, x) \leq (1 + O(\gamma)) \cdot \sum_{x \in \mathcal{X}_j} d(\tilde{c}_j, x) + O(\gamma^2) \cdot \left(\frac{\Lambda}{n^2}\right) \cdot N_j.$$

By adding the above equation over all $j \in [k]$ and combining with Equation (10), we obtain

$$\begin{aligned}
\operatorname{cost}(\mathcal{X}; \mathcal{C}) &\leq \sum_{j=1}^{k} \sum_{x \in \mathcal{X}_j} d(c_j, x) \\
&\leq \sum_{j=1}^{k} (1 + O(\gamma)) \cdot \sum_{x \in \mathcal{X}_j} d(\tilde{c}_j, x) + O(\gamma^2) \cdot N_j \cdot \frac{\Lambda}{n^2} \\
&\leq \rho \cdot (1 + O(\gamma)) \cdot \operatorname{OPT}(\mathcal{X}) + O(\log n) \cdot U(n, d', k, \varepsilon, \delta) \cdot \Lambda^2 + O(\gamma^2) \cdot N_j \cdot \frac{\Lambda}{n^2} \\
&= \rho \cdot (1 + O(\gamma)) \cdot \operatorname{OPT}(\mathcal{X}) + O(\log n) \cdot U(n, d', k, \varepsilon, \delta) \cdot \Lambda^2.
\end{aligned}$$

This concludes the proof of accuracy, by setting $d' = O(\log(k/\gamma)/\gamma^2)$.

To finish, we provide pseudocode for Theorem E.6, as Algorithm 6.

**Algorithm 6** A parallel approximation algorithm for differentially private $k$-means (or $k$-median) with arbitrarily good approximation ratio.

---

1: **procedure** ARBITRARILYGOODAPPROXHIGHDIM($\mathcal{X}, \varepsilon, \delta, \gamma$)  ▷ Will be $(O(\varepsilon), O(\delta))$-DP.

2:   Let $T = \Theta(\gamma^{-4} \log \gamma^{-1} \cdot \text{poly} \log(\frac{nd}{\delta}) \cdot \varepsilon^{-1} \cdot d)$.

3:   Let $d' = O(\gamma^{-2} \log(k/\gamma))$, and pick a random projection $\Pi \in \mathbb{R}^{d' \times d}$.

4:   Use Algorithm 1 to find a $k$-means clustering $\mathcal{C}' = \{c'_1, \ldots, c'_k\}$ of $\Pi\mathcal{X} = \{\Pi x_1, \ldots, \Pi x_n\}$.

5:   Map each point $\Pi x_i : i \in S$ to its nearest neighbor in $\mathcal{C}'$.

6:   **for** $j = 1$ to $k$ **do**

7:     Let $\mathcal{X}_j$ be the set of points $x_i$ such that $\Pi x_i$ is mapped to $c'_j$.

8:     Let $\hat{N}_j = |\mathcal{X}_j| + \text{Lap}(1/\varepsilon)$.

9:     **if** $k$-means **then**

10:       $c_j = \left( O(\Lambda \cdot \varepsilon^{-1} \log \delta^{-1}) \cdot \mathcal{N}(0, I) + \sum_{x_i \in \mathcal{X}_j} x_i \right) / \hat{N}_j$

11:       Replace $c_j$ with 0 if $\|c_j\|_2 \geq 2\Lambda$.

12:     **else**

13:       **if** $\hat{N}_j \geq 2T$ **then**

14:         Sample $T$ points $\hat{\mathcal{X}}_j$ from $\mathcal{X}_j$.

15:         Apply a private coordinate-wise median of $\hat{\mathcal{X}}_j$ to obtain $\hat{c}_j$.

16:         Compute $\hat{R}_j$, a private estimate of the 70% percentile of distances from $\hat{c}_j$ to points in $\hat{\mathcal{X}}_j$.

17:         Set $\tilde{R}_j = \Theta(\hat{R}_j + \Lambda/n^2)$.

18:         Use Lemma E.5 with loss $\ell(c) = \sum_{x \in \hat{\mathcal{X}}_j, \|x - \hat{c}_j\|_2 \leq \gamma^{-1} \cdot \tilde{R}_j} d(x, \hat{c}_j)$ to obtain a private empirical risk minimizer $c_j$.

19:   Return $\mathcal{C} = \{c_1, \ldots, c_k\}$

---

## E.4 A Fully Near-Linear time Algorithm

We finish by improving Theorem E.6 to running in $\tilde{O}(nd)$ sequential time (and $\tilde{O}(n^\theta d)$ parallel time), removing the runtime dependence on $k$.

Consider $p \in \{1, 2\}$ ($p = 1$ corresponds to $k$-median; $p = 2$ corresponds to $k$-means). Given $\mathcal{X}$ and a set of $k$ centers $\mathcal{C}$, let $a_i := d(x_i, \mathcal{C})^p$. Now, consider sampling each $i \in [n]$ with probability $1/k$, to get a subsampled set $S$. We wish to compare the cost of $k \cdot \sum_{i \in S} d(x_i, \mathcal{C})^p$ with $\text{cost}(\mathcal{X}; \mathcal{C}) = \sum_{i=1}^n d(x_i, \mathcal{C})^p$. If we let $a_i'$ be independent random variables, where $a_i' = (k-1)a_i$ with probability $\frac{1}{k}$ and $-a_i$ otherwise, then $\sum a_i'$ has the same distribution as $k \cdot \text{cost}(S; \mathcal{C}) - \text{cost}(\mathcal{X}; \mathcal{C})$, where $S$ is the subsampled set of $\mathcal{X}$. Note that each $a_i'$ has mean 0, variance $O(k \cdot a_i^2)$, and is uniformly bounded by $O(\Lambda^p)$. Therefore, by Bernstein's inequality, we have that

$$\mathbb{P}\left(|k \cdot \text{cost}(S; \mathcal{C}) - \text{cost}(\mathcal{X}; \mathcal{C})| \geq t\right) \leq 2 \exp\left(-\frac{\Omega(t^2)}{k \cdot \sum a_i^2 + \Lambda^p \cdot t}\right).$$

If we want above probability to be at most some small value $\tau$, it suffices for $\frac{t^2}{k \cdot \sum a_i^2} \gtrsim \log \frac{1}{\tau}$ and $\frac{t}{\Lambda^p} \gtrsim \log \frac{1}{\tau}$. Noting that $\sum a_i^2 \leq \max a_i \cdot \sum a_i \lesssim \text{cost}(\mathcal{X}; \mathcal{C}) \cdot \Lambda^p$, it suffices for $t \gtrsim \Lambda^p \log \frac{1}{\tau} + \sqrt{k \log \frac{1}{\tau} \cdot \Lambda^p \cdot \text{cost}(\mathcal{X}; \mathcal{C})}$. By inequality of arithmetic and geometric means, we have that $\sqrt{k \log \frac{1}{\tau} \cdot \Lambda^p \cdot \text{cost}(\mathcal{X}; \mathcal{C})} \leq \frac{k}{\gamma} \log \frac{1}{\tau} \cdot \Lambda^p + \gamma \cdot \text{cost}(\mathcal{X}; \mathcal{C})$ for any $\gamma \in (0, 1]$. Therefore, for a sufficiently large constant $C$,

$$\mathbb{P}\left(|k \cdot \text{cost}(S; \mathcal{C}) - \text{cost}(\mathcal{X}; \mathcal{C})| \geq C\left(\frac{k}{\gamma} \log \frac{1}{\tau} \cdot \Lambda^p + \gamma \cdot \text{cost}(\mathcal{X}; \mathcal{C})\right)\right) \leq \tau.$$

Now, consider creating a $\frac{\Lambda}{n^{10}}$-sized net of $B(0, \Lambda)$, and choosing the $k$ centers in $\mathcal{C}$ from there. There are at most $n^{O(d \cdot k)} = e^{O(d \cdot k \cdot \log n)}$ ways of choosing such centers. In addition, for any set of $k$ centers $\mathcal{C} \subset B(0, \Lambda)$, by mapping each point to its closest point in the net, we do not change the cost of $S$ or $\mathcal{X}$ by more than $n \cdot 2\frac{\Lambda^p}{n^{10}} \leq \frac{\Lambda^p}{n^8}$. Therefore, the probability that $|k \cdot \text{cost}(S; \mathcal{C}) - \text{cost}(\mathcal{X}; \mathcal{C})| \geq C\left(\frac{k}{\gamma} \log \frac{1}{\tau} \cdot \Lambda^p + \gamma \cdot \text{cost}(\mathcal{X}; \mathcal{C})\right)$ for all sets $\mathcal{C}$ of size $k$ is at most $\tau \cdot e^{O(d \cdot k \cdot \log n)}$. Therefore, by replacing $\tau$ with $\frac{\tau}{e^{O(d \cdot k \cdot \log n)}}$, we have that

$$\mathbb{P}\left(\sup_{\substack{\mathcal{C} \subset B(0, \Lambda) \\ |\mathcal{C}| = k}} |k \cdot \text{cost}(S; \mathcal{C}) - \text{cost}(\mathcal{X}; \mathcal{C})| \geq C\left(\frac{k^2 d \log n}{\gamma} \log \frac{1}{\tau} \cdot \Lambda^p + \gamma \cdot \text{cost}(\mathcal{X}; \mathcal{C})\right)\right) \leq \tau.$$

Overall, given a dataset $\mathcal{X}$, we can subsample each element $x_i$ with probability $\frac{1}{k}$ and then solve private $k$-means or $k$-median using Theorem E.6. Due to the subsampling, we now have that with overwhelming probability, each machine only has $O(n^\theta/k)$ points, so the total sequential runtime has been improved to $\tilde{O}(nd)$ and the total time per machine has been improved to $\tilde{O}(n^\theta d)$. The multiplicative error blows up by an additional $1 + O(\gamma)$ factor, and if we think of $\gamma$ as a small but fixed constant, the additive error only increases by $\text{poly}(k, d, \log n)$ if we want our algorithm to succeed with high probability. Finally, the algorithm is still $(\varepsilon, \delta)$-DP by Lemma A.21.

Therefore, an improved version of Theorem E.6 holds where the total sequential runtime is $\tilde{O}(nd)$ and the total time per machine is $\tilde{O}(n^\theta d)$.

**Theorem E.9.** *Suppose that there exists a polynomial-time algorithm that can compute a $\rho$-approximation to $k$-means (resp., $k$-median). Then, for any constant $\rho' > \rho$, there exists an $(\varepsilon, \delta)$-DP algorithm for $k$-means (resp., $k$-median) with multiplicative error $\rho'$ and additive error $\text{poly}\left(k, d, \log n, \varepsilon^{-1}, \log \delta^{-1}\right)$. In addition, the algorithm can be implemented in MPC with $O(1)$ total rounds of communication, total sequential time $\tilde{O}(nd)$, and total time per machine $\tilde{O}(n^\theta d)$, assuming each machine can store $\tilde{O}(n^\theta) \geq \text{poly}(k, d, \log n, \varepsilon^{-1}, \log \delta^{-1})$ points.*