# OpenReview forum: "Near-Optimal Private and Scalable $k$-Clustering"
_NeurIPS.cc/2022/Conference — NeurIPS 2022 Accept_

### Official Review · Reviewer_eYNQ · 2022-06-30

**Rating:** 7
**Confidence:** 3
**Soundness:** 3 good
**Presentation:** 3 good
**Contribution:** 3 good

**Summary:**

The authors develop near-optimal private algorithms for k-means and k-median. Their algorithms boast either better approximation guarantees, or better additive error or they are more efficient, when compared with previous works. Moreover, they can be implemented in the MPC model with constant rounds of communication. The problem is relevant to the machine learning community, while the theoretical results are interesting and non-trivial.

**Questions:**

-

**Limitations:**

-

**Strengths And Weaknesses:**

+ the problem is relevant to the Neurips community, while the paper is well written
+ their algorithms boast either better approximation guarantees, or better additive error or they are more efficient, when compared with previous works. They can also be implemented in the MPC model with constant number of rounds. The charging argument developed in the paper might be of independent interest.

- the paper would have benefited from an experimental evaluation assessing the efficiency of their algorithms on real-world data

---

> ### Author Response · Authors · 2022-08-02
> **Response to Reviewer eYNQ**
>
> Thank you very much for your valuable feedback! Below is our response to your concern.
>
> **Experimental Evaluation:** Our paper was intended as a pure theory paper to develop private, accurate, and efficient (in the MPC model) clustering algorithms. We believe that experimental evaluation to verify accuracy and efficiency would be valuable as future work.

---

### Official Review · Reviewer_64md · 2022-07-02

**Rating:** 8
**Confidence:** 3
**Soundness:** 3 good
**Presentation:** 3 good
**Contribution:** 3 good

**Summary:**

This paper considers differentially private clustering in the massively parallel computation (MPC) model. In particular, this paper gives two private algorithms for $k$-means and $k$-median clustering that achieve constant-factor multiplicative and $\text{poly}(k,d)$ additive error, using a constant number of parallel computation rounds and near-linear runtime in $n$. The algorithms provide a trade-off between the relative approximation and the additive error.

**Questions:**

1. Do the techniques/results have any implications for local differential privacy?

2. Can the results be adjusted to give a coreset with $(1+\gamma)$ multiplicative approximation rather than $(1+\gamma)$ multiplicative approximation to the optimal non-private algorithm? That is, if determining the actual centers/clustering is not restricted to polynomial time, can the multiplicative approximation of the algorithm be boosted to $(1+\gamma)$, perhaps at a cost of additional $\frac{1}{\gamma}$ factors in the number $T$ of points sampled in each ring $\mathcal{X}_{j,r}$ in Algorithm 1?

3. Could you please elaborate more on the novel charging argument and potential applications of the techniques (the latter perhaps at an optimistic level)?

4. It seems the main reason for the roughly $O(nkd)$ runtime for the second algorithm is due to line 1092 mapping each point $x\in X$ to its region $\mathcal{T}_{j,r}$, which is a set of rings of growing radius around a set of centers. Can this assignment be done in $\tilde{O}(nd)$ time by using terminal embeddings, e.g., [MMR19,NN19,EFS+22]?

[MMR19] Konstantin Makarychev, Yury Makarychev, and Ilya P. Razenshteyn: Performance of Johnson-Lindenstrauss Transform for $k$-Means and $k$-Medians Clustering. STOC 2019

[NN19] Shyam Narayanan, Jelani Nelson: Optimal terminal dimensionality reduction in Euclidean space. STOC 2019

[EFS+22] Jon Ergun, Zhili Feng, Sandeep Silwal, David P. Woodruff, Samson Zhou:
Learning-Augmented $k$-means Clustering. ICLR 2022

Post-response update: Thanks for the follow-up explanation. In light of the improved runtime (as well as the responses to the questions raised by other reviewers), I have increased my score from 6 to 8. If space or time permits, please include a similar discussion/analysis/comparison of these works in the full version of the updated manuscript.

**Limitations:**

I do not foresee concerns about the limitations and potential negative societal impact of this work.

**Strengths And Weaknesses:**

Strengths: Previous private algorithms achieve either (1) constant factor multiplicative approximation and $\Gamma^p\cdot\text{poly}(k)$ additive error using $n^{1+\Omega(1)}$ time, where $\Gamma$ is the radius of the input space or (2) $\Omega(\log n)$ multiplicative and $\Gamma^p\cdot n^{\Omega(1)}$ approximation using near-linear time, i.e., roughly $O(nd)$ time, but not both simultaneously. The first algorithm given in this paper manages to achieve "the best of both worlds", getting both constant factor multiplicative approximation and $\Gamma^p\cdot\text{poly}(k)$ additive error using near-linear time, i.e., roughly $O(nd)$ time. Further achieving these results in the MPC model seems like an added bonus.

Weaknesses: The additive error of the algorithms in this paper have worse dependencies in $k$ than the state-of-the-art (when the latter is permitted polynomial time, which is perhaps an unfair comparison). The abstract also mentions "a novel charging argument which might be of independent interest" that I would have liked to hear more about. I understand that details of the analysis may be challenging to describe due to the space limits but given that the techniques are highlighted in the abstract, even a pointer in the first 9 pages to where the charging argument is described in the supplementary material would have been nice.

---

> ### Author Response · Authors · 2022-08-02
> **Response to Reviewer 64md**
>
> Thank you very much for your valuable feedback! Below is our response to your questions and concerns.
>
> **Local Differential Privacy:** Understanding what can be done in the massively parallel LDP setting is certainly an interesting future direction, but our current techniques do not seem to apply to this situation. The main reason is that a first step in our procedure is to split the points into rings, before we create a coreset by taking a random sample in each ring and privatizing. However, in the LDP setting we cannot get much information about the individual points, so we have no idea which points should go to which rings. In the central DP setting this is not a problem because the algorithm may know the full information about the data points as long as the final set of centers doesn’t violate the dataset’s privacy.
>
> **Obtaining a (1+gamma)-approximation:** Yes, in fact, our paper does get such a result. For instance, you can look at our theorem D.1 (lines 1066-1070 on pg 26) in the supplementary PDF, which exactly generates a (1 \pm gamma)-coreset. We have an exponential dependence on the dimension for additive error, but because of the dimension reduction framework of [MMR19], we show how to replace the (1/gamma)^d with k^{\tilde{O}(1/gamma^2)}. Hence we can obtain a (1 \pm gamma)-coreset with additive error roughly poly(k) for any fixed gamma > 0, which implies a (1 \pm gamma)-approximation with additive error poly(k) if we allowed for inefficient algorithms (but still required only O(1) rounds of MPC communication).
>
> **Regarding the Charging Argument (the same question also raised by Reviewer 8gPD):** We give some intuition near the at the bottom of page 4 (lines 164-177), and some more information in page 7 (lines 292-307).
>
> To explain the intuition further, we noted in the intuition paragraph that our algorithm selects a subsample of size T from each bucket. We use a Chernoff bound and some properties of high-dimensional Euclidean space to show that if there are m points in the bucket which are all within R of each other, we accumulate roughly R^2 * m/sqrt{T} additive error, as opposed to a more naive Lambda^2 * m/sqrt{T} error. We can then split the error between each of the m points in the bucket to get Lambda^2/sqrt{T} error per point. We charge the error of each point to the squared distance of each point to its center in the weak approximation, which is roughly equal to R^2. So, when we add over all points, the error across all points is roughly 1/sqrt{T} times the overall cost of the weak approximation. The weak approximation is roughly a d^{O(1)}-multiplicative approximation to the optimal cost (which becomes a (log n)^{O(1)}-multiplicative approximation after dimensionality reduction). So, if the sampling size T is much bigger than (log n)^{O(1)} the total error is much smaller (by a multiplicative factor) than even the optimal cost. Hence, we are able to charge the total large additive error into a small multiplicative error! The main message here is that we can group data points in a way, such that we can carefully control the additive error obtained by each point. At the same time we know the optimal cost is relatively large enough such that the total additive error can be charged as a small multiplicative error.
>
> Some of the above is already in the intuition paragraph, but we are happy to add the additional details to hopefully improve the readers’ understanding of the charging argument.
>
> **Potential applications of our charging technique:** We believe that our approach may also be useful for other approaches that rely on coresets, such as e.g.:
>
> Sepehr Assadi, MohammadHossein Bateni, Vahab S. Mirrokni. Distributed Weighted Matching via Randomized Composable Coresets. ICML 2019.
>
> Sepehr Assadi, MohammadHossein Bateni, Aaron Bernstein, Vahab S. Mirrokni, Cliff Stein. Coresets Meet EDCS: Algorithms for Matching and Vertex Cover on Massive Graphs. SODA 2019.
>
> **Improving the runtime (using possibly terminal embeddings):** We don’t know how to improve the runtime all the way to \tilde{O}(nd), but it should be improvable to \tilde{O}(ndk^{1-O(gamma)}) if the multiplicative error is gamma. This is because one only needs to map each point to a 1 \pm gamma approximate nearest neighbor, which can be done in time d*k^{1-O(gamma)} using, e.g., [AI06].
>
> The original terminal constructions, such as in [NN19], take poly(k) time for each point in X to map, so the overall time would be nd*poly(k) which is even worse. A recent paper [CN21] improved this to k^{1-O(gamma^2)} per point, but we believe this still means using terminal embeddings would not provide any advantage over just using approximate nearest neighbor algorithms.
>
> [AI06]: Alexandr Andoni and Piotr Indyk. Near-Optimal Hashing Algorithms for Approximate Nearest Neighbor in High Dimensions. FOCS 2006.
>
> [CN21]: Yeshwanth Cherapanamjeri and Jelani Nelson. Terminal Embeddings in Sublinear Time. FOCS 2021.

---

> > ### Comment · Reviewer_64md · 2022-08-04
> > **On Author Response**
> >
> > Thanks for the response and in particular, for resolving my questions about LDP and overall $(1+\gamma)$-multiplicative approximation. I still have the concerns about the responses to novel charging argument and the terminal embedding questions, however. Perhaps the authors could clarify on the following:
> >
> > The charging argument seems to be partitioning the points into rings so that the "contribution" for each point in a ring with respect to the assigned center (perhaps in a bicriteria algorithm) is roughly the same, which allows the variance of sampling procedures to be controlled easier. How does this approach differ from prior work that seemingly similarly partition the points into rings around each assigned center, e.g., in the bottom of page 24 of [CLSS22] or in the bottom of page 12 in the arXiv version of [CSS21]?
> >
> > [CLSS22] Vincent Cohen-Addad, Kasper Green Larsen, David Saulpic, Chris Schwiegelshohn:
> > Towards Optimal Lower Bounds for k-median and k-means Coresets. CoRR abs/2202.12793, 2022, https://arxiv.org/pdf/2202.12793.pdf
> >
> > [CSS21] Vincent Cohen-Addad, David Saulpic, Chris Schwiegelshohn: A new coreset framework for clustering. STOC 2021: 169-182
> >
> > Regarding runtime improvement, the author response indicated that the runtime can be improved to $\tilde{O}(ndk^{1-O(\gamma)})$ if the multiplicative error is $\gamma$ because each point just needs to be mapped a $(1 \pm\gamma)$ approximate nearest neighbor, which can be done in time $dk^{1-O(\gamma)}$. The response also notes that recent terminal embedding constructions require $\text{poly}(k)$ time for each point in $X$ to map, but then concludes that the overall time would be $nd\text{ poly}(k)$.
> >
> > I do not think this last conclusion is tight. In particular, the authors already note in first step in the framework of line 154 that the dimension can first be reduced to $O(\log k)$ using a terminal embedding. Then the nearest neighbor search can be performed in dimension $O(\log k)$ rather than dimension $d$, at the cost of a slight distortion due to the dimensionality reduction. However, this distortion can be absorbed into the distortion by the nearest neighbor search. Moreover, the overall runtime would now be $\tilde{O}(nd+nk)$ since the nearest neighbor search is now performed in dimension $O(\log k)$. This seems to be the approach currently being used by Algorithm 3 in the arXiv version of [EFS+22] and I'm curious whether it could also be applied here.
> >
> > [EFS+22] Jon Ergun, Zhili Feng, Sandeep Silwal, David P. Woodruff, Samson Zhou: Learning-Augmented $k$-means Clustering. ICLR 2022
> >
> > On a separate note, do there need to be privacy considerations for the dimensionality reduction and nearest neighbor search procedures or is that black-boxed into the constant-factor private sequential algorithm? I'm aware of the Johnson-Lindenstrauss transformation preserving privacy but not sure about terminal embeddings.

---

> > > ### Comment · Reviewer_64md · 2022-08-07
> > > **On Author Response**
> > >
> > > I wanted to check whether my understanding in the last exchange was correct or if I have a misunderstanding of the algorithmic approach of this paper. Let me know either way, thanks!

---

> > > > ### Author Response · Authors · 2022-08-07
> > > > **Response to Updated Concerns of Reviewer 64md**
> > > >
> > > > Below are our responses to your updated concerns. Thank you for your patience!
> > > >
> > > > **On the charging argument for coresets:** The notion of rings for computing coresets was in fact introduced by Chen [Che09]. The recent works [CSS21,CLSS22] that you mentioned have indeed used the notion of rings of Chen: their contributions lie in the use made of these rings and how they are further partitioned into groups to reduce the overall variance. Our use of the rings is different from the one in [CSS21,CLSS22] -- in fact it is unclear how to make use of the notion of groups of [CSS21,CLSS22] in the DP setting.
> > > >
> > > > In our case, the main novelty is a different analysis of the rings which allows us to convert additive error into multiplicative error. Indeed, unlike in the standard setting (as in [Che09, CSS21,CLSS22]), the DP setting requires some additive error, and a naive analysis of the rings will contribute some additive error for each machine in the DP setting. Since there are n^{theta} machines, this means the additive error would be proportional to n^{theta}, which is too large. However, we show that for most machines (except for roughly k of them), the additive error induced by each ring can be absorbed by the multiplicative error of the ring, which improves our error. The [Che09, CSS21, CLSS22] papers never had additive error to begin with (since they are not DP), so they did not have to deal with this issue.
> > > >
> > > > [Che09]: On coresets for k-median and k-means clustering in metric and Euclidean spaces and their applications. SIAM J. Comput., 2009.
> > > >
> > > > **On terminal embeddings and a faster runtime:** Regarding improving the runtime to $O(nd + poly(k))$ rather than $O(ndk+poly(k))$, we now believe it is doable, even without terminal embeddings. Thank you very much for asking again about this!
> > > >
> > > > When obtaining a constant ($O(1)$) approximation, we were able to do it in $O(nd + poly(k))$ time. We subsampled each point with probability $1/k^{0.01}$ and then applied an approximate nearest neighbor data structure which took $k^{0.01} d$ time per point, but the subsampling means we only need to do this for $n/k^{0.01}$ points. This multiplies the additive approximation by $k^{0.01}$, but we treat this as small. When trying to get an arbitrarily good approximation, we need to use an exact (or $(1+\gamma)$-approximate) nearest neighbor which takes roughly $O(kd)$ time per point, so we need to sample each point with probability $1/k$ instead. We believe this will blow up the additive approximation by a factor of $k$, but that is not bad because the additive approximation in this case is anyway $k^{O(1/\gamma^2)}$. When originally working on this paper, we thought the additive approximation has a $\sqrt{n}$ multiplicative factor by some rough calculation (just in the arbitrarily good approximation setting), but this is not the case. So we can actually do better than what we previously thought. We will add the explanation for the improved runtime in the final version of our paper.
> > > >
> > > > We note none of this actually requires terminal embeddings, and just needs subsampling and nearest neighbor calculations. The paper [EFS+22] you mentioned, while it can map the centers in $poly(k)$ time, it needs $O(kd)$ time for each non-center point. However, they subsample every point with probability $1/k$ (see line 1 of the algorithm 3, and Theorem 3.4 which assumes every $|A_x|$ has size at least $\gamma \cdot k \log k/\alpha)$ which is similar to what we need to do.

---

> > > > > ### Comment · Reviewer_64md · 2022-08-07
> > > > > **Acknowledgement**
> > > > >
> > > > > Thanks for the follow-up explanation. In light of the improved runtime (as well as the responses to the questions raised by other reviewers), I have increased my score from 6 to 8. If space or time permits, please include a similar discussion/analysis/comparison of these works in the full version of the updated manuscript.

---

> > > > > > ### Author Response · Authors · 2022-08-08
> > > > > > **Thank you!**
> > > > > >
> > > > > > Yes, we will definitely include this discussion (both about the faster analysis, and how our charging argument is novel) in the paper. Hopefully the addition to the charging argument intuition can go in the main body, we'll probably just give a pointer to the analysis for the O(nd + poly(k)) time analysis, and update the supplementary section.
> > > > > >
> > > > > > Thank you very much both for your very thoughtful and constructive criticism, and for updating your score.

---

### Official Review · Reviewer_8gPD · 2022-07-08

**Rating:** 8
**Confidence:** 3
**Soundness:** 3 good
**Presentation:** 4 excellent
**Contribution:** 3 good

**Summary:**

The authors address the problem of differentially-private clustering in the massively parallel computation model. They provide the first algorithms that simultaneously achieve
- differential privacy,
- constant factor approximation,
- near-optimal additive error of poly(k,d) or poly(kd),
- near-linear running time.
The algorithms need a constant number of rounds in the MPC model.

As I understand it, the key to the improved running time is devising a distributed algorithm that can manage with the local $n^\theta$ memory of the $n^{1-\theta}$ available machines.



**Questions:**

One thing to note is that the authors might want to devote more space to the intuition of the charging argument, which seems to be one of the most innovative elements. Other parts of the algorithms rely heavily on known data structures and DP tricks.

**Limitations:**

The authors have adequately addressed the limitations and potential negative societal impact of their work?

**Strengths And Weaknesses:**

The paper is well written and presented, even featuring brief discussions of why straightforward ideas do not work, which is appreciated.

The results are a clear improvement with respect to the state of the art. In particular, they improve substantially w.r.t. previous results in the MPC model.

I do not have any strong objections, bearing in mind that I have not read the appendix.

---

> ### Author Response · Authors · 2022-08-02
> **Response to Reviewer 8gPD**
>
> Thank you very much for your valuable feedback! Below is our response to your questions and concerns.
>
> **Regarding the Charging Argument:** We give some intuition near the bottom of page 4 (lines 164-177), and some more information in page 7 (lines 292-307).
>
> To explain the intuition further, we noted in the intuition paragraph that our algorithm selects a subsample of size T from each bucket. We use a Chernoff bound and some properties of high-dimensional Euclidean space to show that if there are m points in the bucket which are all within R of each other, we accumulate roughly R^2 * m/sqrt{T} additive error, as opposed to a more naive Lambda^2 * m/sqrt{T} error. We can then split the error between each of the m points in the bucket to get Lambda^2/sqrt{T} error per point. We charge the error of each point to the squared distance of each point to its center in the weak approximation, which is roughly equal to R^2. So, when we add over all points, the error across all points is roughly 1/sqrt{T} times the overall cost of the weak approximation. The weak approximation is roughly a d^{O(1)}-multiplicative approximation to the optimal cost (which becomes a (log n)^{O(1)}-multiplicative approximation after dimensionality reduction). So, if the sampling size T is much bigger than (log n)^{O(1)} the total error is much smaller (by a multiplicative factor) than even the optimal cost. Hence, we are able to charge the total additive error into a small multiplicative error! The main message here is that we can group data points in a way, such that we can carefully control the additive error obtained by each point. At the same time we know the optimal cost is relatively large enough such that the total additive error can be charged as a small multiplicative error.
>
> Some of the above is already in the intuition paragraph, but we are happy to add the additional details to hopefully improve the readers’ understanding of the charging argument.
>
> **Limitations:** We are not sure if you were asking a question or not. Assuming you were, we remark that we are presenting a privacy preserving paper, and that differential privacy is the gold standard of privacy but it must be used correctly to protect user privacy. Importantly, differential privacy must be provably maintained to be implemented in practice. We are not aware of any negative consequences or negative social impacts of our work.

---

### Official Review · Reviewer_GEMX · 2022-07-14

**Rating:** 7
**Confidence:** 3
**Soundness:** 4 excellent
**Presentation:** 4 excellent
**Contribution:** 3 good

**Summary:**

The paper gives improved guarantees for differential private computation of $k$-means and $k$-median clustering of $n$ points in the unit ball of $d$-dimensional Euclidean space. There are three concerns: the approximation guarantee, the additive error, and the parallel runtime and work. This work achieves simultaneously the best (or comparable to the best) known guarantees for all three: $O(1)$ approximation, $\hbox{poly}(k,d)$ additive error, running in $O(1)$ parallel rounds and total work nearly linear in $n$ and polynomial in $k$. The method is to compute a bicriteria approximation using quad trees (embedded into HSTs), then apply to the excess number of centers any constant factor approximation private algorithm. To eliminate the dependence on $\hbox{exp}(d)$, these steps are preceded by dimension reduction to $O(\log k)$ dimensions.

**Questions:**

None.

**Limitations:**

None.

**Strengths And Weaknesses:**

It's an interesting result on a problem that has been studied recently quite intensively. The results are somewhat incremental, mostly with respect to the techniques used, but the bottom line, especially the efficient parallel computation, seems very interesting.

---

> ### Author Response · Authors · 2022-08-02
> **Response to Reviewer GEMX**
>
> Thank you very much for your valuable feedback!

---

### Meta-Review · Area_Chair_WXkc · 2022-08-25

**Recommendation:** Accept
**Confidence:** Certain

**Metareview:**

The paper gives new algorithms for k-means and k-median clustering with differential privacy in the massively parallel computation (MPC) model. These are fundamental clustering problems and the MPC model is a common abstraction for relevant distributed systems such as Hadoop, Spark. The new algorithms have approximation factors and additive errors close to those of best known algorithms for the single machine setting, with a constant number of rounds of communication and near linear total work. All of these parameters are close to optimal, except for possibly a few factors of k, the number of clusters. All reviewers appreciate the new solution to important problems in a relevant setting, especially the near optimality in several dimensions at once: the number of rounds, the total work, the approximation factor, and the additive error.

**Award:**

No

---

### Decision · Program_Chairs · 2022-09-14

Accept